# VisOnlyQA: Large Vision Language Models Still Struggle with Visual Perception of Geometric Information

**Ryo Kamoi, Yusen Zhang, Sarkar Snigdha Sarathi Das, Ranran Haoran Zhang, Rui Zhang**
Penn State University
{ryokamoi, rmz5227}@psu.edu

## Abstract

Large Vision Language Models (LVLMs) have achieved remarkable performance in various vision-language tasks. However, it is still unclear how accurately LVLMs can perceive visual information in images. In particular, the capability of LVLMs to perceive geometric information, such as shape, angle, and size, remains insufficiently analyzed, although the perception of these properties is crucial for tasks that require a detailed visual understanding. In this work, we introduce VisOnlyQA, a dataset for evaluating the geometric perception of LVLMs, and reveal that LVLMs often cannot accurately perceive basic geometric information in images, while human performance is nearly perfect. VisOnlyQA consists of 12 tasks that directly ask about geometric information in geometric shapes, charts, chemical structures, and 3D shapes. Our experiments highlight the following findings: (i) State-of-the-art LVLMs struggle with basic geometric perception. 23 LVLMs we evaluate, including GPT-4o and Gemini 2.5 Pro, work poorly on VisOnlyQA. (ii) Additional training data does not resolve this issue. Fine-tuning on the training set of VisOnlyQA is not always effective, even for in-distribution tasks. (iii) LLM may be the bottleneck. LVLMs using stronger LLMs exhibit better geometric perception on VisOnlyQA, while it does not require complex reasoning, suggesting that the way LVLMs process information from visual encoders is a bottleneck. The datasets, code, and model responses are provided at https://github.com/psunlpgroup/VisOnlyQA.

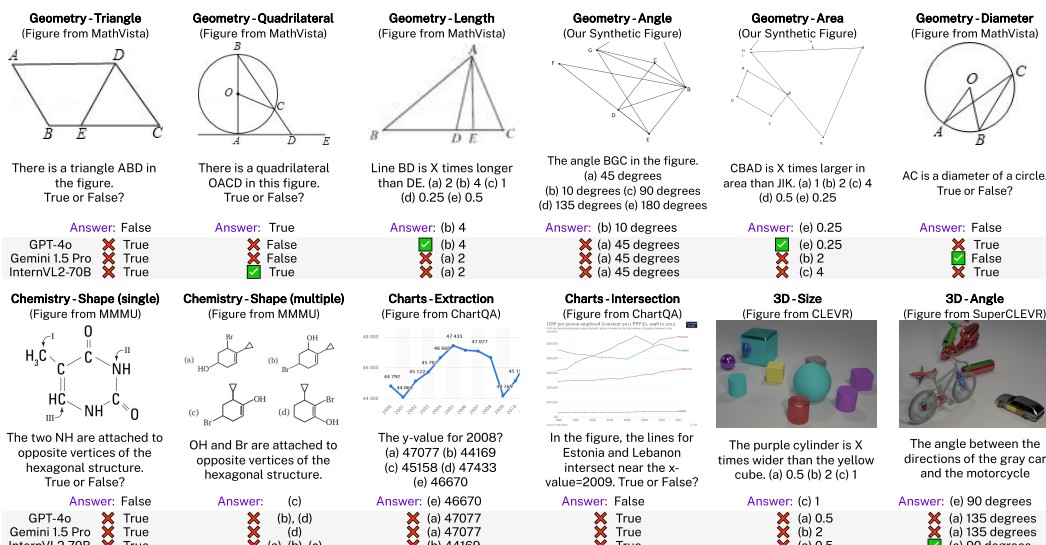

Figure 1: Examples from 12 tasks in VisOnlyQA and answers from LVLMs. Figures in VisOnlyQA are from existing datasets or generated by us, and all questions are created by us. Questions in this figure are abbreviated. Refer to Appendix H for full inputs and responses.

# 1   Introduction

Large Vision Language Models (LVLMs) have demonstrated significant advancement across a range of challenging vision-language tasks that require expert-level reasoning and knowledge (Liu et al., 2024; Chen et al., 2024b; OpenAI, 2024a). However, their ability to perceive visual information in images has not been sufficiently studied (Zhang et al., 2024a; Li et al., 2024). Specifically, it remains unclear how accurately LVLMs can perceive geometric information, such as shape, angle, and size, while geometric perception is fundamental to understanding visual information in images and is commonly required in vision-language tasks (Balachandran et al., 2024; Gao et al., 2025; Xing et al., 2025).

A primary obstacle to studying the geometric perception of LVLMs lies in the absence of a dataset suitable for analyzing this capability, as in Table 1. (1) Recent popular datasets for evaluating LVLMs, such as MMMU (Yue et al., 2024) and MathVista (Lu et al., 2024), target tasks that require expert-level reasoning and knowledge. The performance of LVLMs on these datasets is largely affected by multiple capabilities and is not suitable for analyzing specific capabilities. (2) While there exist datasets designed for evaluating LVLMs at perceiving visual information in images (Antol et al., 2015; Goyal et al., 2017; Li et al., 2024), they often evaluate LVLMs in high-level comprehension tasks, such as scene understanding, which are not suitable for analyzing geometric perception and also do not necessarily require accurate perception of geometric information.

In this work, we propose VisOnlyQA, a new dataset designed to evaluate how accurately LVLMs can perceive basic geometric information in images. As in Figure 1, our dataset includes questions that directly ask about basic and common geometric information (e.g., length, angle, and shape) in diverse scientific figures, including geometric shapes, chemical structures, charts, and 3D shapes. VisOnlyQA has favorable properties for analyzing the capability of LVLMs to perceive geometric information: (1) The questions in our dataset do not involve challenging reasoning or knowledge, enabling us to exclusively evaluate the geometric perception of LVLMs independent of other capabilities. (2) We use scientific figures to create unambiguous questions that directly ask about geometric information in images, which require accurate geometric perception.

We evaluate 23 LVLMs and observe that state-of-the-art LVLMs, including GPT-4o and Gemini 2.5 Pro, perform poorly in the basic geometric perception tasks in VisOnlyQA (48.8% and 79.0% in accuracy on the Real split), while human performance is nearly perfect (93.5%), as in Figure 2. This result indicates that existing LVLMs often cannot accurately perceive common geometric information in images, such as shape, angle, and size (§4.1). In addition, we observe that this limitation persists even on simple geometric shapes consisting of only two or three lines (§4.2). This finding raises concerns about the faithfulness of LVLMs to visual input in vision-language tasks.

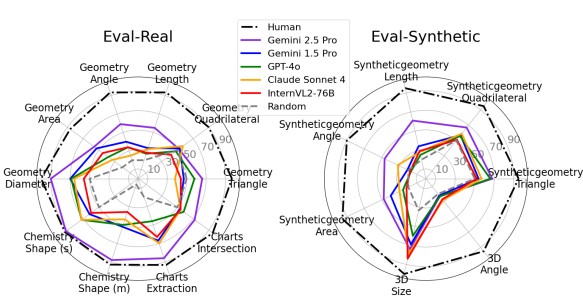

Figure 2: LVLMs perform poorly on VisOnlyQA, while human performance is nearly perfect. Table 5 provides detailed results.

To explore approaches to improve the capability of LVLMs to perceive geometric information, we evaluate LVLMs fine-tuned on the training set of VisOnlyQA. We observe that fine-tuning largely improves performance in some tasks and models, indicating that the lack of training data is a part of the reason why LVLMs cannot accurately perceive geometric information. However, at the same time, fine-tuning does not always improve their performance on VisOnlyQA, even for in-domain data, and our result shows that task properties and model size largely influence the performance after fine-tuning (§4.4). While these findings suggest that enhancing the geometric perception of LVLMs is not straightforward, our experiments also indicate a trend with implications for future improvement — LVLMs using larger

| Dataset | Require Accurate Geometric Perception | Specifically Targeting Visual Perception | Decoupled Evaluation of Geometric Perception | Image Categories | Question Categories |
|---|---|---|---|---|---|
| MM-Vet (2024) | △ | ✗ | ✗ | General Figures | Scene understanding, Math |
| SEED-Bench (2024) | △ | ✓ | ✗ | General Figures | Scene understanding |
| CharXiv (2024b) | △ | ✗ | ✗ | Charts | Math, Information extraction |
| MathVista (2024) | △ | ✗ | ✗ | Math, Synthetic | Math |
| MMMU (2024) | △ | ✗ | ✗ | Math, Academic, Charts | Academic exams |
| VisOnlyQA (ours) | ✓ | ✓ | ✓ | Math, Chemistry, Charts, 3D | Geometric information |

Table 1: VisOnlyQA is designed to evaluate the ability of LVLMs to perceive geometric information while removing the influence of other capabilities like reasoning. Popular datasets often evaluate multiple capabilities simultaneously, making them unsuitable for analyzing a specific capability. Existing datasets for evaluating visual perception often target high-level tasks and do not require the accurate perception of geometric information.

language models exhibit better performance on VisOnlyQA when using the same visual encoders. This is a counterintuitive result because our dataset evaluates the perception of geometric information, which does not involve complex reasoning or knowledge. This finding suggests that the way to process visual information encoded by visual encoders is a bottleneck to understanding geometric information in images, and strong language models are required for LVLMs to effectively process visual information (§4.5).

In summary, VisOnlyQA reveals that current LVLMs still lack the capability to accurately perceive basic geometric information, such as shape, angle, and size, and simply scaling model size or training data is insufficient to fully overcome this limitation.

## 2 Related Work

**Datasets for evaluating the geometric perception of LVLMs.** HallusionBench (Guan et al., 2024) and IllusionVQA (Shahgir et al., 2024) show that LVLMs exhibit poor perception of geometric information in misleading figures, such as illusive geometric shapes. Our work, in contrast, focuses on images and questions reflecting common and practical applications. Fu et al. (2024) evaluate LVLMs by assessing the performance gaps when providing figures and text that include identical information (e.g., chess games in text and image representations). Our dataset provides a more direct way to evaluate and analyze geometric perception. Gao et al. (2025) reports that GPT4-V suffers from hallucinations when describing geometric shapes. Our dataset offers a more detailed analysis and shows that more recent models struggle with perceiving geometric information, even in much simpler tasks. GePBench (Xing et al., 2025) is a contemporaneous study with a similar motivation while specifically focusing on the relationships between multiple geometric shapes.

**Datasets that evaluate multiple capabilities of LVLMs.** Popular datasets for evaluating LVLMs often target tasks that involve complex reasoning or knowledge, such as mathematical reasoning (Lu et al., 2021a; Chen et al., 2021; 2022; Lu et al., 2024; Gupta et al., 2024), chart understanding (Kafle et al., 2018; Methani et al., 2020; Masry et al., 2022), and academic exams (Lu et al., 2022a;b; Yue et al., 2024). While these tasks also often involve the perception of geometric information in images, performance on these datasets is largely influenced by multiple capabilities, making them unsuitable for isolating and analyzing specific capabilities of LVLMs.

**Datasets that evaluate the visual perception of LVLMs.** Various datasets have been proposed to evaluate the visual perception of LVLMs, and they also often require geometric perception. However, popular datasets (Antol et al., 2015; Goyal et al., 2017; Gurari et al., 2018; Fu et al., 2023; Liu et al., 2025; Xu et al., 2025; Li et al., 2024) often do not require accurate perception of geometric information, as they mainly target tasks that ask for a high-level understanding of images, such as object recognition and scene understanding. This is potentially due to the difficulty in creating unambiguous questions that directly ask about geometric information on general images. In this work, we target scientific figures because they enable us to annotate unambiguous questions about geometric information.

| | Geometry | | | | | | Chemistry | | Charts | | 3D | | Total |
|---|---|---|---|---|---|---|---|---|---|---|---|---|---|
| | Triangle | Quadri-lateral | Length | Angle | Area | Diameter | Shape (s) | Shape (m) | Extra-ction | Inter-section | Size | Angle | |
| Eval-Real | 100 | 100 | 100 | 100 | 100 | 100 | 50 | 50 | 100 | 100 | – | – | 900 |
| Eval-Synthetic | 100 | 100 | 100 | 100 | 100 | – | – | – | – | – | 100 | 100 | 700 |
| Train | 10k | 10k | 10k | 10k | 10k | – | – | – | – | – | 10k | 10k | 70k |
| Answer Format | True/ False | True/ False | True/ False | 5 options | 5 options | 5 options | True/ False | Select Multiple | 5 options | True/ False | 3 options | 5 options | |

Table 2: Dataset statistics of VisOnlyQA. VisOnlyQA-Eval-Real includes figures in existing datasets and human-annotated questions. VisOnlyQA-Eval-Synthetic and VisOnlyQA-Train comprise synthetic figures and automatically generated questions.

| Number of Lines | | 2 | 3 | 4 | 5 | 6 |
|---|---|---|---|---|---|---|
| | Triangle | – | 50 | 50 | 50 | 50 |
| Geometry | Length | 50 | 50 | 50 | 50 | 50 |
| | Angle | 50 | 50 | 50 | 50 | 50 |

(a) Geometric shapes with different numbers of lines.

| Angle between Two Lines | | 0 | 45 | 90 |
|---|---|---|---|---|
| Geometry | Length | 50 | 50 | 50 |

(b) Length task with different angles between two lines.

Table 3: Statistics of datasets for analysis (Figure 4), which are based on Eval-Synthetic.

## 3 VisOnlyQA **Dataset**

We introduce VisOnlyQA, a new dataset designed to evaluate and analyze the capability of LVLMs to perceive geometric information in images, such as shape, angle, and size. Each instance of VisOnlyQA consists of a figure, a multiple-choice question, and an answer label. As in Figure 1 and Table 2, VisOnlyQA includes 12 tasks on figures in four categories: geometric shapes, chemical structures, charts, and 3D shapes.

**What are the favorable properties of our dataset?** We design VisOnlyQA to be suitable for analyzing the capability of LVLMs to perceive geometric information in images. Specifically, our dataset includes questions that directly ask for a precise perception of basic geometric information in scientific figures. (1) This approach prevents questions from demanding challenging reasoning or knowledge. As a result, the perception of geometric information is the only bottleneck for recent LVLMs when solving tasks in VisOnlyQA, and their performance on this dataset is not largely influenced by other capabilities (§4.3). This property of our dataset enables a direct evaluation of the geometric perception of LVLMs independent of other capabilities. (2) In addition, scientific figures enable us to create unambiguous questions that directly ask about geometric information in images.

**How does our dataset help understand LVLM behavior on general images?** While our dataset targets scientific figures to make unambiguous questions that directly ask about geometric information, it targets fundamental geometric information commonly required to understand the details of broad types of images, including real-world images (Xing et al., 2025). As our dataset includes simple and basic tasks involving common geometric perception, poor performance on VisOnlyQA raises concerns about the reliability of LVLMs in real-world vision-language tasks, not only in scientific domains.

### 3.1 Sources of Figures

VisOnlyQA includes two types of figures: Real and Synthetic. The **Real** figures are from existing datasets. We use figures in popular datasets to evaluate whether LVLMs truly understand images in those datasets. It also ensures that images are from real-world distributions and not adversarially created. Although we use existing images, all questions in VisOnlyQA are newly annotated. The **Synthetic** figures are automatically generated. The primary purpose of synthetic figures is to provide large-scale training data to analyze fine-tuned models. In addition, for evaluation, they ensure that there is no bias caused by human annotations because both images and questions are synthetically generated.

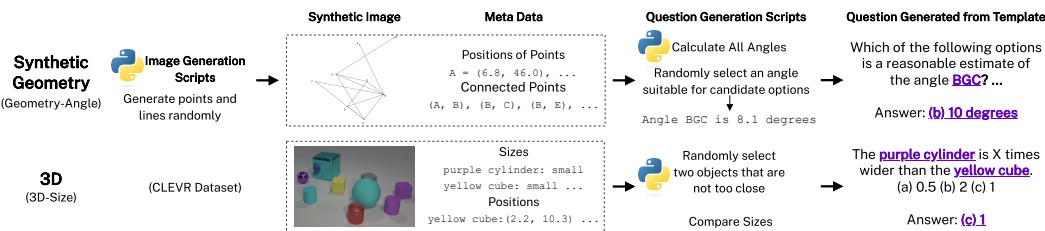

Figure 3: Construction process of synthetic images and questions in VisOnlyQA-Eval-Synthetic and VisOnlyQA-Train. This process does not involve language models and uses precise metadata, guaranteeing the correctness of generated question-answer pairs.

**Real Figures.** We use figures in popular datasets: **geometric shapes** in MathVista (Lu et al., 2024), which includes Geometry3K (Lu et al., 2021a), GeoQA+ (Cao & Xiao, 2022), GEOS (Seo et al., 2015), and UniGeo (Chen et al., 2022), **chemistry** figures in MMMU (Yue et al., 2024), and **charts** in ChartQA (Masry et al., 2022) and CharXiv (Wang et al., 2024b).

**Synthetic Figures.** For **geometric shapes**, we create a new dataset, SyntheticGeometry, by generating geometric shapes by writing Python scripts based on an open source project (Leeb, 2024) reproducing AlphaGeometry (Trinh et al., 2024). For **3D shapes**, we use CLEVR (Johnson et al., 2017) and SuperCLEVR (Li et al., 2023c).

## 3.2 Question Annotation

**For Real figures — Human annotation.** We manually annotate questions and answers for ten tasks on the Real figures. We provide question templates and instructions to annotators, and each question-answer pair is annotated by one annotator and verified by another annotator. The annotators are PhD students specializing in natural language processing.

**For Synthetic figures — Synthetic questions.** We generate synthetic questions using the metadata of the synthetic figures and question templates, as in Figure 3. For **geometric shapes**, we use the metadata in SyntheticGeometry, including the positions of points and lines. We write Python scripts to compute the geometric information (shape, length, angle, and area) from the metadata and generate question-answer pairs for five tasks. For **3D shapes**, we write Python scripts to generate question-answer pairs about the relative sizes of objects in CLEVR (3D-Size) and angles between objects in SuperCLEVR (3D-Angle) using the metadata in CLEVR (positions and sizes) and SuperCLEVR (positions and angles).

## 3.3 Dataset Design and Statistics

**Data split.** VisOnlyQA includes three splits: Eval-Real, Eval-Synthetic, and Train. Eval-Real includes 900 instances for ten tasks on three categories of figures from existing datasets (geometry, chemistry, and charts). Eval-Synthetic and Train include 700 and 70k instances for seven tasks on two categories of synthetic figures (geometry and 3D).

**Analysis dataset.** In addition to the main dataset, we provide a dataset based on Eval-Synthetic for a detailed analysis. We create datasets consisting of simple geometric shapes with a different number of lines and the Length task with different angles between two lines, as shown in Table 3 and Figure 4.

**Reducing biases.** To make label distribution balanced, we instructed the annotators to make an equal number of questions for each option. We also shuffled the options to remove biases caused by the order of the options. In addition, to avoid biases caused by the wording, we include a negative version of the questions for all true or false questions (e.g., *There is a triangle ABC in this figure* and *There is no triangle ABC in this figure*).

**Annotation quality.** We manually evaluate 100 randomly selected instances from the Eval-Real set and confirm that all cases contain valid questions with correct answers, demonstrating the high quality of our dataset.

**Human performance.** We provide randomly sampled questions to three new annotators (300 instances in total) for the Real split and two new annotators (140 instances in total) for the Synthetic split. The average human performance is 93.5% and 95.0% in accuracy (5), showing that VisOnlyQA is easy for humans. We observe that most human errors are due to their mistakes, such as misreading the order of symbols, rather than issues in the dataset.

**Geometric shapes.** Table 4 shows statistics of geometric shapes in VisOnlyQA. For the Real figures, we manually annotate 24 figures to get the statistics. For the Synthetic figures, we calculate the statistics of all images. Geometric shapes in the Synthetic data include more points, lines, and circles, with larger standard deviations.

| | # Points | # Lines | # Circles |
|---|---|---|---|
| Real | 5.0 ($\pm$ 1.3) | 5.8 ($\pm$ 2.2) | 0.2 ($\pm$ 0.4) |
| Synthetic | 9.3 ($\pm$ 3.2) | 10.6 ($\pm$ 4.5) | 0.4 ($\pm$ 0.5) |

Table 4: Average number ($\pm$ std dev) of points, lines, circles in geometric shapes in VisOnlyQA.

## 4 Experiments

We evaluate 23 open and proprietary LVLMs and five fine-tuned LVLMs on VisOnlyQA. Our experiments aim to answer the following research questions:

- **RQ1:** Can existing LVLMs accurately perceive geometric information? (§4.1, 4.2)
- **RQ2:** Does VisOnlyQA evaluate the capability to perceive geometric information independent of other capabilities, such as reasoning and knowledge? (§4.3)
- **RQ3:** Does additional training data improve the geometric perception of LVLMs? (§4.4)
- **RQ4:** Do language models of LVLMs influence their geometric perception? (§4.5)

**Models:** We evaluate 23 LVLMs in 9 model families, including **15 open models**: Phi-3.5-Vision (Microsoft, 2024), LLaVA-Next (8B, 34B) (Li et al., 2025), Llama 3.2-Vision (11B, 90B) (Meta, 2024), Molmo (7B-D, 72B) (Deitke et al., 2025), Qwen2-VL (2B, 7B, 72B) (Wang et al., 2024a), InternVL2 (4B, 8B, 26B, 40B, 76B) (OpenGVLab Team, 2024); and **8 proprietary models**: Claude Sonnet 3.5 (Anthropic, 2024), Sonnet 4, Opus 4 (Anthropic, 2025), GPT-4o-mini, GPT-4o (OpenAI, 2024a;b), Gemini-1.5 Flash and Pro (Google, 2024), and Gemini 2.5 Pro (Google, 2025). Refer to Appendix B for details.

**Prompts:** We evaluate two types of zero-shot prompts: with and without chain-of-thought reasoning (Wei et al., 2022; Kojima et al., 2022). Full prompts are in Appendix C.

### 4.1 LVLMs Cannot Accurately Perceive Basic Geometric Information in VisOnlyQA

Table 5 shows the accuracy of LVLMs on Eval-Real and Eval-Synthetic (with no chain-of-thought). The performance of LVLMs is far from perfect on all tasks, with the best average accuracies of 79.0% and 55.4% by Gemini 2.5 Pro on the Real and Synthetic splits, while human performance is nearly perfect (93.5% and 95.0%).

Our results show that larger LVLMs exhibit better capability in perceiving geometric information but also indicate that simply scaling model size does not lead to human-level performance. Specifically, even large models perform near-randomly on some tasks, including `Geometry-Triangle`, `Quadrilateral`, and `Charts-Intersection` in the Real split, as well as `Geometry-Angle` and `3D-Angle` in the Synthetic split. Gemini 2.5 Pro is the only model that achieves high performance on chemistry and chart figures, but it still exhibits near-random performance on most tasks involving geometric shapes. This is a cautionary observation, indicating that existing LVLMs still cannot accurately perceive basic geometric information, such as angle, shape, and intersection. Appendix H provides examples of model responses.

### 4.2 LVLMs Exhibit Poor Geometric Perception Even on Simple Geometric Shapes

Results in Section 4.1 show that LVLMs exhibit poor geometric perception capabilities. To further analyze this limitation, we create a dataset for analysis (Table 3) that includes simple geometric shapes with different complexities and tasks with different difficulties.

| | Geometry | | | | | | Chemistry | | Charts | | Average |
|---|---|---|---|---|---|---|---|---|---|---|---|
| | Triangle | Quadri-lateral | Diameter | Length | Angle | Area | Shape (s) | Shape (m) | Extraction | Inter-section | |
| Random | 50.0 | 50.0 | 20.0 | 20.0 | 20.0 | 50.0 | 50.0 | 6.2 | 20.0 | 50.0 | 34.2 |
| Phi-3.5-vision | 48.0 | 50.0 | 17.0 | 17.0 | 27.0 | 50.0 | 54.0 | 10.0 | 29.0 | 50.0 | 35.6 |
| LLaVA-Next 8B | 50.0 | 50.0 | 16.0 | 15.0 | 26.0 | 49.0 | 42.0 | 4.0 | 22.0 | 49.0 | 33.3 |
| LLaVA-Next 34B | 49.0 | 50.0 | 30.0 | 15.0 | 22.0 | 44.0 | 34.0 | 10.0 | 35.0 | 50.0 | 35.2 |
| Llama 3.2 11B | 50.0 | 47.0 | 17.0 | 15.0 | 26.0 | 43.0 | 34.0 | 8.0 | 32.0 | 50.0 | 33.4 |
| Llama 3.2 90B | 51.0 | 46.0 | 14.0 | 28.0 | 27.0 | 48.0 | 60.0 | 20.0 | 35.0 | 45.0 | 37.1 |
| MolMo 7B-D | 49.0 | 45.0 | 20.0 | 11.0 | 23.0 | 56.0 | 40.0 | 12.0 | 31.0 | 48.0 | 34.3 |
| MolMo 72B | 44.0 | 47.0 | 22.0 | 25.0 | 33.0 | 50.0 | 48.0 | 30.0 | 46.0 | 52.0 | 39.8 |
| Qwen2-VL-2B | 43.0 | 44.0 | 15.0 | 19.0 | 26.0 | 47.0 | 38.0 | 12.0 | 27.0 | 45.0 | 32.3 |
| Qwen2-VL-7B | 50.0 | 50.0 | 23.0 | 19.0 | 34.0 | 46.0 | 46.0 | 16.0 | 45.0 | 52.0 | 38.9 |
| Qwen2-VL-72B | 44.0 | 52.0 | 27.0 | 27.0 | 37.0 | 61.0 | 56.0 | 36.0 | 53.0 | 53.0 | 44.4 |
| InternVL2-4B | 50.0 | 56.0 | 30.0 | 17.0 | 18.0 | 49.0 | 54.0 | 16.0 | 38.0 | 53.0 | 38.4 |
| InternVL2-8B | 44.0 | 36.0 | 29.0 | 30.0 | 27.0 | 56.0 | 50.0 | 22.0 | 52.0 | 56.0 | 40.7 |
| InternVL2-26B | 44.0 | 47.0 | 24.0 | 22.0 | 26.0 | 55.0 | 58.0 | 28.0 | 47.0 | 46.0 | 39.3 |
| InternVL2-40B | 43.0 | 45.0 | 32.0 | 23.0 | 31.0 | 57.0 | 28.0 | 30.0 | 61.0 | 58.0 | 42.1 |
| InternVL2-76B | 44.0 | 42.0 | 28.0 | 34.0 | 45.0 | 56.0 | 60.0 | 36.0 | 63.0 | 54.0 | 46.0 |
| Claude Sonnet 3.5 | 50.0 | 47.0 | 23.0 | 20.0 | 33.0 | 59.0 | 52.0 | 40.0 | 61.0 | 52.0 | 43.4 |
| Claude Sonnet 4 | 38.0 | 57.0 | 32.0 | 25.0 | 33.0 | 66.0 | 72.0 | 44.0 | 70.0 | 54.0 | 48.1 |
| Claude Opus 4 | 41.0 | 47.0 | 35.0 | 34.0 | 36.0 | 60.0 | 72.0 | 50.0 | 80.0 | 50.0 | 49.3 |
| GPT-4o-mini | 45.0 | **66.0** | 26.0 | 19.0 | 30.0 | 58.0 | 58.0 | 32.0 | 40.0 | 53.0 | 42.4 |
| GPT-4o | 58.0 | 48.0 | 27.0 | 34.0 | 38.0 | 69.0 | 72.0 | 50.0 | 46.0 | 58.0 | 48.8 |
| Gemini 1.5 Flash | 47.0 | 51.0 | 25.0 | 24.0 | 39.0 | 60.0 | 68.0 | 42.0 | 58.0 | 58.0 | 49.2 |
| Gemini 1.5 Pro | 47.0 | 53.0 | 33.0 | 40.0 | 53.0 | 70.0 | 62.0 | 52.0 | 67.0 | 53.0 | 52.6 |
| Gemini 2.5 Pro | **66.0** | 52.0 | **55.0** | **59.0** | **56.0** | **90.0** | **92.0** | **88.0** | **86.0** | **72.0** | **79.0** |
| Human | 96.7 | 90.0 | 93.3 | 93.3 | 86.7 | 100.0 | 93.3 | 93.0 | 93.3 | 95.0 | 93.5 |

(a) Accuracy on VisOnlyQA-Eval-Real.

| | Geometry | | | | | 3D | | Average |
|---|---|---|---|---|---|---|---|---|
| | Triangle | Quadrilateral | Length | Angle | Area | Size | Angle | |
| Random | 50.0 | 50.0 | 20.0 | 20.0 | 20.0 | 33.3 | 20.0 | 30.5 |
| Phi-3.5-vision | 54.0 | 55.0 | 15.0 | 22.0 | 21.0 | 39.0 | 20.0 | 32.3 |
| LLaVA-Next 8B | 50.0 | 50.0 | 17.0 | 21.0 | 19.0 | 26.0 | 19.0 | 28.9 |
| LLaVA-Next 34B | 51.0 | 50.0 | 25.0 | 24.0 | 20.0 | 48.0 | **32.0** | 35.7 |
| Llama 3.2 11B | 54.0 | 52.0 | 31.0 | 21.0 | 21.0 | 32.0 | 21.0 | 33.1 |
| Llama 3.2 90B | 61.0 | 56.0 | 12.0 | 16.0 | 20.0 | 45.0 | 26.0 | 33.7 |
| MolMo 7B-D | 49.0 | 56.0 | 22.0 | 20.0 | 14.0 | 29.0 | 27.0 | 31.0 |
| MolMo 72B | 51.0 | 55.0 | 23.0 | 22.0 | 18.0 | 50.0 | 27.0 | 35.1 |
| Qwen2-VL-2B | 50.0 | 50.0 | 31.0 | 23.0 | 20.0 | 38.0 | 23.0 | 33.6 |
| Qwen2-VL-7B | 58.0 | 59.0 | 24.0 | 18.0 | 22.0 | 58.0 | 21.0 | 37.1 |
| Qwen2-VL-72B | 51.0 | 56.0 | 33.0 | 21.0 | 26.0 | 76.0 | 27.0 | 41.4 |
| InternVL2-4B | 50.0 | 51.0 | 21.0 | 24.0 | 18.0 | 57.0 | 18.0 | 34.1 |
| InternVL2-8B | 51.0 | 57.0 | 21.0 | 17.0 | 23.0 | 46.0 | 30.0 | 35.0 |
| InternVL2-26B | 51.0 | 53.0 | 30.0 | 23.0 | 21.0 | 72.0 | 25.0 | 39.3 |
| InternVL2-40B | 51.0 | 54.0 | 30.0 | 23.0 | 21.0 | 69.0 | 25.0 | 39.0 |
| InternVL2-76B | 52.0 | 51.0 | 29.0 | 18.0 | 22.0 | **84.0** | 27.0 | 40.4 |
| Claude Sonnet 3.5 | 61.0 | 63.0 | 33.0 | 20.0 | 34.0 | 62.0 | 22.0 | 42.1 |
| Claude Sonnet 4 | 57.0 | 59.0 | 28.0 | 32.0 | 30.0 | 79.0 | 28.0 | 44.7 |
| Claude Opus 4 | 50.0 | 53.0 | 36.0 | 21.0 | 24.0 | **84.0** | 26.0 | 42.0 |
| GPT-4o-mini | 60.0 | 51.0 | 21.0 | 20.0 | 18.0 | 27.0 | 23.0 | 31.4 |
| GPT-4o | 66.0 | 56.0 | 25.0 | 17.0 | 26.0 | 60.0 | 23.0 | 39.0 |
| Gemini 1.5 Flash | 54.0 | 51.0 | 29.0 | 21.0 | 19.0 | 60.0 | 21.0 | 36.4 |
| Gemini 1.5 Pro | 54.0 | 57.0 | 34.0 | 21.0 | 40.0 | 69.0 | 22.0 | 42.4 |
| Gemini 2.5 Pro | **68.0** | **67.0** | **61.0** | **47.0** | **48.0** | 74.0 | 23.0 | **55.4** |
| Human | 95.0 | 95.0 | 95.0 | 90.0 | 95.0 | 100.0 | 95.0 | 95.0 |

(b) Accuracy on VisOnlyQA-Eval-Synthetic.

Table 5: Accuracy of LVLMs on VisOnlyQA-Eval with no chain-of-thought reasoning. All LVLMs perform much worse than humans and are comparable to random performance in many tasks. Bold font indicates the best model performance in each column.

We evaluate InternVL2 76B and Gemini 1.5 Pro. First, Table 6 shows that these models consistently exhibit poor geometric perception on geometric shapes with different numbers of lines (i.e., complexity). As shown in Figure 4, even on simple geometric shapes that only include two or three lines, LVLMs cannot accurately perceive shape, length, and angle. Second, Table 7 shows that the angle between two lines does not largely influence the performance on the Length task, which compares the lengths of two lines, while we expected that this task would be more difficult when the angle is larger. These results suggest that the current LVLMs face fundamental challenges in geometric perception, regardless of the complexity of the geometric shapes or the difficulty of the tasks.

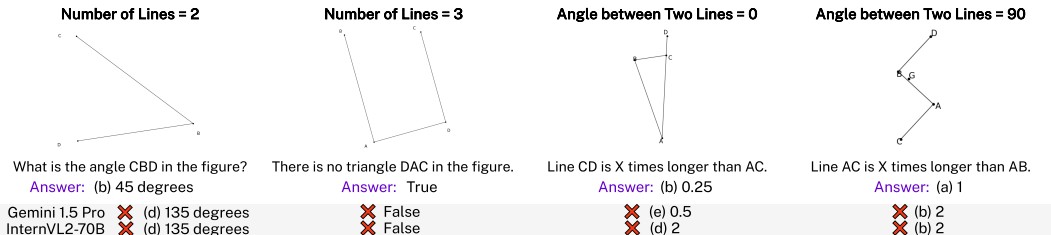

Figure 4: Example figures and model outputs for the analysis dataset. LVLMs exhibit poor geometric perception even on very simple geometric shapes.

| # Lines | 2 | 3 | 4 | 5 | 6 | 2 | 3 | 4 | 5 | 6 |
|---|---|---|---|---|---|---|---|---|---|---|
| Triangle | – | 50.0 | 52.0 | 50.0 | 50.0 | 56.0 | 54.0 | 62.0 | 48.0 | – |
| Length | 34.0 | 20.0 | 24.0 | 22.0 | 30.0 | 42.0 | 42.0 | 44.0 | 44.0 | 38.0 |
| Angle | 18.0 | 20.0 | 22.0 | 20.0 | 22.0 | 24.0 | 30.0 | 26.0 | 22.0 | 30.0 |

(a) InternVL2 76B  (b) Gemini 1.5 Pro

Table 6: Accuracy of LVLMs on simple geometric shapes.

| Angle | 0 | 45 | 90 |
|---|---|---|---|
| InternVL2 | 24.0 | 16.0 | 22.0 |
| Gemini | 36.0 | 38.0 | 36.0 |

Table 7: Accuracy on the Length task with different angles between two lines.

## 4.3 VisOnlyQA **Evaluates Geometric Perception Independent of Other Capabilities**

To verify our claim that VisOnlyQA evaluates the capability to perceive geometric information independent of other capabilities, this section demonstrates that our dataset does not involve reasoning or knowledge difficult for recent LVLMs. If recent LVLMs do not make mistakes in reasoning or knowledge on our dataset, we can conclude that the performance of LVLMs on this dataset evaluates the capability to perceive geometric information alone. In this section, we examine chain-of-thought reasoning of LVLMs for error analysis.

**Error analysis in chain-of-thought.** Chain-of-thought reasoning provides clues to analyzing why LVLMs make mistakes. We manually annotate errors in chain-of-thought reasoning by six models on VisOnlyQA-Eval-Real and provide the results in Figure 5. We manually annotate error categories for 250 responses (50 responses for each model). Following prior work (Yue et al., 2024; Zhang et al., 2024a), we classify their errors into the following categories. Refer to Appendix E for details.

- **Question Understanding Error**: LVLMs understand questions incorrectly.
- **Visual Perception Error**: LVLMs do not correctly perceive visual information. In our dataset, this category only involves errors in geometric perception.
- **Reasoning Error**: Reasoning on perceived information includes mistakes.
- **Minor Problems in Reasoning**: Reasoning is insufficient or redundant.

We observe that almost all errors are visual perception errors, as in Figure 5, verifying that our dataset evaluates the geometric perception of LVLMs independent of other capabilities. Specifically, almost all errors

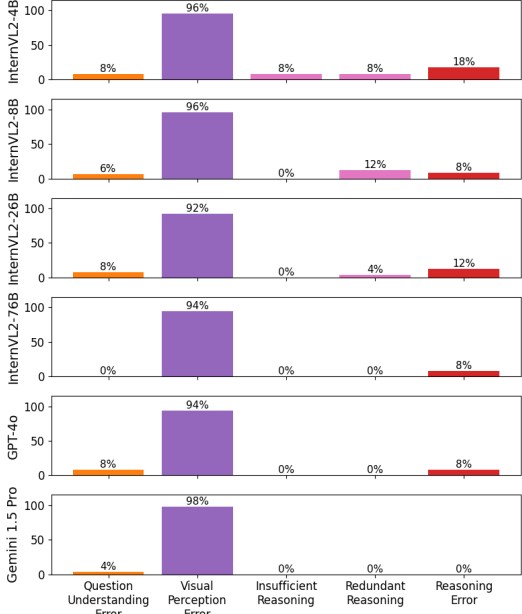

Figure 5: Error categories in chain-of-thought reasoning by LVLMs on VisOnlyQA-Eval-Real. Almost all errors are visual perception errors, verifying that our dataset evaluates the geometric perception of LVLMs independent of other capabilities. Each response can include multiple categories of errors.

made by Gemini 1.5 Pro do not involve anything other than visual perception errors, indicating that VisOnlyQA can evaluate geometric perception almost entirely independent from other capabilities for future models stronger than Gemini 1.5 Pro. However, at the same time, we need to be cautious when comparing the performances of LVLMs with weaker reasoning capabilities, as up to 10% of their mistakes on our dataset may not involve visual perception errors; if the performance difference between two weak models in our dataset is small, we cannot conclude either is better at geometric perception. Still, these errors in other capabilities do not affect our conclusion that existing LVLMs exhibit clear limitations in perceiving geometric information.

**Chain-of-thought does not consistently improve performance.** We also observe that chain-of-thought does not consistently improve the performance of LVLMs on VisOnlyQA (Appendix F.1). This result differs from observations on datasets for the visual *reasoning* tasks, where chain-of-thought largely improves performance (Wu et al., 2023; Chen et al., 2024a; Zhang et al., 2024a). This result is consistent with our claim that reasoning is not a bottleneck in our dataset for recent LVLMs and does not largely influence the final performance.

## 4.4 Additional Training Data Does Not Always Improve Geometric Perception

**Motivation and hypothesis.** We hypothesize that current LVLMs struggle to perceive geometric information due to a lack of training data requiring this capability, consistent with the motivation of prior work (Gao et al., 2025; Xing et al., 2025). To verify this hypothesis, we evaluate LVLMs fine-tuned on VisOnlyQA-Train.

**Settings.** We fine-tune InternVL2 (4B, 8B, 26B) (OpenGVLab Team, 2024), Qwen2-VL (2B, 7B) (Wang et al., 2024a), and Phi-3.5-Vision (Microsoft, 2024) on each task in VisOnlyQA-Train (7 tasks in total) and evaluate on Eval-Synthetic (in-distribution; figures from the same distribution as the Train data) and Eval-Real (out-of-distribution). To evaluate the maximum possible performance, we fine-tune each model in a single-task setting on 10k training data. In total, we fine-tune seven models independently for each LVLM. We use prompts without chain-of-thought. Refer to Appendix D for detailed settings.

**Improvement by fine-tuning depends on task properties.** As shown in Figure 6, LVLMs fine-tuned on VisOnlyQA-Train exhibit both positive and negative results in VisOnlyQA. **Positive results:** All models achieve near-perfect performance in 3D-Size after fine-tuning, and models larger than 7B show large improvement even on the out-of-distribution figures in Geometry-Length and Area. This result partially supports our hypothesis that training data for existing LVLMs are insufficient and indicates that our approach of using synthetic training data has the potential to improve the capability of LVLMs to perceive geometric information. **Negative results:** However, fine-tuned models are still often much worse than human performance, even on in-distribution figures. Specifically, fine-tuning almost does not improve performance in 3D-Angle, and we observe relatively small improvements

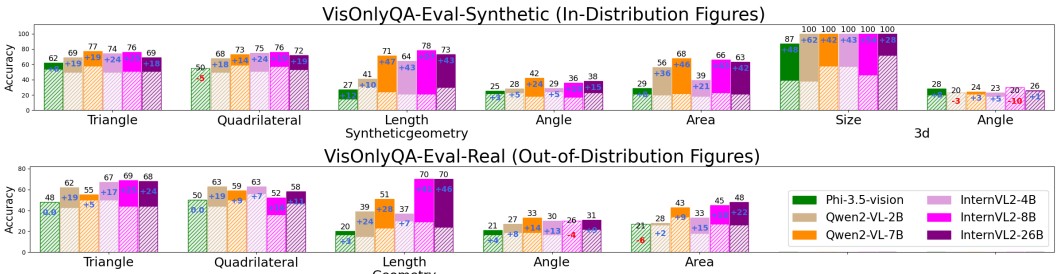

Figure 6: Accuracy after fine-tuning on VisOnlyQA-Train. We evaluate on VisOnlyQA-Eval-Synthetic, which is generated from the same distribution as the training data, and VisOnlyQA-Eval-Real, which includes images from different distributions. The numbers above the bars represent the accuracy after fine-tuning, and the ones inside the bars represent the improvements from the original models (white bars with hatches). Details are in Table 13.

on `Geometry-Triangle`, `Quadrilateral`, and `Angle`, even on in-distribution figures. This result indicates that fine-tuning on datasets that require accurate perception of geometric information is not always effective, depending on the properties of target tasks.

**Improvement by fine-tuning depends on model sizes.** Figure 6 shows that models larger than 7B tend to achieve greater performance gains after fine-tuning. Specifically, models larger than 7B exhibit much larger improvements than smaller models in `Geometry-Length` and `Area`. This result suggests that model sizes largely influence their capability to perceive geometric information, even when training data for target tasks is available. **Saturation:** However, we also observe that InternVL2-26B achieves almost the same performance as InternVL2-8B after fine-tuning. It suggests that simply fine-tuning larger models on our datasets will not achieve human performance.

**Was our hypothesis supported? — Partially.** Our results indicate that the insufficiency of training data is one of the reasons why the current LVLMs often cannot accurately perceive geometric information in images. However, depending on target tasks and models, additional training data does not always resolve the issue.

### 4.5 Larger Language Models Improve the Geometric Perception of LVLMs

InternVL2 4B and 8B, and Qwen2-VL 2B and 7B, respectively, use the same vision transformer (ViT) within each pair while differing in their language models. We expected the visual encoders to play a major role in geometric perception and models using the same ViT to perform similarly on VisOnlyQA, particularly after fine-tuning, since fine-tuning would help models understand tasks, further reducing the impact of the reasoning capability of language

| | ViT | LLM | Original | | Fine-tuned | |
|---|---|---|---|---|---|---|
| | | | Real | Synthetic | Real | Synthetic |
| InternVL2-4B | 304M | 3.8B | 38.4 | 34.1 | 46.0 | 57.7 |
| InternVL2-8B | 304M | 7.7B | 40.7 | 35.0 | 52.4* | 64.6* |
| Qwen2-VL-2B | 675M | 1.5B | 32.3 | 33.6 | 43.8 | 54.6 |
| Qwen2-VL-7B | 675M | 7.6B | 38.9* | 37.1* | 48.2* | 65.0* |

Table 8: Larger language models improve the performance of LVLMs on VisOnlyQA-Eval when using the same visual encoders. *: Larger model is better ($p < 0.05$, paired bootstrap (Koehn, 2004)).

models of LVLMs. However, as shown in Table 8, there are performance gaps between LVLMs using the same ViT and different language models, and the gaps become larger after fine-tuning. This observation indicates that language models of LVLMs affect the capability to perceive geometric information, and the influence of LLMs of LVLMs is not limited to reasoning or knowledge. This result suggests that language models play a crucial role in processing visual information encoded by ViT, and strong language models are needed even in geometric perception tasks that do not involve challenging reasoning or knowledge.

## 5 Conclusion

This work evaluates the capability of LVLMs to perceive geometric information in images, such as shape, angle, and size, and reveals that the current LVLMs still often cannot accurately perceive basic geometric information. We introduce VisOnlyQA, a new dataset designed for evaluating the geometric perception of LVLMs independent of other capabilities, such as reasoning. Our experiments on VisOnlyQA show a cautionary observation indicating that LVLMs still cannot accurately perceive basic visual information and may not be faithful to the input images in vision-language tasks. We also create a training set of VisOnlyQA to investigate approaches to improve the geometric perception of LVLMs. Our analysis of models fine-tuned on the training data suggests that simply scaling model size or training data does not fully resolve this issue in the perception of geometric information.

## Reproducibility Statement

In our GitHub repository, we provide our VisOnlyQA dataset, code for dataset creation and all experiments, and model responses.[1] The appendix includes details of model access, prompts, and hyperparameters.

## Acknowledgment

This work was supported by NSF CAREER Award IIS-2338418. We also thank OpenAI's Researcher Access Program for providing API credits. We appreciate VLMEvalKit for supporting our dataset.[2] We are grateful to Kai Katsumata for the valuable discussions and to Xueqing Wu for constructive feedback on our dataset. We appreciate valuable suggestions from anonymous reviewers, including those recommending experiments in Section 4.2.

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

# Table of Contents of Appendix

# A    Additional Related Work

**Large vision language models.**    Recent LVLMs often consist of vision transformers (ViT) (Dosovitskiy et al., 2021) and large language models (Ouyang et al., 2022; OpenAI, 2023), which are jointly trained on vision language tasks such as image captioning and visual question answering (Alayrac et al., 2022; Li et al., 2023b; Liu et al., 2023b; Ye et al., 2024). Powered by the multi-modal pre-training on transformers, various open source (Liu et al., 2023b; 2024; Chen et al., 2024c;b; Bai et al., 2023; Zhu et al., 2024; Xue et al., 2024; Microsoft, 2024; Deitke et al., 2025) and proprietary (OpenAI, 2024a; Anthropic, 2024; Google, 2024) LVLMs have been developed in recent years. Several studies also propose models for specific applications, such as mathematical reasoning (Zhang et al., 2025), chart understanding (Liu et al., 2023a; Masry et al., 2023), medical images (Li et al., 2023a), and text-rich image understanding (Zhang et al., 2024b).

**Synthetic images for training and evaluating visual perception.** In this work, we create synthetic geometric shapes for evaluating and training geometric perception. There is prior work that uses **synthetic geometric shapes** for evaluating or training geometric reasoning. GeomVerse (Kazemi et al., 2024) is a synthetic evaluation dataset generated from a predefined set of shapes and formulas. AutoGeo (Huang et al., 2025) is a large-scale training dataset created by a rule-based pipeline. G-LLaVA (Gao et al., 2025) uses a dataset generated from text-only LLMs to improve performance in geometric problems. There also exist datasets that use synthetic figures in **other domains** to evaluate the visual perception of LVLMs on tasks including visual question answering (Antol et al., 2015; Zhang et al., 2016; Kuhnle & Copestake, 2017; Lu et al., 2021b), chart understanding (Kahou et al., 2018; Kafle et al., 2018), visual reasoning (Suhr et al., 2017), mathematical reasoning (Lu et al., 2021a), diagram understanding (Giledereli et al., 2024), 3D object understanding (Johnson et al., 2017; Koch et al., 2019; Li et al., 2023c), and color distinction (Hyeon-Woo et al., 2025).

# B  Model Access

This section provides details of the model access and model parameters we use in Section 4.1. For all models, we use a temperature of zero or do_sample=False. The model responses in this paper were collected between October 1, 2024, and March 9, 2025.

## B.1  Proprietary Models

**OpenAI GPT.** We access GPT-4o (OpenAI, 2023; 2024a;b) models via OpenAI API.[3] We evaluate gpt-4o-mini-2024-07-18 and gpt-4o-2024-08-06 with the parameter of detail: high, which make the model to receive high resolution images.[4]

**Anthropic Claude.** We access Claude 3.5 Anthropic (2024) and Claude 4 (Anthropic, 2025) via Anthropic API.[5] We evaluate claude-3-5-sonnet-20240620, claude-sonnet-4-20250514, and claude-opus-4-20250514.

**Google Gemini.** We access Gemini 1.5 (Google, 2024) and Gemini 2.5 (Google, 2025) via Google Cloud.[6] We evaluate gemini-1.5-flash-002, gemini-1.5-pro-002, and gemini-2.5-pro-preview-05-06.

## B.2  Open Models

We evaluate models published on Hugging Face Model Hub.[7] For InternVL2 (OpenGVLab Team, 2024), Qwen2-VL (Wang et al., 2024a), and Phi-3.5-vision (Microsoft, 2024), we evaluate the models using code released by the authors.[8] For other models, we evaluate using VLMEvalKit (Duan et al., 2024).[9] Refer to Table 9 for the models we evaluate.

For Qwen2-VL, we set max_pixels=1280*28*28.[10]

---

[3]https://platform.openai.com/
[4]https://platform.openai.com/docs/guides/vision/low-or-high-fidelity-image-understanding
[5]https://console.anthropic.com/
[6]https://cloud.google.com/
[7]https://huggingface.co/models
[8]InternVL2: https://github.com/OpenGVLab/InternVL, Qwen2-VL: https://github.com/QwenLM/Qwen2-VL, Phi-3.5-vision: https://github.com/microsoft/Phi-3CookBook
[9]https://github.com/open-compass/VLMEvalKit
[10]https://huggingface.co/Qwen/Qwen2-VL-72B-Instruct#image-resolution-for-performance-boost

| | |
|---|---|
| Phi-3.5-vision | `microsoft/Phi-3.5-vision-instruct` |
| LLaVA-Next 8B | `llava_next_llama3` |
| LLaVA-Next 34B | `llava_next_yi_34b` |
| MolMo 7B-D | `molmo-7B-D-0924` |
| MolMo 72B | `molmo-72B-0924` |
| Llama 3.2 11B | `Llama-3.2-11B-Vision-Instruct` |
| Llama 3.2 90B | `Llama-3.2-90B-Vision-Instruct` |
| Qwen2-VL-2B | `Qwen/Qwen2-VL-2B-Instruct` |
| Qwen2-VL-7B | `Qwen/Qwen2-VL-7B-Instruct` |
| Qwen2-VL-72B | `Qwen/Qwen2-VL-72B-Instruct` |
| InternVL2-4B | `OpenGVLab/InternVL2-4B` |
| InternVL2-8B | `OpenGVLab/InternVL2-8B` |
| InternVL2-26B | `OpenGVLab/InternVL2-26B` |
| InternVL2-40B | `OpenGVLab/InternVL2-40B` |
| InternVL2-76B | `OpenGVLab/InternVL2-Llama3-76B` |
| Claude Sonnet 3.5 | `claude-3-5-sonnet-20240620` |
| Claude Sonnet 4 | `claude-sonnet-4-20250514` |
| Claude Opus 4 | `claude-opus-4-20250514` |
| GPT-4o-mini | `gpt-4o-mini-2024-07-18` |
| GPT-4o | `gpt-4o-2024-08-06` |
| Gemini 1.5 Flash | `gemini-1.5-flash-002` |
| Gemini 1.5 Pro | `gemini-1.5-pro-002` |
| Gemini 2.5 Pro | `gemini-2.5-pro-preview-05-06` |

Table 9: LVLMs we evaluate in this paper. For open models, this table shows model names in Hugging Face or VLMEvalKit.

# C   Details of LVLM Evaluation

This section provides details of experiments in Section 4.1 and 4.3.

**Prompts.** Table 10 shows two types of prompts with and without chain-of-thought we use to evaluate LVLMs on VisOnlyQA in Section 4.

| Prompt Type | Prompt |
|---|---|
| w/o chain-of-thought | {question}

Your response should only include the final answer ({response_type}). Do not include any reasoning or explanation in your response. |
| w/ chain-of-thought | {question}

In your response, provide a short explanation or reasoning for your answer. Then, provide the final answer ({response_type}). |

Table 10: Prompts we use when evaluating LVLMs on VisOnlyQA. {response_type} specifies the format of final answers, such as (a, b, c, d) or (True, False)

**Postprocessing.** We extract the selected options from responses from LVLMs using GPT-4o. We instruct GPT-4o with the following prompt, where {response_type} is final answers for each task, such as "a, b, c, d, e" or "True, False".

> Your task is to extract the final answer (selected option) from the response. Your response should only include {response_type}.
> Question: {question}
> Response: {response}

We use the following prompt for Chemistry-Shape(m).

> Your task is to extract the final answer from the response. Your response should only include the final answer(s) in a format of "a", "a,b", "a,c,d", "a,b,c,d". For example, "(a), (b), (c), (d)" should be converted to "a,b,c,d".
> Question: {question}
> Response: {response}

## D   Details of Fine-tuning

We fine-tune InternVL2 (4B, 8B, and 26B) (OpenGVLab Team, 2024), Qwen2-VL (2B and 7B) (Wang et al., 2024a), and Phi-3.5-vision (Microsoft, 2024). We use the following parameters for our fine-tuning. For other parameters, we use fine-tuning code and hyperparameters provided by the authors, and we fine-tune each model for three epochs.[11]

- InternVL2
    - We use a learning rate of 4e-6 for the 4B and 8B models and 2e-6 for the 26B models, which are ten times smaller than the original settings. We find the original learning rate is too large for our dataset and does not decrease training loss effectively in some tasks.
- Qwen2-VL
    - We use the total batch size of 128.
    - We use `warmup_steps` of 100 instead of `warmup_ratio` of 0.1 in the original setting because our dataset is relatively small.

---

[11]InternVL2: `https://github.com/OpenGVLab/InternVL/tree/main/internvl_chat/shell/internvl2.0/2nd_finetune/`, Qwen2-VL: `https://github.com/hiyouga/LLaMA-Factory/blob/main/examples/train_full/qwen2vl_full_sft.yaml`, Phi-3.5-vision: `https://github.com/microsoft/Phi-3CookBook/blob/main/code/04.Finetuning/vision_finetuning/`

# E   Details of Chain-of-Thought Error Analysis

Table 11 shows an example for each error label in the analysis of chain-of-thought reasoning in Section 4.3.

| Error Label | Image | Model | Response |
|---|---|---|---|
| **Question Understanding Error** |  | GPT-4o | In the figure, CAEB is not a quadrilateral. A quadrilateral must have four distinct sides and four vertices. However, CAEB shares the side AE with triangle AEC and triangle AEB, making it not a closed four-sided figure. Final answer: False. |
| **Visual Perception Error** |  | Gemini 1.5 Pro | The statement is True. While points A, D, and P exist in the figure, they are not connected by line segments to form a closed shape. Therefore, triangle ADP does not exist in the figure. Final Answer: The final answer is $\boxed{\text{True}}$ |
| **Reasoning Error** |  | InternVL2-76B | To estimate the ratio of the areas of triangles QNM and QML, we can visually compare their bases and heights. From the diagram, it appears that the base of triangle QNM (which is NM) is roughly twice the length of the base of triangle QML (which is ML). Assuming the heights of the triangles are approximately the same, the ratio of their areas would be the square of the ratio of their corresponding sides. Since the ratio of the bases is approximately 2:1, the ratio of the areas would be (2:1)², which is 4:1. Therefore, triangle QNM is approximately 4 times larger in area than triangle QML. Final answer: (c) 4 |
| **Insufficient Reasoning** |  | Gemini 1.5 Pro | The figure shows a triangle with three edges and three vertices, which are explicitly connected. True |
| **Redundant Reasoning** |  | InternVL2-26B | The double line between the two hexagonal structures in the left chemical structure represents a double bond. This is a common way to depict a double bond in chemical structures, where two lines represent a single bond and one line represents a double bond. Final answer: True. |

Table 11: Examples for each error label in chain-of-thought error analysis.

# F    Additional Results

This section provides additional results of experiments in Section 4.

## F.1    Improvements by Chain-of-Thought

Table 12 shows improvement by chain-of-thought reasoning on VisOnlyQA-Eval-Real. We do not observe a consistent improvement from the setting that does not use chain-of-thought reasoning and only generates the final answer.

| | Geometry | | | | | | Chemistry | | Charts | | Average |
|---|---|---|---|---|---|---|---|---|---|---|---|
| | Triangle | Quadri-lateral | Diameter | Length | Angle | Area | Shape (s) | Shape (m) | Extraction | Inter-section | |
| Phi-3.5-vision | -2.0 | -1.0 | 8.0 | 4.0 | -3.0 | 18.0 | -2.0 | 2.0 | 6.0 | 1.0 | 3.4 |
| LLaVA-Next 8B | 4.0 | 2.0 | 9.0 | 1.0 | -7.0 | -1.0 | -2.0 | 14.0 | 2.0 | 0.0 | 1.8 |
| LLaVA-Next 34B | 3.0 | 1.0 | -7.0 | 5.0 | -3.0 | 0.0 | 4.0 | 2.0 | -1.0 | 3.0 | 0.4 |
| Llama 3.2 11B | 0.0 | -6.0 | 3.0 | 6.0 | -6.0 | 3.0 | 8.0 | 8.0 | 9.0 | -3.0 | 1.6 |
| Llama 3.2 90B | -11.0 | 2.0 | 8.0 | 2.0 | 5.0 | 6.0 | 2.0 | 8.0 | 15.0 | 5.0 | 4.1 |
| MolMo 7B-D | 0.0 | 1.0 | 3.0 | 8.0 | 3.0 | -3.0 | -6.0 | 10.0 | -3.0 | 5.0 | 1.8 |
| MolMo 72B | -1.0 | -1.0 | 2.0 | 2.0 | 11.0 | 1.0 | -10.0 | 4.0 | -11.0 | 0.0 | 0.0 |
| Qwen2-VL-2B | 0.0 | 0.0 | -2.0 | 0.0 | 4.0 | 0.0 | -6.0 | -4.0 | -4.0 | 3.0 | -0.4 |
| Qwen2-VL-7B | -1.0 | 2.0 | 3.0 | 2.0 | -5.0 | 2.0 | -4.0 | -4.0 | 3.0 | 1.0 | 0.3 |
| Qwen2-VL-72B | 3.0 | -2.0 | 5.0 | -2.0 | 5.0 | 1.0 | 20.0 | 2.0 | -5.0 | 3.0 | 2.1 |
| InternVL2-4B | -2.0 | -13.0 | -8.0 | 6.0 | 5.0 | 0.0 | 6.0 | -2.0 | -1.0 | 4.0 | -0.8 |
| InternVL2-8B | 0.0 | 7.0 | -3.0 | -10.0 | 2.0 | 6.0 | 2.0 | -2.0 | 2.0 | 0.0 | 0.4 |
| InternVL2-26B | -2.0 | 3.0 | 3.0 | 1.0 | 10.0 | -3.0 | 2.0 | 2.0 | 10.0 | 3.0 | 3.0 |
| InternVL2-40B | 7.0 | -1.0 | 1.0 | 1.0 | 11.0 | 3.0 | 18.0 | 8.0 | -6.0 | -3.0 | 2.9 |
| InternVL2-76B | 3.0 | 2.0 | -1.0 | -7.0 | 1.0 | 0.0 | -6.0 | -2.0 | -5.0 | -2.0 | -1.4 |
| Claude Sonnet 3.5 | 4.0 | 3.0 | 5.0 | 4.0 | -2.0 | -5.0 | 32.0 | 2.0 | 20.0 | 10.0 | 6.2 |
| GPT-4o-mini | 3.0 | -2.0 | 3.0 | -2.0 | 3.0 | 4.0 | 14.0 | 6.0 | -9.0 | 0.0 | 1.1 |
| GPT-4o | -3.0 | 1.0 | 3.0 | -8.0 | 4.0 | 1.0 | 10.0 | 4.0 | 5.0 | -1.0 | 1.0 |
| Gemini 1.5 Flash | 3.0 | -5.0 | 1.0 | -1.0 | -1.0 | 7.0 | 4.0 | 0.0 | 7.0 | 6.0 | -0.8 |
| Gemini 1.5 Pro | 0.0 | 8.0 | 2.0 | -6.0 | -5.0 | 2.0 | 10.0 | 4.0 | 5.0 | 3.0 | 1.8 |

Table 12: Improvement by Chain-of-Thought Reasoning.

## F.2 Improvements by Fine-tuning

Table 13 shows the performance of LVLMs fine-tuned on VisOnlyQA-Train, which corresponds to Figure 6.

| | | | Geometry | | | | | 3D | | Average |
|---|---|---|---|---|---|---|---|---|---|---|
| | | | Triangle | Quadri-lateral | Length | Angle | Area | Size | Angle | |
| | Random | | 50.0 | 50.0 | 20.0 | 20.0 | 20.0 | 33.3 | 20.0 | |
| VisOnlyQA-Eval-Synthetic (In-Distribution) | Phi-3.5-vision | Original | 54.0 | 55.0 | 15.0 | 22.0 | 21.0 | 39.0 | 20.0 | 32.3 |
| | | Fine-tuned | 62.0 | 50.0 | 27.0 | 25.0 | 29.0 | 87.0 | 28.0 | 44.0 |
| | Qwen2-VL-2B | Original | 50.0 | 50.0 | 31.0 | 23.0 | 20.0 | 38.0 | 23.0 | 33.6 |
| | | Fine-tuned | 69.0 | 68.0 | 41.0 | 28.0 | 56.0 | 100.0 | 20.0 | 54.6 |
| | Qwen2-VL-7B | Original | 58.0 | 59.0 | 24.0 | 18.0 | 22.0 | 58.0 | 21.0 | 37.1 |
| | | Fine-tuned | 77.0 | 73.0 | 71.0 | 42.0 | 68.0 | 100.0 | 24.0 | 65.0 |
| | InternVL2-4B | Original | 50.0 | 51.0 | 21.0 | 24.0 | 18.0 | 57.0 | 18.0 | 34.1 |
| | | Fine-tuned | 74.0 | 75.0 | 64.0 | 29.0 | 39.0 | 100.0 | 23.0 | 57.7 |
| | InternVL2-8B | Original | 51.0 | 57.0 | 21.0 | 17.0 | 23.0 | 46.0 | 30.0 | 35.0 |
| | | Fine-tuned | 76.0 | 76.0 | 78.0 | 36.0 | 66.0 | 100.0 | 20.0 | 64.6 |
| | InternVL2-26B | Original | 51.0 | 53.0 | 30.0 | 23.0 | 21.0 | 72.0 | 25.0 | 39.3 |
| | | Fine-tuned | 69.0 | 72.0 | 73.0 | 38.0 | 63.0 | 100.0 | 26.0 | 63.0 |
| VisOnlyQA-Eval-Real (Out-of-Distribution) | Phi-3.5-vision | Original | 48.0 | 50.0 | 17.0 | 17.0 | 27.0 | – | – | 31.8 |
| | | Fine-tuned | 48.0 | 50.0 | 20.0 | 21.0 | 21.0 | – | – | 32.0 |
| | Qwen2-VL-2B | Original | 43.0 | 44.0 | 15.0 | 19.0 | 26.0 | – | – | 29.4 |
| | | Fine-tuned | 62.0 | 63.0 | 39.0 | 27.0 | 28.0 | – | – | 43.8 |
| | Qwen2-VL-7B | Original | 50.0 | 50.0 | 23.0 | 19.0 | 34.0 | – | – | 35.2 |
| | | Fine-tuned | 55.0 | 59.0 | 51.0 | 33.0 | 43.0 | – | – | 48.2 |
| | InternVL2-4B | Original | 50.0 | 56.0 | 30.0 | 17.0 | 18.0 | – | – | 34.2 |
| | | Fine-tuned | 67.0 | 63.0 | 37.0 | 30.0 | 33.0 | – | – | 46.0 |
| | InternVL2-8B | Original | 44.0 | 36.0 | 29.0 | 30.0 | 27.0 | – | – | 33.2 |
| | | Fine-tuned | 69.0 | 52.0 | 70.0 | 26.0 | 45.0 | – | – | 52.4 |
| | InternVL2-26B | Original | 44.0 | 47.0 | 24.0 | 22.0 | 26.0 | – | – | 32.6 |
| | | Fine-tuned | 68.0 | 58.0 | 70.0 | 31.0 | 48.0 | – | – | 55.0 |

Table 13: Accuracy of LVLMs fine-tuned on VisOnlyQA-Train. We evaluate the fine-tuned models on VisOnlyQA-Eval-Synthetic, which is generated from the same distribution as the fine-tuning data, and VisOnlyQA-Eval-Real, which includes images from different distributions. This table corresponds to Figure 6.

## F.3 Fine-tuning of Different Components of LVLMs

Experiments in Section 4.4 fine-tune LLMs of LVLMs, which is the default setting of the fine-tuning code by the authors of InternVL. In Table 14, we show the performance of fine-tuning different components of InternVL2-8B. These findings are consistent with the analysis in Section 4.5, reinforcing the conclusion that the LLM component plays a central role in enabling geometric perception in LVLMs. In addition, fine-tuning the LLM component alone yields comparable performance to fine-tuning all model parameters.

| Fine-tuning | Length (Synthetic) | Length (Real) | Area (Synthetic) | Area (Real) |
|---|---|---|---|---|
| w/o Fine-tuning | 21.0 | 29.0 | 23.0 | 27.0 |
| ViT Only | 62.0 | 47.0 | 46.0 | 29.0 |
| LLM Only | 78.0 | 70.0 | 66.0 | 45.0 |
| All Parameters | 83.0 | 71.0 | 68.0 | 42.0 |

Table 14: Accuracy of InternVL2-8B with different fine-tuning settings.

## G  Computational Resources

To evaluate and fine-tune open models, we use a server with eight NVIDIA A100 SXM4 80GB GPUs. Almost all experiments can be done on four A100 GPUs, but the training of InternVL2-26B requires eight A100 GPUs in the configurations of the training code provided by the InterVL authors. Refer to Appendix D for detailed settings of fine-tuning.

## H   Example Data and Model Outputs

Table 15 to 65 show examples from VisOnlyQA and model responses. The examples in these tables are uniformly randomly selected from Eval-Real and Eval-Synthetic.

All instances in our dataset and all responses from LVLMs are provided in our GitHub repository.

| Image | Question | Gold Answer |
|---|---|---|
| 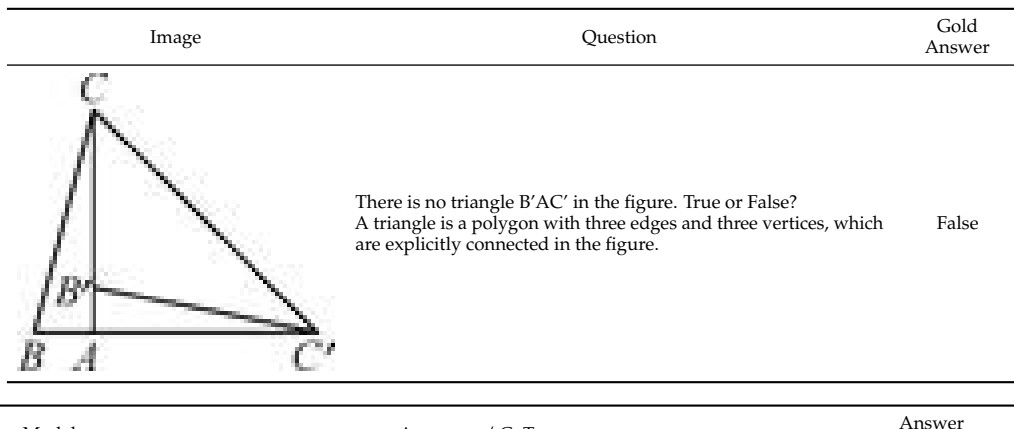 | There is no triangle B'AC' in the figure. True or False? A triangle is a polygon with three edges and three vertices, which are explicitly connected in the figure. | False |

| Model | Answer w/ CoT | | Answer w/o CoT | |
|---|---|---|---|---|
| InternVL2-4B | The figure shows a triangle with vertices labeled A, B, and C. The triangle B'AC' is not present in the figure. False | ✓ | False | ✓ |
| InternVL2-8B | False. In the given figure, there is a triangle B'AC' formed by the points B', A, and C'. The vertices B', A, and C' are explicitly connected by edges, forming a closed figure. Therefore, a triangle B'AC' does exist in the figure. | ✓ | False | ✓ |
| InternVL2-26B | To determine if there is a triangle B'AC' in the figure, we need to check if there are three vertices connected by edges. In the given figure, we can see that there are three vertices: B', A, and C'. These vertices are connected by edges, forming a triangle. Therefore, the statement "There is no triangle B'AC' in the figure" is false. Final answer: False. | ✓ | False | ✓ |
| InternVL2-76B | The figure shows a triangle labeled as B'AC'. This triangle has three edges (B'A, B'C', and A'C') and three vertices (B', A, and C'), which are all connected. Therefore, the statement "There is no triangle B'AC' in the figure" is False. | ✓ | False | ✓ |
| Claude 3.5 Sonnet | Let's analyze the figure: 1. The image shows a triangle with vertices labeled A, B, and C. 2. There's a line segment from B to a point on AC, labeled B'. 3. This creates two triangles within the original triangle: ABB' and BB'C. 4. However, there is no triangle explicitly labeled as B'AC' in this figure. While we can see a triangle formed by points B', A, and C, it's not specifically labeled or highlighted as a distinct triangle in the given diagram. Reasoning: The question asks about the existence of a triangle B'AC' in the figure. Although the points B', A, and C do form a triangle, it's not explicitly drawn or labeled as a separate entity in the diagram. Final answer: True (There is no triangle explicitly labeled or identified as B'AC' in the figure) | ✗ | False | ✓ |
| GPT-4o | In the figure, triangle \( B'AC' \) is formed by the vertices \( B' \), \( A \), and \( C' \). These points are explicitly connected by edges, forming a closed shape with three sides. Therefore, triangle \( B'AC' \) does exist in the figure. Final answer: False. | ✓ | True | ✗ |
| Gemini 1.5 Pro | The statement is **True**. There is a triangle BAC with vertices B, A, and C. While points B', C' are marked on the segments BA and BC respectively, there are no line segments drawn connecting A to either B' or C'. Therefore, no triangle B'AC' exists. Final Answer: The final answer is $\boxed{True}$ | ✗ | False | ✓ |

Table 15: Examples of dataset and model responses for Geometry-Triangle (1)

| Image | Question | Gold Answer |
|---|---|---|
| 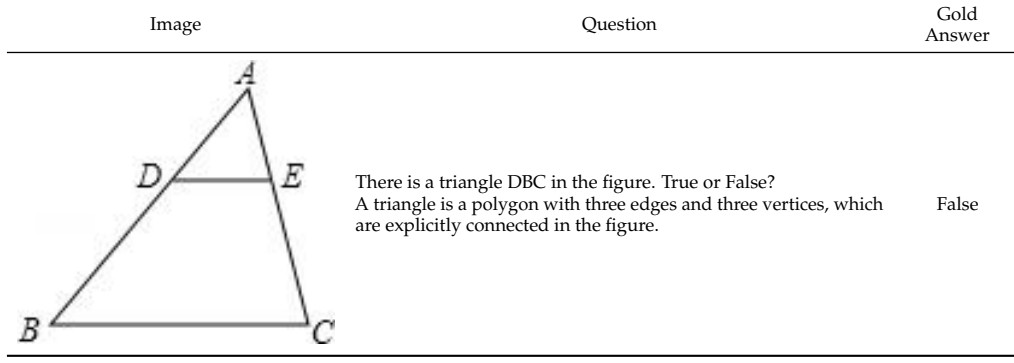 | There is a triangle DBC in the figure. True or False? A triangle is a polygon with three edges and three vertices, which are explicitly connected in the figure. | False |

| Model | Answer w/ CoT | | Answer w/o CoT | |
|---|---|---|---|---|
| InternVL2-4B | The figure shows a triangle with three edges and three vertices, which are explicitly connected. True | ✗ | True | ✗ |
| InternVL2-8B | False. The figure provided does not show a triangle DBC. Instead, it shows a triangle ABC with a line segment DE intersecting it. The vertices of the triangle are A, B, and C, and the line segment DE is drawn inside the triangle, creating two smaller triangles, ADE and DEC. There is no triangle DBC present in the figure. | ✓ | False | ✓ |
| InternVL2-26B | A triangle is a polygon with three edges and three vertices. In the given figure, triangle DBC has three edges (DB, BC, and CD) and three vertices (D, B, and C), which are explicitly connected. Therefore, the statement "There is a triangle DBC in the figure" is true. True. | ✗ | True | ✗ |
| InternVL2-76B | The figure shows a triangle labeled as DBC, with vertices D, B, and C connected by edges. Therefore, it is a valid triangle. True | ✗ | True | ✗ |
| Claude 3.5 Sonnet | Let's analyze the figure: 1. The image shows a shape with three distinct vertices labeled D, B, and C. 2. These three points are connected by straight line segments. 3. The shape formed by connecting these three points creates a closed figure with three sides. This configuration meets the definition of a triangle: a polygon with three edges and three vertices. Reasoning: The figure clearly depicts a three-sided closed shape with vertices labeled D, B, and C, which constitutes a triangle. Final answer: True | ✗ | True | ✗ |
| GPT-4o | In the figure, there is a triangle labeled \( \triangle ABC \). The line segment \( DE \) is parallel to \( BC \) and divides \( \triangle ABC \) into two smaller triangles, \( \triangle ADE \) and \( \triangle BDE \). However, there is no triangle labeled \( \triangle DBC \) in the figure. Final answer: False. | ✓ | False. | ✓ |
| Gemini 1.5 Pro | The figure shows points D, B, and C connected by line segments DB, BC, and CD. These three segments and the points D, B, and C form a triangle. Final Answer: The final answer is $\boxed{True}$ | ✗ | True | ✗ |

Table 16: Examples of dataset and model responses for Geometry-Triangle (2)

| Image | Question | Gold Answer |
|---|---|---|
|  | There is no triangle ACD in the figure. True or False? A triangle is a polygon with three edges and three vertices, which are explicitly connected in the figure. | False |

| Model | Answer w/ CoT | | Answer w/o CoT | |
|---|---|---|---|---|
| InternVL2-4B | The figure shows a quadrilateral ABCD with vertices A, B, C, and D. The lines connecting these vertices form a closed shape with four sides, which is a characteristic of a quadrilateral. However, the figure does not explicitly show a triangle ACD. A triangle is a polygon with three edges and three vertices. In the given figure, there is no triangle ACD explicitly drawn. Therefore, the answer is True. | ✗ | True | ✗ |
| InternVL2-8B | False In the given figure, a quadrilateral ABCD is shown with vertices A, B, C, and D. The lines AD, BC, and AC are drawn, forming triangles ACD, ABC, and ABD. Therefore, a triangle ACD does exist in the figure. | ✓ | False | ✓ |
| InternVL2-26B | To determine if triangle ACD exists in the figure, we need to check if the three vertices A, C, and D are connected by edges. 1. Vertex A is connected to vertex C by the edge AC. 2. Vertex C is connected to vertex D by the edge CD. 3. Vertex D is connected to vertex A by the edge DA. Since all three vertices are connected by edges, triangle ACD does exist in the figure. Therefore, the statement "There is no triangle ACD in the figure" is false. Final answer: False. | ✓ | False | ✓ |
| InternVL2-76B | The figure shows a quadrilateral ABCD with diagonals AC and BD intersecting at point O. The vertices A, C, and D are connected by edges, forming a triangle ACD. Therefore, the statement "There is no triangle ACD in the figure" is False. | ✓ | False | ✓ |
| Claude 3.5 Sonnet | Let's analyze the figure carefully: 1. We can see four distinct points: A, B, C, and D. 2. There's a point O in the center, which appears to be the intersection of the diagonals. 3. We can clearly see lines connecting A to D, A to C, and C to D. Given these observations: - The line AD forms one side of the potential triangle ACD. - The line AC forms another side of the potential triangle ACD. - The line CD forms the third side of the potential triangle ACD. These three lines connect the three points A, C, and D, forming a closed shape with three sides and three vertices. This meets the definition of a triangle. Therefore, there is indeed a triangle ACD in the figure. Final answer: False | ✓ | False | ✓ |
| GPT-4o | In the figure, points A, C, and D are connected by line segments, forming a closed shape with three edges and three vertices. This satisfies the definition of a triangle. Therefore, the statement "There is no triangle ACD in the figure" is False. | ✓ | False | ✓ |
| Gemini 1.5 Pro | The vertices A, C, and D are connected by line segments AC and AD, forming the triangle ACD. Final Answer: False | ✓ | False | ✓ |

Table 17: Examples of dataset and model responses for `Geometry-Triangle` (3)

| Image | Question | Gold Answer |
|---|---|---|
| 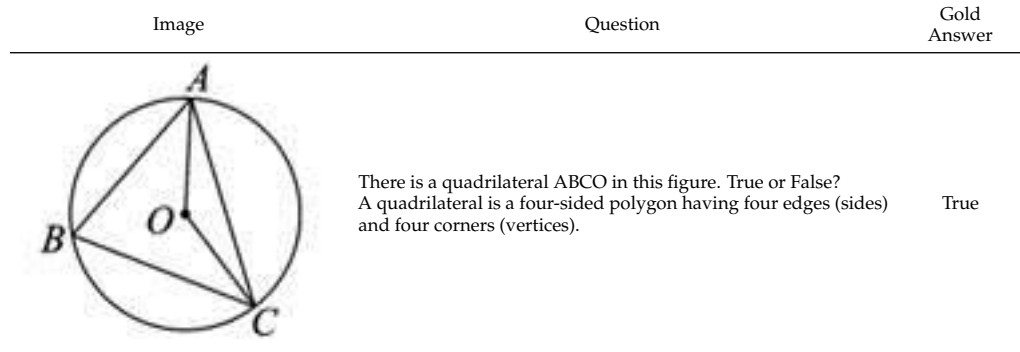 | There is a quadrilateral ABCO in this figure. True or False? A quadrilateral is a four-sided polygon having four edges (sides) and four corners (vertices). | True |

| Model | Answer w/ CoT | | Answer w/o CoT | |
|---|---|---|---|---|
| InternVL2-4B | The figure shows a circle with points A, B, C, and O on the circumference. The line segments AB, BC, CA, and AO form a quadrilateral. True | ✓ | False | ✗ |
| InternVL2-8B | False. Reasoning: The figure shows a quadrilateral ABCO, but it is not a standard quadrilateral. A standard quadrilateral has four sides and four vertices. However, in this figure, point O is the center of the circle, and the lines AB, BC, and CA are chords of the circle. The line segment AC is a diameter of the circle, and point O is the midpoint of AC. Therefore, the figure does not form a standard quadrilateral but rather a cyclic quadrilateral (a quadrilateral inscribed in a circle) with additional properties due to the circle's center. | ✗ | False | ✗ |
| InternVL2-26B | A quadrilateral is defined as a four-sided polygon. In the given figure, quadrilateral ABCO is formed by the four vertices A, B, C, and O, and the four edges connecting these vertices. Therefore, the statement "A quadrilateral is a four-sided polygon having four edges (sides) and four corners (vertices)" is true. True. | ✓ | False | ✗ |
| InternVL2-76B | The figure shows a circle with four points labeled A, B, C, and O. These points are connected by lines forming a four-sided polygon. Therefore, the figure represents a quadrilateral. Final answer: True | ✓ | False | ✗ |
| Claude 3.5 Sonnet | Let's analyze the figure carefully: 1. The image shows a circle with three distinct points labeled A, B, and C on its circumference. 2. There's also a point O, which appears to be the center of the circle. 3. Lines are drawn connecting points A, B, and C to each other and to point O. 4. While there are multiple triangles formed (such as ABC, ABO, BCO, and ACO), there isn't a four-sided shape visible. 5. A quadrilateral would require four distinct vertices connected by four sides, which we don't see in this figure. Explanation: The figure shows a circle with inscribed triangles, but it does not contain a four-sided polygon. The shape ABCO is not a quadrilateral because O is connected to all other points, forming triangles instead of a four-sided shape. Final answer: False | ✗ | False | ✗ |
| GPT-4o | In the figure, \(A\), \(B\), \(C\), and \(O\) are points. However, the lines form a triangle \(ABC\) and three additional line segments \(AO\), \(BO\), and \(CO\) that meet at point \(O\). This does not form a closed four-sided polygon, as there are not four distinct sides forming a quadrilateral. Final answer: False. | ✗ | False | ✗ |
| Gemini 1.5 Pro | The figure shows points A, B, C, and O connected by line segments to form a closed, four-sided shape. These four points represent the vertices, and the lines connecting them represent the edges. Thus, the figure represents a quadrilateral. Final Answer: The final answer is $\boxed{True}$ | ✓ | True | ✓ |

Table 18: Examples of dataset and model responses for Geometry-Quadrilateral (1)

| Image | Question | Gold Answer |
|---|---|---|
| 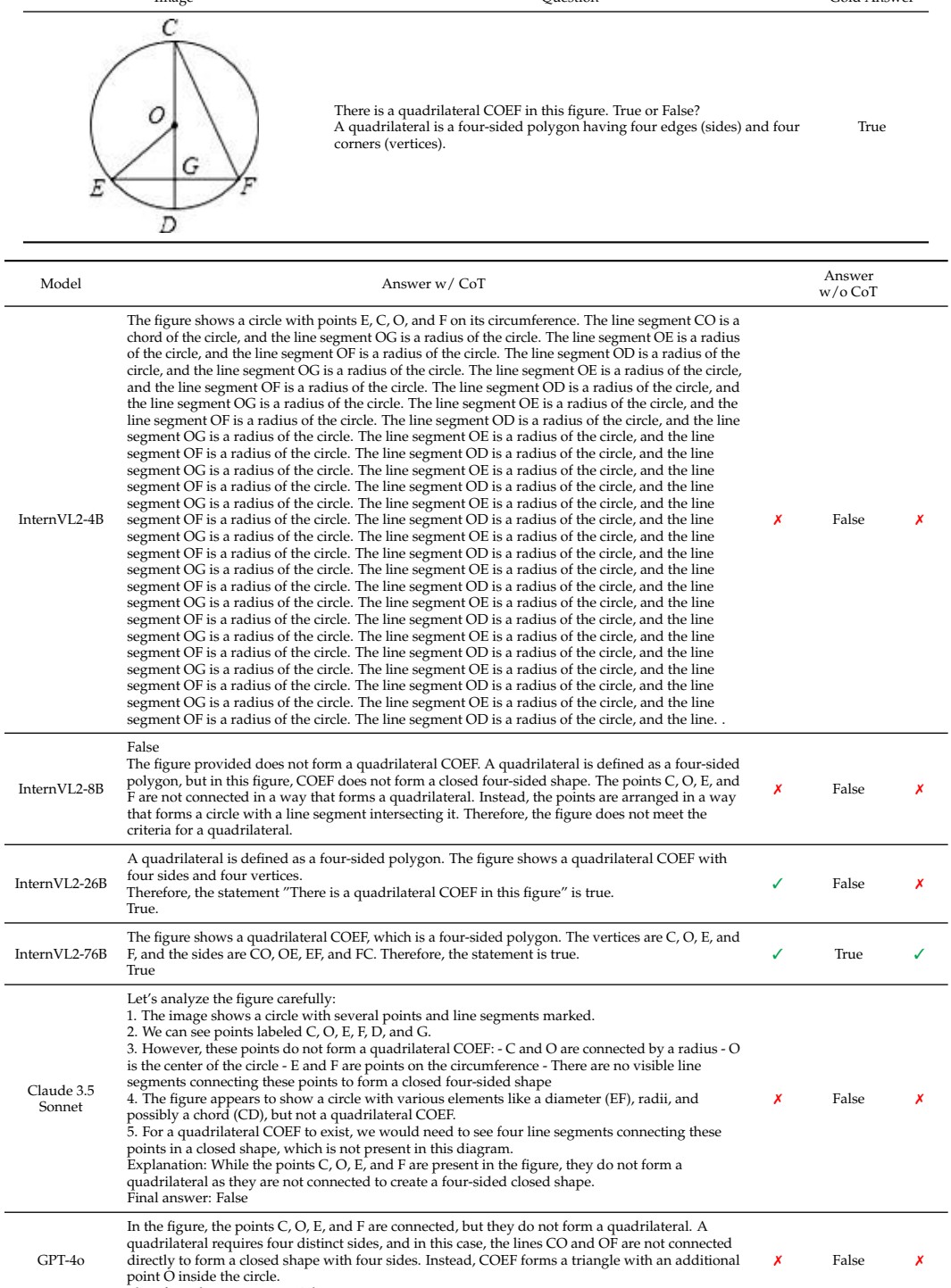 | There is a quadrilateral COEF in this figure. True or False? A quadrilateral is a four-sided polygon having four edges (sides) and four corners (vertices). | True |

| Model | Answer w/ CoT | | Answer w/o CoT | |
|---|---|---|---|---|
| InternVL2-4B | The figure shows a circle with points E, C, O, and F on its circumference. The line segment CO is a chord of the circle, and the line segment OG is a radius of the circle. The line segment OE is a radius of the circle, and the line segment OF is a radius of the circle. The line segment OD is a radius of the circle, and the line segment OG is a radius of the circle. The line segment OE is a radius of the circle, and the line segment OF is a radius of the circle. The line segment OD is a radius of the circle, and the line segment OG is a radius of the circle. The line segment OE is a radius of the circle, and the line segment OF is a radius of the circle. The line segment OD is a radius of the circle, and the line segment OG is a radius of the circle. The line segment OE is a radius of the circle, and the line segment OF is a radius of the circle. The line segment OD is a radius of the circle, and the line segment OG is a radius of the circle. The line segment OE is a radius of the circle, and the line segment OF is a radius of the circle. The line segment OD is a radius of the circle, and the line segment OG is a radius of the circle. The line segment OE is a radius of the circle, and the line segment OF is a radius of the circle. The line segment OD is a radius of the circle, and the line segment OG is a radius of the circle. The line segment OE is a radius of the circle, and the line segment OF is a radius of the circle. The line segment OD is a radius of the circle, and the line segment OG is a radius of the circle. The line segment OE is a radius of the circle, and the line segment OF is a radius of the circle. The line segment OD is a radius of the circle, and the line segment OG is a radius of the circle. The line segment OE is a radius of the circle, and the line segment OF is a radius of the circle. The line segment OD is a radius of the circle, and the line segment OG is a radius of the circle. The line segment OE is a radius of the circle, and the line segment OF is a radius of the circle. The line segment OD is a radius of the circle, and the line. . | ✗ | False | ✗ |
| InternVL2-8B | False The figure provided does not form a quadrilateral COEF. A quadrilateral is defined as a four-sided polygon, but in this figure, COEF does not form a closed four-sided shape. The points C, O, E, and F are not connected in a way that forms a quadrilateral. Instead, the points are arranged in a way that forms a circle with a line segment intersecting it. Therefore, the figure does not meet the criteria for a quadrilateral. | ✗ | False | ✗ |
| InternVL2-26B | A quadrilateral is defined as a four-sided polygon. The figure shows a quadrilateral COEF with four sides and four vertices. Therefore, the statement "There is a quadrilateral COEF in this figure" is true. True. | ✓ | False | ✗ |
| InternVL2-76B | The figure shows a quadrilateral COEF, which is a four-sided polygon. The vertices are C, O, E, and F, and the sides are CO, OE, EF, and FC. Therefore, the statement is true. True | ✓ | True | ✓ |
| Claude 3.5 Sonnet | Let's analyze the figure carefully: 1. The image shows a circle with several points and line segments marked. 2. We can see points labeled C, O, E, F, D, and G. 3. However, these points do not form a quadrilateral COEF: - C and O are connected by a radius - O is the center of the circle - E and F are points on the circumference - There are no visible line segments connecting these points to form a closed four-sided shape 4. The figure appears to show a circle with various elements like a diameter (EF), radii, and possibly a chord (CD), but not a quadrilateral COEF. 5. For a quadrilateral COEF to exist, we would need to see four line segments connecting these points in a closed shape, which is not present in this diagram. Explanation: While the points C, O, E, and F are present in the figure, they do not form a quadrilateral as they are not connected to create a four-sided closed shape. Final answer: False | ✗ | False | ✗ |
| GPT-4o | In the figure, the points C, O, E, and F are connected, but they do not form a quadrilateral. A quadrilateral requires four distinct sides, and in this case, the lines CO and OF are not connected directly to form a closed shape with four sides. Instead, COEF forms a triangle with an additional point O inside the circle. Therefore, the statement is False. | ✗ | False | ✗ |
| Gemini 1.5 Pro | The given figure has points C, O, E, and F. These four points can be connected to form a four-sided polygon. Therefore, COEF is a quadrilateral. Final Answer: The final answer is $\boxed{True}$ | ✓ | True | ✓ |

Table 19: Examples of dataset and model responses for `Geometry-Quadrilateral` (2)

| Image | Question | Gold Answer |
|---|---|---|
|  | There is no quadrilateral CBPQ in this figure. True or False? A quadrilateral is a four-sided polygon having four edges (sides) and four corners (vertices). | True |

| Model | Answer w/ CoT | | Answer w/o CoT | |
|---|---|---|---|---|
| InternVL2-4B | To determine if there is a quadrilateral CBPQ in the figure, we need to identify if there are four vertices (corners) and four edges (sides) that form a closed shape. In the given figure, we can see the following points: - Point A - Point B - Point C - Point P However, there is no point Q in the figure. Therefore, there is no quadrilateral CBPQ in this figure. Final answer: True | ✓ | True | ✓ |
| InternVL2-8B | False The figure does not contain a quadrilateral CBPQ. A quadrilateral is defined as a four-sided polygon, but in this figure, the points C, B, P, and Q do not form a closed four-sided shape. Instead, the points form a triangle (ABC) and a segment (PQ) that intersects the circle at point P. Therefore, there is no quadrilateral CBPQ in this figure. | ✗ | False | ✗ |
| InternVL2-26B | A quadrilateral is a four-sided polygon, and in the given figure, the shape CBPQ has only three sides and three vertices. Therefore, it cannot be classified as a quadrilateral. True. | ✓ | True | ✓ |
| InternVL2-76B | The figure shows a circle with points C, B, P, and Q. However, the quadrilateral CBPQ is not formed because the line segment BP does not exist in the figure. Therefore, the statement "There is no quadrilateral CBPQ in this figure" is true. True | ✓ | False | ✗ |
| Claude 3.5 Sonnet | Let's analyze the figure carefully: 1. We can see a circle with center C. 2. There's a triangle with vertices A, B, and what appears to be P. 3. Point Q is not explicitly labeled, but there is a point where the line from A intersects the circle. Now, let's consider if we can form a quadrilateral CBPQ: - C is the center of the circle - B is clearly marked - P appears to be marked at the top of the triangle - Q, while not labeled, could be considered as the point where line AP intersects the circle If we connect these four points (C, B, P, and the unlabeled intersection point as Q), we can indeed form a four-sided figure with four vertices. Therefore, it is possible to identify a quadrilateral CBPQ in this figure, even though Q is not explicitly labeled. Final answer: False | ✗ | False | ✗ |
| GPT-4o | In the figure, points C, B, P, and Q are not all connected to form a closed shape with four sides. A quadrilateral requires four connected edges forming a closed loop. Here, the points do not form such a shape. Therefore, the statement is **True**. | ✓ | True | ✓ |
| Gemini 1.5 Pro | The statement is **False**. The points C, B, P, and Q are connected by line segments CB, BP, PQ, and QC, forming a closed four-sided figure. This meets the definition of a quadrilateral. It doesn't matter that some of the lines forming it happen to also be part of other shapes (triangle and circle). Final Answer: The final answer is $\boxed{False}$ | ✗ | False | ✗ |

Table 20: Examples of dataset and model responses for Geometry-Quadrilateral (3)

| Image | Question | Gold Answer |
|---|---|---|
|  | Line AP is X times longer than CB. Which of the following options is a reasonable estimate of X? You only need to estimate from the visual information and do not need to do any mathematical reasoning. (a) 0.25 (b) 2 (c) 4 (d) 0.5 (e) 1 | c |

| Model | Answer w/ CoT | | Answer w/o CoT | |
|---|---|---|---|---|
| InternVL2-4B | To estimate the value of X, we need to visually compare the lengths of line AP and CB.
1. **Observation**: - Line AP is the hypotenuse of the right triangle AOB. - Line CB is the radius of the circle and is perpendicular to line AB at point B.
2. **Comparison**: - Since AB is a radius of the circle, it is equal to the radius of the circle, which is a fixed length. - AP is the hypotenuse of the right triangle AOB, which is longer than the radius AB.
3. **Estimation**: - Given that AB is a fixed length and AP is significantly longer, we can estimate that AP is approximately 2 times longer than AB.
4. **Conclusion**: - Based on the visual information, a reasonable estimate for the value of X (the ratio of AP to CB) is 2.
Therefore, the final answer is (b) 2. | ✗ | d | ✗ |
| InternVL2-8B | To estimate the value of $X$, we can visually compare the lengths of line segments $AP$ and $CB$ in the given diagram.
1. **Visual Comparison**: - $AP$ appears to be about 4 times the length of $CB$. This is because $AP$ extends beyond $CB$ by a significant proportion, making it appear much longer.
2. **Reasoning**: - Given the visual estimation, $AP$ is approximately 4 times the length of $CB$. This is a rough estimate based on the relative lengths observed in the diagram.
Therefore, the most reasonable estimate of $X$ is:
**Final Answer: (c) 4** | ✓ | b | ✗ |
| InternVL2-26B | To estimate the value of $X$, we need to compare the lengths of $AP$ and $CB$ visually.
1. Observe the lengths of $AP$ and $CB$ in the diagram. 2. Estimate the ratio of $AP$ to $CB$ by visually comparing their lengths.
From the visual information, it appears that $AP$ is approximately four times the length of $CB$. Therefore, a reasonable estimate for $X$ is 4.
The correct answer is (c) 4. | ✓ | (b) 2 | ✗ |
| InternVL2-76B | To estimate the length of AP relative to CB, we can visually compare the segments.
- CB is a chord of the circle. - AP extends from the circle's edge at A to point P, which is significantly farther away than B.
From the visual information, it appears that AP is roughly four times the length of CB.
Therefore, the reasonable estimate of X is 4.
Final answer: (c) 4 | ✓ | (b) | ✗ |
| Claude 3.5 Sonnet | Looking at the image, I can see a circle with its center O, and a triangle APB where P is a point on the circle. Line CB is a chord of the circle, passing through its diameter.
Visually comparing the lengths of AP and CB:
1. AP extends from a point on the circle to a point well outside it. 2. CB is a chord within the circle, noticeably shorter than the diameter. 3. AP appears to be significantly longer than CB - roughly 3 to 4 times longer.
Among the given options, the one that best matches this visual estimate is (c) 4. Option (a) 0.25 and (d) 0.5 would imply AP is shorter than CB, which is clearly not the case. Option (e) 1 would mean they're equal, which is also not true. Option (b) 2 is closer but still seems to underestimate the difference.
Therefore, based on visual estimation alone, the most reasonable answer is:
(c) 4 | ✓ | c | ✓ |
| GPT-4o | From the visual information, line segment $AP$ appears to be approximately twice the length of segment $CB$. Therefore, a reasonable estimate for $X$ is 2.
Final answer: (b) 2 | ✗ | c | ✓ |
| Gemini 1.5 Pro | Visually, AP appears roughly twice as long as the diameter of the circle. CB is a chord that is close in length to the diameter (although clearly shorter). Therefore, AP appears to be roughly twice the length of CB.
Final Answer: (b) | ✗ | b | ✗ |

Table 21: Examples of dataset and model responses for Geometry-Length (1)

| Image | Question | Gold Answer |
|---|---|---|
|  | Line CE is X times longer than AD. Which of the following options is a reasonable estimate of X? You only need to estimate from the visual information and do not need to do any mathematical reasoning. (a) 0.5 (b) 1 (c) 2 (d) 4 (e) 0.25 | a |

| Model | Answer w/ CoT | | Answer w/o CoT | |
|---|---|---|---|---|
| InternVL2-4B | To estimate the value of X, we need to visually compare the lengths of CE and AD. 1. **Observation**: - CE is the chord of the circle. - AD is the radius of the circle. 2. **Comparison**: - From the diagram, it is clear that CE is much longer than AD. - The radius (AD) appears to be about half the length of the diameter, which is twice the length of the radius. - CE appears to be about twice the length of the radius. 3. **Estimation**: - Since CE is approximately twice the length of AD, we can estimate that X is around 2. Therefore, the reasonable estimate of X is 2. **Final Answer**: (c) 2 | ✗ | c | ✗ |
| InternVL2-8B | To estimate the value of $X$ based on the visual information, we need to compare the lengths of $CE$ and $AD$. 1. **Visual Comparison**: - $CE$ appears to be about twice the length of $AD$. This is a rough estimate based on the visual proportions in the image. 2. **Reasoning**: - If $CE$ is twice as long as $AD$, then $X$ would be 2. Thus, the most reasonable estimate of $X$ is 2. **Final Answer**: (c) 2 | ✗ | c | ✗ |
| InternVL2-26B | To estimate the value of $X$, we can visually compare the lengths of $CE$ and $AD$ in the given circle diagram. From the diagram, it appears that $CE$ is approximately twice the length of $AD$. This suggests that $X$ is around 2. Therefore, the reasonable estimate for $X$ is (c) 2. So, the final answer is (c) 2. | ✗ | c | ✗ |
| InternVL2-76B | To estimate the length of CE relative to AD, observe the visual proportions in the circle. Line CE appears to be roughly twice the length of AD. Given the options: (a) 0.5 (b) 1 (c) 2 (d) 4 (e) 0.25 The most reasonable estimate is (c) 2. Final answer: (c) 2 | ✗ | (c) 2 | ✗ |
| Claude 3.5 Sonnet | Based on the visual information provided in the image, I can estimate the relative lengths of lines CE and AD. Line CE appears to be a diameter or very close to a diameter of the circle, as it passes through or very near the center. Line AD, on the other hand, is a chord that does not pass through the center and is visibly shorter than CE. Comparing these two lines visually, CE looks to be about twice the length of AD. It's clearly longer, but not dramatically so - certainly not four times as long. At the same time, CE is definitely more than just a little bit longer than AD, ruling out the options where CE would be equal to or shorter than AD. Given these observations, the most reasonable estimate for X (where CE is X times longer than AD) would be 2. Therefore, my answer is: (c) 2 | ✗ | b | ✗ |
| GPT-4o | From the image, line $CE$ appears to be about twice as long as line $AD$. Visually estimating, $CE$ is significantly longer than $AD$, but not excessively so. Therefore, the reasonable estimate for $X$ is 2. The final answer is (c) 2. | ✗ | c | ✗ |
| Gemini 1.5 Pro | CE looks to be roughly the same length as AD. Therefore, X is approximately 1. Final Answer: (b) | ✗ | (a) | ✓ |

Table 22: Examples of dataset and model responses for `Geometry-Length` (2)

| Image | Question | Gold Answer |
|---|---|---|
|  | Line BC is X times longer than AD. Which of the following options is a reasonable estimate of X? You only need to estimate from the visual information and do not need to do any mathematical reasoning. (a) 1 (b) 0.25 (c) 2 (d) 4 (e) 0.5 | d |

| Model | Answer w/ CoT | | Answer w/o CoT | |
|---|---|---|---|---|
| InternVL2-4B | To estimate the value of X, we need to visually compare the lengths of BC and AD. 1. **Observation**: - AD appears to be about half the length of BC. - This is a rough visual estimate based on the proportions of the triangle. 2. **Reasoning**: - If AD is approximately half the length of BC, then BC is approximately twice the length of AD. - Therefore, X, which represents the ratio of BC to AD, is approximately 2. 3. **Conclusion**: - The reasonable estimate of X from the visual information is 2. **Final Answer**: (c) 2 | ✗ | c | ✗ |
| InternVL2-8B | To estimate the value of \( X \) based on the visual information, we need to compare the lengths of \( BC \) and \( AD \). 1. **Visual Comparison**: - \( BC \) appears to be about twice the length of \( AD \). This is a rough estimate based on the visual proportions in the image. 2. **Reasoning**: - If \( BC \) is twice as long as \( AD \), then \( X \) would be 2. Therefore, the most reasonable estimate of \( X \) is 2. **Final Answer**: (c) 2 | ✗ | c | ✗ |
| InternVL2-26B | To estimate the value of \( X \), we need to compare the lengths of \( BC \) and \( AD \) visually. From the diagram, it appears that \( BC \) is significantly longer than \( AD \). A reasonable estimate for \( X \) would be a value that reflects this visual disparity. Option (a) 1 suggests that \( BC \) is equal in length to \( AD \), which is not visually accurate. Option (b) 0.25 suggests that \( BC \) is one-fourth the length of \( AD \), which also does not match the visual information. Option (c) 2 suggests that \( BC \) is twice the length of \( AD \), which is closer to the visual estimate but still not quite accurate. Option (d) 4 suggests that \( BC \) is four times the length of \( AD \), which visually seems to be a reasonable estimate. Option (e) 0.5 suggests that \( BC \) is half the length of \( AD \), which is not visually accurate. Given the visual information, the most reasonable estimate for \( X \) is 4. Final answer: (d) 4 | ✓ | c | ✗ |
| InternVL2-76B | To estimate the length of BC relative to AD, observe the visual proportions in the triangle. The line segment AD appears to be approximately half the length of BC. Given the options: (a) 1 - This would mean AD and BC are equal, which is not visually accurate. (b) 0.25 - This would mean BC is four times longer than AD, which seems too long. (c) 2 - This would mean BC is twice as long as AD, which looks about right. (d) 4 - This would mean BC is four times longer than AD, which seems too long. (e) 0.5 - This would mean BC is half the length of AD, which is incorrect. The most reasonable estimate from the visual information is (c) 2. Final answer: (c) | ✗ | (d) | ✓ |
| Claude 3.5 Sonnet | Looking at the image, we can see a triangle ABC with a smaller triangle ADE inside it. The line BC forms the base of the larger triangle, while AD is one of the sides of the smaller triangle. Visually comparing the lengths of BC and AD: 1. BC appears to be significantly longer than AD. 2. BC seems to span almost the entire width of the image, while AD is a much shorter line segment. 3. AD looks to be roughly 1/4 to 1/3 of the length of BC. Given these observations, we can estimate that BC is approximately 3 to 4 times longer than AD. This means X, which represents how many times longer BC is compared to AD, should be around 3 or 4. Among the given options, the closest value to this visual estimate is 4. Therefore, the final answer is: (d) 4 | ✓ | c | ✗ |
| GPT-4o | From the visual information, line BC appears to be significantly longer than line AD. It looks like BC is about four times the length of AD. Therefore, a reasonable estimate for X is 4. The final answer is (d) 4. | ✓ | d | ✓ |
| Gemini 1.5 Pro | BC appears to be roughly twice the length of AD. Final Answer: (c) | ✗ | c | ✗ |

Table 23: Examples of dataset and model responses for Geometry-Length (3)

| Image | Question | Gold Answer |
|---|---|---|
|  | Which of the following options is a reasonable estimate of the angle CAB in the figure? You only need to estimate from the visual information and do not need to do any mathematical reasoning. (a) 90 degrees (b) 45 degrees (c) 135 degrees (d) 10 degrees (e) 180 degrees | a |

| Model | Answer w/ CoT | | Answer w/o CoT | |
|---|---|---|---|---|
| InternVL2-4B | To estimate the angle CAB, we can visually assess the triangle and the right angle at point D. The angle at point A is formed by the line segments AC and AB. Since the triangle is a right triangle with a right angle at D, the sum of the angles in a triangle is 180 degrees. The right angle at D is 90 degrees, so the remaining two angles (CAB and BDC) must add up to 90 degrees.
Given that angle BDC is a right angle, angle CAB must be less than 90 degrees. Among the given options, the only reasonable estimate for angle CAB is 45 degrees.
Final answer: (b) 45 degrees | ✗ | b | ✗ |
| InternVL2-8B | The figure shows a right triangle with a right angle at point D. The angle CAB is the angle at the top of the triangle, which is opposite the side BC. Given that the right angle is at D, and the triangle is a right triangle, the angle CAB must be one of the other two angles in the triangle.
In a right triangle, the sum of the two non-right angles is 90 degrees. Therefore, the angle CAB must be less than 90 degrees.
Among the given options: - (a) 90 degrees is incorrect because it would make the triangle a straight line. - (b) 45 degrees is a reasonable estimate for an angle in a right triangle. - (c) 135 degrees is incorrect because it is greater than 90 degrees. - (d) 10 degrees is too small and not a reasonable estimate for an angle in a right triangle. - (e) 180 degrees is incorrect because it would make the triangle a straight line.
Therefore, the most reasonable estimate for the angle CAB is 45 degrees.
Final answer: (b) 45 degrees | ✗ | b | ✗ |
| InternVL2-26B | To estimate the angle CAB, we can use the concept of similar triangles. Since AD is the altitude of the triangle ABC, it divides the triangle into two smaller right triangles, $\ensuremath{\triangle}$ABD and $\ensuremath{\triangle}$ACD. In a right triangle, the angle opposite the altitude is a right angle (90 degrees). Therefore, $\ensuremath{\angle}$BAD and $\ensuremath{\angle}$CAD are both 90 degrees. Since $\ensuremath{\angle}$BAC is the angle between the two legs of the right triangle, it must be the remaining angle in the triangle, which is 90 degrees. Therefore, the angle CAB is 90 degrees.
The final answer is (a) 90 degrees. | ✓ | (b) 45 degrees | ✗ |
| InternVL2-76B | To estimate the angle CAB, observe the triangle ABC and the line AD. The line AD appears to bisect the angle BAC into two equal parts. Given that triangle ABC is isosceles (AB = AC), the angle at A (BAC) is typically around 70-80 degrees. Since AD bisects this angle, each of the two resulting angles (BAD and CAD) would be approximately half of the original angle.
Therefore, the angle CAB (which is one of these bisected angles) would be around 45 degrees.
Final answer: (b) 45 degrees | ✗ | (b) 45 degrees | ✗ |
| Claude 3.5 Sonnet | Based on the visual information provided in the image, I can estimate the angle CAB as follows:
The triangle shown appears to be roughly isosceles, with the two sides CA and BA looking approximately equal in length. The angle at the top vertex (angle CAB) seems to be less than a right angle (90 degrees), but not by a large amount.
Given this visual assessment, the angle CAB looks to be close to 45 degrees. It's clearly much larger than 10 degrees and significantly smaller than 90 degrees or anything above that.
Among the given options, 45 degrees (option b) is the most reasonable estimate that matches what we can see in the figure.
Final answer: (b) | ✗ | b | ✗ |
| GPT-4o | The angle \( \angle CAB \) appears to be an acute angle, as it is less than a right angle (90 degrees). It is also significantly smaller than 45 degrees. Therefore, the most reasonable estimate from the given options is 10 degrees.
Final answer: (d) 10 degrees. | ✗ | b | ✗ |
| Gemini 1.5 Pro | Angle CAB looks less than 90 degrees, and closer to half of 90 degrees. 45 degrees is a reasonable estimate.
Final Answer: (b) | ✗ | b | ✗ |

Table 24: Examples of dataset and model responses for `Geometry-Angle` (1)

| Image | Question | Gold Answer |
|---|---|---|
|  | Which of the following options is a reasonable estimate of the angle AOD in the figure? You only need to estimate from the visual information and do not need to do any mathematical reasoning. (a) 180 degrees (b) 10 degrees (c) 135 degrees (d) 90 degrees (e) 45 degrees | d |

| Model | Answer w/ CoT | | Answer w/o CoT | |
|---|---|---|---|---|
| InternVL2-4B | To estimate the angle AOD, we need to consider the shape of the quadrilateral ABCD. The figure shows that ABCD is a square, as all sides are equal and all angles are right angles (90 degrees). In a square, the diagonals are equal in length and bisect each other at right angles. Therefore, each diagonal divides the square into two congruent right-angled triangles. Since the diagonals of a square bisect the angles of the square, each angle at the vertices of the square is divided into two equal parts. Thus, the angle AOD is half of the right angle at vertex A, which is 90 degrees. Therefore, the reasonable estimate for the angle AOD is 45 degrees. Final answer: (e) 45 degrees | ✗ | d | ✓ |
| InternVL2-8B | To estimate the angle AOD in the figure, we can observe the following: 1. The figure shows a square ABCD with diagonals AC and BD intersecting at point O. 2. In a square, the diagonals are equal in length and bisect each other at right angles (90 degrees). Since the diagonals of a square intersect at right angles, the angle AOD is 90 degrees. Therefore, the reasonable estimate of the angle AOD is: (d) 90 degrees | ✓ | c | ✗ |
| InternVL2-26B | To estimate the angle AOD, we can observe that the figure is a rhombus, which means all sides are equal and opposite angles are equal. Since the diagonals of a rhombus bisect each other at right angles (90 degrees), the angle AOD is half of the angle formed by the diagonals at point O. Given that the diagonals bisect each other at right angles, the angle AOD is half of 90 degrees, which is 45 degrees. Therefore, the answer is (e) 45 degrees. | ✗ | (d) 90 degrees | ✓ |
| InternVL2-76B | To estimate the angle AOD, observe the figure and note the position of points A, O, and D. The diagonals of the parallelogram intersect at point O, and in a parallelogram, the diagonals bisect each other. Therefore, angle AOD is one of the angles formed by the intersection of the diagonals. From the visual information, angle AOD appears to be a right angle because the diagonals intersect at 90 degrees in a parallelogram. Thus, the reasonable estimate of angle AOD is 90 degrees. Final answer: (d) 90 degrees | ✓ | (d) 90 degrees | ✓ |
| Claude 3.5 Sonnet | Looking at the figure, we can see that AOD is one of the angles formed by the diagonals of what appears to be a quadrilateral (possibly a square or rectangle). The diagonals intersect at point O, creating four angles. From visual inspection, the angle AOD looks to be approximately half of a right angle. A right angle is 90 degrees, and half of that would be 45 degrees. The angle certainly doesn't look like 180 degrees (which would be a straight line), 10 degrees (which would be a very narrow angle), or 135 degrees (which would be larger than a right angle). It also doesn't appear to be a full 90 degrees, as it's visibly smaller than the right angles at the corners of the quadrilateral. Therefore, based on visual estimation alone, the most reasonable estimate for angle AOD is 45 degrees. Final answer: (e) 45 degrees | ✗ | d | ✓ |
| GPT-4o | From the visual information in the figure, angle $\angle AOD$ appears to be an acute angle, significantly less than 90 degrees. It looks closer to a small angle, possibly around 45 degrees. Therefore, the reasonable estimate for angle $\angle AOD$ is: (e) 45 degrees | ✗ | e | ✗ |
| Gemini 1.5 Pro | The angle AOD appears to be close to a right angle. A right angle measures 90 degrees. Final Answer: (d) | ✓ | c | ✗ |

Table 25: Examples of dataset and model responses for Geometry-Angle (2)

| Image | Question | Gold Answer |
|---|---|---|
|  | Which of the following options is a reasonable estimate of the angle BMN in the figure? You only need to estimate from the visual information and do not need to do any mathematical reasoning. (a) 90 degrees (b) 135 degrees (c) 180 degrees (d) 45 degrees (e) 10 degrees | b |

| Model | Answer w/ CoT | | Answer w/o CoT | |
|---|---|---|---|---|
| InternVL2-4B | To estimate the angle BMN in the figure, we need to consider the shape and the given angles. The figure shows a quadrilateral with a diagonal, and the angles at vertices A and C are right angles (90 degrees).
Given that the angles at A and C are right angles, the sum of the angles in a quadrilateral is 360 degrees. Since the other two angles (at vertices B and D) are not directly given, we can infer that the angle BMN is likely to be a part of the remaining angles that sum up to 360 degrees.
Considering the options provided: (a) 90 degrees - This is the angle at vertex A, not BMN. (b) 135 degrees - This is a reasonable estimate for the angle BMN, as it is a large angle but not as large as 180 degrees. (c) 180 degrees - This would imply a straight line, which is not the case here. (d) 45 degrees - This is too small for the angle BMN. (e) 10 degrees - This is too small for the angle BMN.
Therefore, the most reasonable estimate for the angle BMN is 135 degrees.
Final answer: (b) 135 degrees. | ✓ | b | ✓ |
| InternVL2-8B | The angle BMN appears to be a right angle, as indicated by the small square at the intersection of lines BM and MN. This typically represents a 90-degree angle in geometric diagrams. Therefore, the most reasonable estimate of the angle BMN is 90 degrees.
Final answer: (a) 90 degrees | ✗ | b | ✓ |
| InternVL2-26B | To estimate the angle BMN, we can observe the given geometric figure. The figure shows a right-angled triangle (ABD) and a rectangle (ABCD). In a rectangle, all interior angles are 90 degrees. Since BMN is part of the rectangle, it is reasonable to estimate that angle BMN is also 90 degrees.
Final answer: (a) 90 degrees | ✗ | (b) 135 degrees | ✓ |
| InternVL2-76B | To estimate the angle BMN, observe the figure and the relative positions of points B, M, and N. The angle BMN appears to be a right angle because line BM is perpendicular to line MN.
Therefore, the reasonable estimate of the angle BMN is 90 degrees.
Final answer: (a) 90 degrees | ✗ | (b) 135 degrees | ✓ |
| Claude 3.5 Sonnet | Looking at the figure, I can see that angle BMN is formed by two lines that meet at point M. This angle appears to be significantly larger than a right angle (90 degrees), but it's not a straight line (180 degrees).
The angle looks to be roughly halfway between a right angle and a straight line. This would put it close to 135 degrees, which is one of the options provided.
It's clearly much larger than 45 degrees or 10 degrees, and noticeably larger than 90 degrees. However, it doesn't extend all the way to form a straight line, so 180 degrees would be too large.
Based on this visual estimation, the most reasonable estimate among the given options would be 135 degrees.
Final answer: (b) | ✓ | d | ✗ |
| GPT-4o | The angle \( \angle BMN \) appears to be an acute angle, as it is smaller than a right angle. Visually, it seems closer to a small angle rather than a large one. Among the options, 10 degrees is the most reasonable estimate for an acute angle.
Final answer: (e) 10 degrees | ✗ | d | ✗ |
| Gemini 1.5 Pro | Angle BMN appears to be slightly larger than a right angle (90 degrees). 135 degrees seems like a reasonable estimate.
Final Answer: (b) | ✓ | b | ✓ |

Table 26: Examples of dataset and model responses for Geometry-Angle (3)

| Image | Question | Gold Answer |
|---|---|---|
| 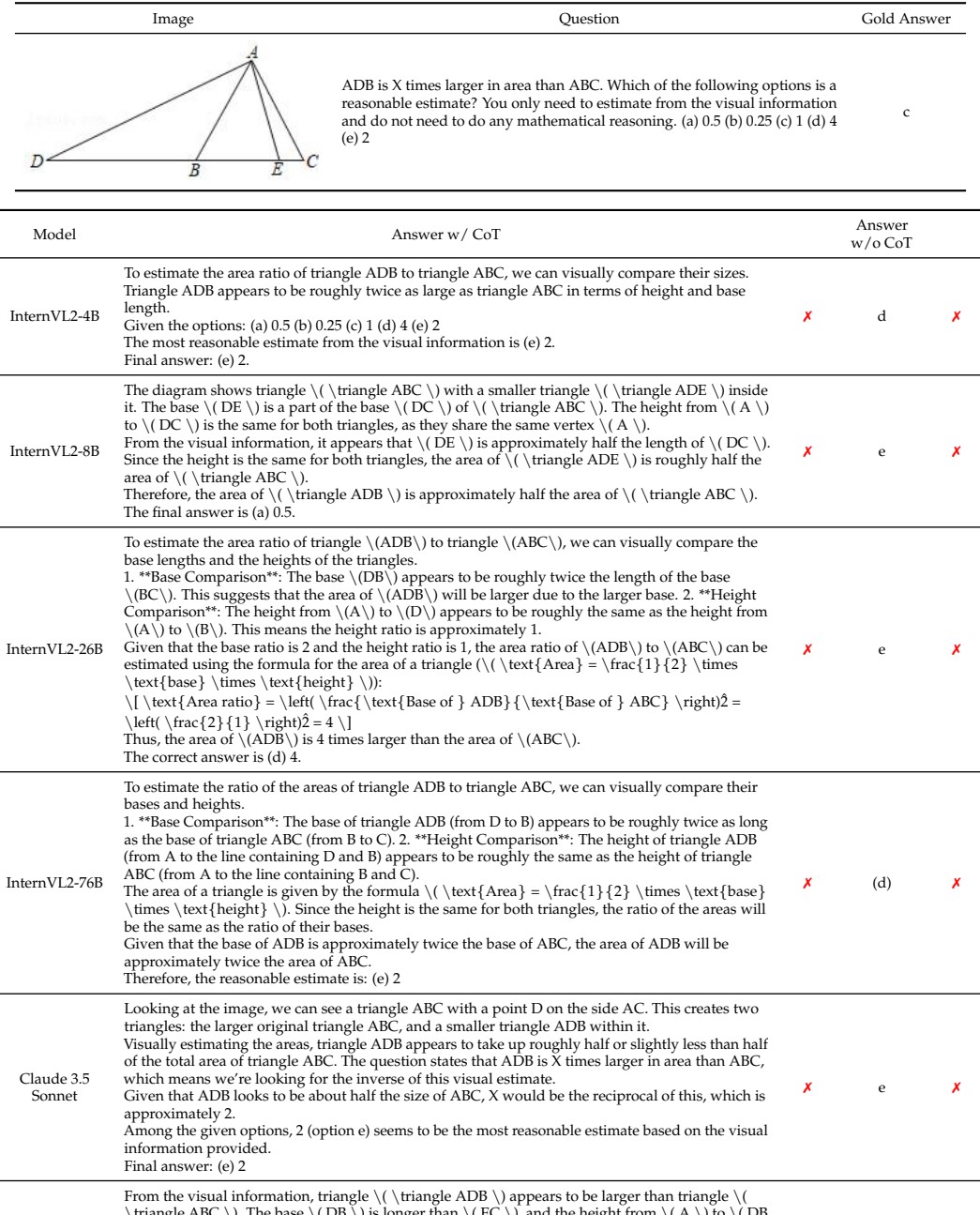 | ADB is X times larger in area than ABC. Which of the following options is a reasonable estimate? You only need to estimate from the visual information and do not need to do any mathematical reasoning. (a) 0.5 (b) 0.25 (c) 1 (d) 4 (e) 2 | c |

| Model | Answer w/ CoT | | Answer w/o CoT | |
|---|---|---|---|---|
| InternVL2-4B | To estimate the area ratio of triangle ADB to triangle ABC, we can visually compare their sizes. Triangle ADB appears to be roughly twice as large as triangle ABC in terms of height and base length. Given the options: (a) 0.5 (b) 0.25 (c) 1 (d) 4 (e) 2 The most reasonable estimate from the visual information is (e) 2. Final answer: (e) 2. | ✗ | d | ✗ |
| InternVL2-8B | The diagram shows triangle $\triangle ABC$ with a smaller triangle $\triangle ADE$ inside it. The base $DE$ is a part of the base $DC$ of $\triangle ABC$. The height from $A$ to $DC$ is the same for both triangles, as they share the same vertex $A$. From the visual information, it appears that $DE$ is approximately half the length of $DC$. Since the height is the same for both triangles, the area of $\triangle ADE$ is roughly half the area of $\triangle ABC$. Therefore, the area of $\triangle ADB$ is approximately half the area of $\triangle ABC$. The final answer is (a) 0.5. | ✗ | e | ✗ |
| InternVL2-26B | To estimate the area ratio of triangle $ADB$ to triangle $ABC$, we can visually compare the base lengths and the heights of the triangles. 1. **Base Comparison**: The base $DB$ appears to be roughly twice the length of the base $BC$. This suggests that the area of $ADB$ will be larger due to the larger base. 2. **Height Comparison**: The height from $A$ to $D$ appears to be roughly the same as the height from $A$ to $B$. This means the height ratio is approximately 1. Given that the base ratio is 2 and the height ratio is 1, the area ratio of $ADB$ to $ABC$ can be estimated using the formula for the area of a triangle ($\text{Area} = \frac{1}{2} \times \text{base} \times \text{height}$): $$\text{Area ratio} = \left(\frac{\text{Base of } ADB}{\text{Base of } ABC}\right)^2 = \left(\frac{2}{1}\right)^2 = 4$$ Thus, the area of $ADB$ is 4 times larger than the area of $ABC$. The correct answer is (d) 4. | ✗ | e | ✗ |
| InternVL2-76B | To estimate the ratio of the areas of triangle ADB to triangle ABC, we can visually compare their bases and heights. 1. **Base Comparison**: The base of triangle ADB (from D to B) appears to be roughly twice as long as the base of triangle ABC (from B to C). 2. **Height Comparison**: The height of triangle ADB (from A to the line containing D and B) appears to be roughly the same as the height of triangle ABC (from A to the line containing B and C). The area of a triangle is given by the formula $\text{Area} = \frac{1}{2} \times \text{base} \times \text{height}$. Since the height is the same for both triangles, the ratio of the areas will be the same as the ratio of their bases. Given that the base of ADB is approximately twice the base of ABC, the area of ADB will be approximately twice the area of ABC. Therefore, the reasonable estimate is: (e) 2 | ✗ | (d) | ✗ |
| Claude 3.5 Sonnet | Looking at the image, we can see a triangle ABC with a point D on the side AC. This creates two triangles: the larger original triangle ABC, and a smaller triangle ADB within it. Visually estimating the areas, triangle ADB appears to take up roughly half or slightly less than half of the total area of triangle ABC. The question states that ADB is X times larger in area than ABC, which means we're looking for the inverse of this visual estimate. Given that ADB looks to be about half the size of ABC, X would be the reciprocal of this, which is approximately 2. Among the given options, 2 (option e) seems to be the most reasonable estimate based on the visual information provided. Final answer: (e) 2 | ✗ | e | ✗ |
| GPT-4o | From the visual information, triangle $\triangle ADB$ appears to be larger than triangle $\triangle ABC$. The base $DB$ is longer than $EC$, and the height from $A$ to $DB$ is the same as the height to $EC$. This suggests that $\triangle ADB$ is significantly larger than $\triangle ABC$. A reasonable estimate is that $\triangle ADB$ is about twice the size of $\triangle ABC$. Final answer: (e) 2 | ✗ | e | ✗ |
| Gemini 1.5 Pro | ADB appears roughly twice as large as ABC. This is because ADB encompasses most of the area within triangle ADC, which itself is roughly the same area as triangle ABC. Therefore, ADB is approximately 2 times larger than ABC. Final Answer: (e) | ✗ | e | ✗ |

Table 27: Examples of dataset and model responses for Geometry-Area (1)

| Image | Question | Gold Answer |
|---|---|---|
| 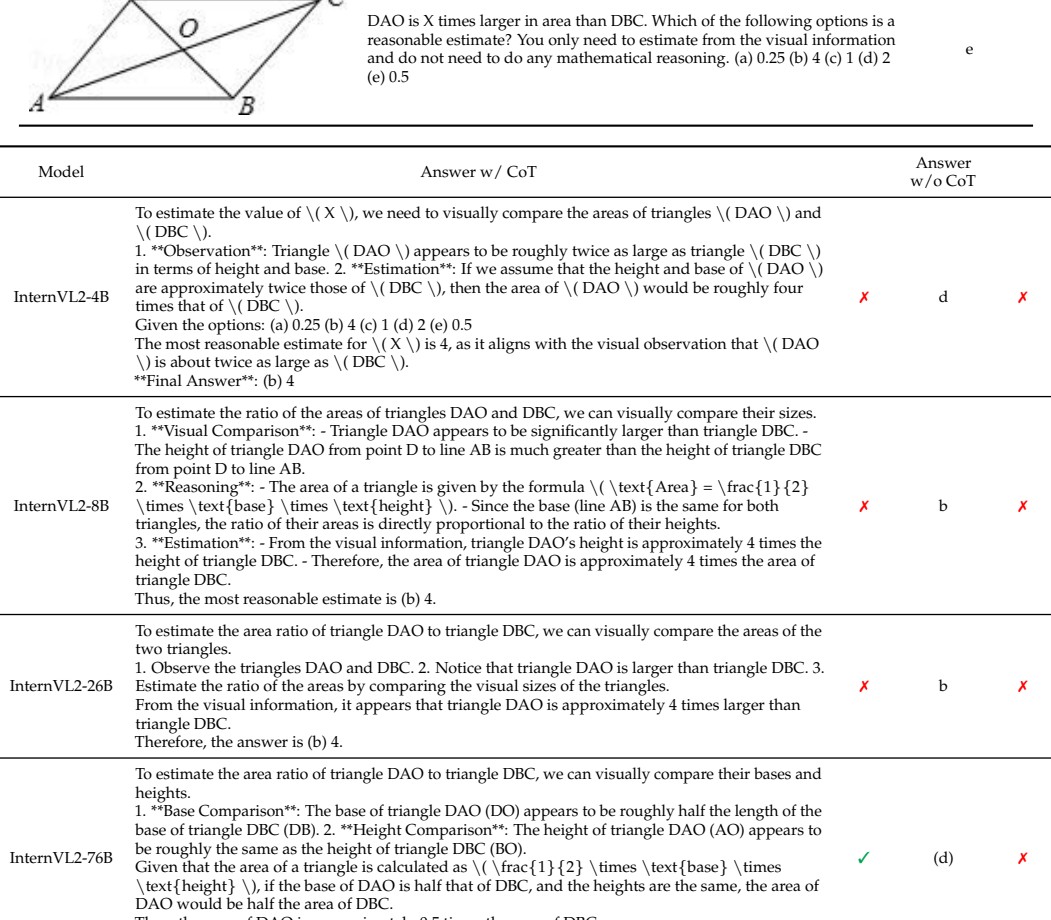 | DAO is X times larger in area than DBC. Which of the following options is a reasonable estimate? You only need to estimate from the visual information and do not need to do any mathematical reasoning. (a) 0.25 (b) 4 (c) 1 (d) 2 (e) 0.5 | e |

| Model | Answer w/ CoT | | Answer w/o CoT | |
|---|---|---|---|---|
| InternVL2-4B | To estimate the value of \( X \), we need to visually compare the areas of triangles \( DAO \) and \( DBC \).
1. **Observation**: Triangle \( DAO \) appears to be roughly twice as large as triangle \( DBC \) in terms of height and base. 2. **Estimation**: If we assume that the height and base of \( DAO \) are approximately twice those of \( DBC \), then the area of \( DAO \) would be roughly four times that of \( DBC \).
Given the options: (a) 0.25 (b) 4 (c) 1 (d) 2 (e) 0.5
The most reasonable estimate for \( X \) is 4, as it aligns with the visual observation that \( DAO \) is about twice as large as \( DBC \).
**Final Answer**: (b) 4 | ✗ | d | ✗ |
| InternVL2-8B | To estimate the ratio of the areas of triangles DAO and DBC, we can visually compare their sizes.
1. **Visual Comparison**: - Triangle DAO appears to be significantly larger than triangle DBC. - The height of triangle DAO from point D to line AB is much greater than the height of triangle DBC from point D to line AB.
2. **Reasoning**: - The area of a triangle is given by the formula \( \text{Area} = \frac{1}{2} \times \text{base} \times \text{height} \). - Since the base (line AB) is the same for both triangles, the ratio of their areas is directly proportional to the ratio of their heights.
3. **Estimation**: - From the visual information, triangle DAO's height is approximately 4 times the height of triangle DBC. - Therefore, the area of triangle DAO is approximately 4 times the area of triangle DBC.
Thus, the most reasonable estimate is (b) 4. | ✗ | b | ✗ |
| InternVL2-26B | To estimate the area ratio of triangle DAO to triangle DBC, we can visually compare the areas of the two triangles.
1. Observe the triangles DAO and DBC. 2. Notice that triangle DAO is larger than triangle DBC. 3. Estimate the ratio of the areas by comparing the visual sizes of the triangles.
From the visual information, it appears that triangle DAO is approximately 4 times larger than triangle DBC.
Therefore, the answer is (b) 4. | ✗ | b | ✗ |
| InternVL2-76B | To estimate the area ratio of triangle DAO to triangle DBC, we can visually compare their bases and heights.
1. **Base Comparison**: The base of triangle DAO (DO) appears to be roughly half the length of the base of triangle DBC (DB). 2. **Height Comparison**: The height of triangle DAO (AO) appears to be roughly the same as the height of triangle DBC (BO).
Given that the area of a triangle is calculated as \( \frac{1}{2} \times \text{base} \times \text{height} \), if the base of DAO is half that of DBC, and the heights are the same, the area of DAO would be half the area of DBC.
Thus, the area of DAO is approximately 0.5 times the area of DBC.
Final answer: (e) 0.5 | ✓ | (d) | ✗ |
| Claude 3.5 Sonnet | Looking at the image, we can see a quadrilateral ABCD with a diagonal AC that divides it into two triangles: DAO and DBC.
Visually estimating the areas:
Triangle DAO appears to be significantly larger than triangle DBC. It covers more than half of the entire quadrilateral, while DBC covers less than half.
The question states that DAO is X times larger in area than DBC. Given this information and the visual estimate, we can reasonably conclude that DAO is about twice the size of DBC.
Among the given options, the value that best represents this relationship is 2.
Therefore, my reasoning leads to the answer: (d) 2 | ✗ | b | ✗ |
| GPT-4o | From the image, triangle \( \triangle DAO \) and triangle \( \triangle DBC \) share the same height from point \( D \) to line \( AC \). The base \( AO \) of \( \triangle DAO \) appears to be shorter than the base \( BC \) of \( \triangle DBC \).
Visually, \( \triangle DAO \) seems to be about half the size of \( \triangle DBC \). Therefore, a reasonable estimate for the ratio of their areas is 0.5.
The final answer is (e) 0.5. | ✓ | (e) 0.5 | ✓ |
| Gemini 1.5 Pro | The triangle DAO appears to be roughly half the size of triangle DBC. Therefore, DBC is roughly twice as large as DAO, meaning that DAO is roughly 0.5 times larger than DBC.
Final Answer: (e) | ✓ | e | ✓ |

Table 28: Examples of dataset and model responses for `Geometry-Area` (2)

| Image | Question | Gold Answer |
|---|---|---|
| 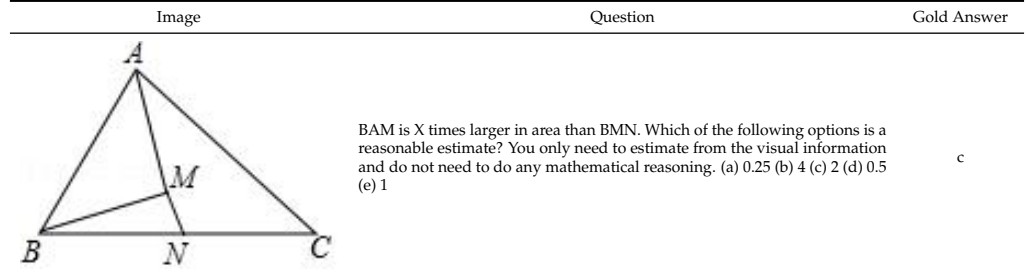 | BAM is X times larger in area than BMN. Which of the following options is a reasonable estimate? You only need to estimate from the visual information and do not need to do any mathematical reasoning. (a) 0.25 (b) 4 (c) 2 (d) 0.5 (e) 1 | c |

| Model | Answer w/ CoT | | Answer w/o CoT | |
|---|---|---|---|---|
| InternVL2-4B | To estimate the value of \( X \), we need to visually compare the areas of triangles \( BAM \) and \( BMN \).
1. **Observation**: Triangle \( BAM \) is visually larger than triangle \( BMN \). 2. **Estimation**: By visually comparing the two triangles, it appears that \( BAM \) is roughly 4 times larger in area than \( BMN \).
Therefore, the reasonable estimate for \( X \) is 4.
**Final Answer**: (b) 4 | ✗ | b | ✗ |
| InternVL2-8B | To estimate the ratio of the areas of triangles BAM and BMN, we can visually compare their sizes.
1. **Visual Comparison**: Triangle BAM appears to be significantly larger than triangle BMN. The base of triangle BAM (AB) is much longer than the base of triangle BMN (BN), and the height from point A to BC is also greater than the height from point M to BC.
2. **Area Relationship**: The area of a triangle is given by the formula \( \text{Area} = \frac{1}{2} \times \text{base} \times \text{height} \). Since both triangles share the same height (the perpendicular distance from A to BC), the ratio of their areas will be directly proportional to the ratio of their bases.
3. **Base Comparison**: The base of triangle BAM (AB) is approximately 4 times the length of the base of triangle BMN (BN).
4. **Area Ratio**: Given that the height is the same for both triangles, the area of triangle BAM will be approximately 4 times the area of triangle BMN.
Therefore, the most reasonable estimate for the ratio of the areas of triangles BAM to BMN is 4.
**Final Answer**: (b) 4 | ✗ | b | ✗ |
| InternVL2-26B | To estimate the ratio of the areas of triangles BAM and BMN, we can visually compare the bases and the heights of the triangles.
1. **Base Comparison**: The base of triangle BAM (AB) appears to be roughly twice the length of the base of triangle BMN (BN). This suggests that the area of triangle BAM will be approximately four times the area of triangle BMN, assuming the height remains constant.
2. **Height Comparison**: The height of triangle BAM (AM) appears to be roughly the same as the height of triangle BMN (MN). This means that the height does not significantly affect the area ratio, as the height is a common factor in both triangles.
Given these observations, the area of triangle BAM (ABM) is likely to be around four times the area of triangle BMN (BMN).
Therefore, the correct answer is (b) 4. | ✗ | c | ✓ |
| InternVL2-76B | To estimate the area ratio between triangle BAM and triangle BMN, observe the following:
1. The height of triangle BAM from point A to side BM is significantly larger than the height of triangle BMN from point M to side BN. 2. The base lengths of both triangles are proportional to the segments BM and MN of line BC.
Given that the height of BAM is roughly twice the height of BMN, and the base lengths are in a similar proportion, the area of triangle BAM would be approximately four times the area of triangle BMN.
Therefore, the reasonable estimate for the area ratio is 4.
Final answer: (b) 4 | ✗ | (b) | ✗ |
| Claude 3.5 Sonnet | Looking at the image, we can see a triangle ABC with a point M on side AC and a point N on side BC. These points create two triangles: the larger triangle BAM and the smaller triangle BMN. Visually estimating the areas, it appears that triangle BAM is noticeably larger than triangle BMN, but not dramatically so. It looks to be about twice the size, give or take.
Among the given options, the one that best matches this visual estimate is:
(c) 2
This suggests that BAM is approximately 2 times larger in area than BMN, which seems reasonable based on the visual information provided in the diagram.
Final answer: c | ✓ | b | ✗ |
| GPT-4o | From the image, triangle \( \triangle BAM \) appears to be larger than triangle \( \triangle BMN \). The area of \( \triangle BAM \) seems to be approximately twice the area of \( \triangle BMN \), as \( M \) is roughly halfway along \( \overline{AN} \) and \( \overline{BM} \) is a common side.
Therefore, a reasonable estimate for the ratio of the areas is 2.
Final answer: (c) 2 | ✓ | b | ✗ |
| Gemini 1.5 Pro | Triangle BAM looks roughly twice the size of triangle BMN. Therefore, X is approximately 2.
Final Answer: (c) | ✓ | (c) | ✓ |

Table 29: Examples of dataset and model responses for Geometry-Area (3)

| Image | Question | Gold Answer |
|---|---|---|
|  | In the figure, the line AB is a diameter of a circle. True or False? | True |

| Model | Answer w/ CoT | | Answer w/o CoT | |
|---|---|---|---|---|
| InternVL2-4B | True.
The line AB is a diameter of a circle because it passes through the center of the circle, O, and connects two points on the circumference, A and B. By definition, a diameter is a straight line passing from side to side through the center of a circle or sphere. | ✓ | True | ✓ |
| InternVL2-8B | False.
In the given figure, the line segment AB is not a diameter of the circle. A diameter is a line segment that passes through the center of the circle and has its endpoints on the circle. However, in this figure, the line segment AB does not pass through the center of the circle (point O). Instead, it is a chord that connects two points on the circumference of the circle. Therefore, AB is not a diameter. | ✗ | False | ✗ |
| InternVL2-26B | To determine if the statement is true or false, we need to analyze the given figure. The line AB is shown as a straight line passing through the center O of the circle, which is a characteristic of a diameter. A diameter is defined as a straight line passing from side to side through the center of a circle, thus dividing it into two equal parts. Given that AB passes through the center O and extends from one side of the circle to the other, it meets the criteria of being a diameter.
Therefore, the statement is true.
True. | ✓ | True | ✓ |
| InternVL2-76B | To determine if the line AB is a diameter of the circle, we need to check if it passes through the center of the circle, denoted as O.
From the figure, it is clear that the line AB does indeed pass through the center O. Therefore, AB is a diameter of the circle.
Final answer: True | ✓ | True | ✓ |
| Claude 3.5 Sonnet | Let's analyze this step-by-step:
1) In a circle, a diameter is defined as a line segment that passes through the center of the circle and has its endpoints on the circle. It's the longest chord of the circle.
2) In this figure, we can see that line AB passes through point O, which appears to be the center of the circle.
3) Both points A and B are on the circumference of the circle.
4) The line AB divides the circle into two equal halves.
5) These characteristics are consistent with the definition of a diameter.
Therefore, based on the information provided in the image, the statement appears to be correct.
Final answer: True | ✓ | True | ✓ |
| GPT-4o | In the figure, $\( AB \)$ is a chord of the circle, and $\( E \)$ is a point on the circle such that $\( \triangle AEB \)$ is inscribed. For $\( AB \)$ to be a diameter, the angle $\( \angle AEB \)$ must be a right angle (90 degrees) according to the inscribed angle theorem. However, the figure does not provide enough information to confirm that $\( \angle AEB \)$ is a right angle.
Therefore, based on the given information, we cannot conclude that $\( AB \)$ is a diameter.
Final answer: False. | ✗ | True. | ✓ |
| Gemini 1.5 Pro | A diameter of a circle is a line segment that passes through the center of the circle and whose endpoints lie on the circle. In this figure, AB passes through point O (the circle's center) and points A and B lie on the circle.
Final Answer: True | ✓ | True | ✓ |

Table 30: Examples of dataset and model responses for Geometry-Diameter-Radius (1)

| Image | Question | Gold Answer |
|---|---|---|
|  | In the figure, the line DC is a diameter of a circle. True or False? | True |

| Model | Answer w/ CoT | | Answer w/o CoT | |
|---|---|---|---|---|
| InternVL2-4B | True.
The line DC passes through the center of the circle, O, and extends to both ends of the circle, touching the circumference at points C and F. This means that DC is a diameter of the circle.
Final answer: True. | ✓ | True | ✓ |
| InternVL2-8B | False
In the figure, the line segment CD is not a diameter of the circle. A diameter is a line segment that passes through the center of the circle and has its endpoints on the circle. In the given figure, the line segment CD does not pass through the center of the circle, and therefore it is not a diameter. | ✗ | False | ✗ |
| InternVL2-26B | To determine if the statement is true or false, we need to analyze the given figure.
The figure shows a circle with a line segment DC passing through the center O. Since DC passes through the center of the circle and extends to both ends of the circle, it is a diameter.
Therefore, the statement "the line DC is a diameter of a circle" is true.
True. | ✓ | True | ✓ |
| InternVL2-76B | To determine if the line DC is a diameter of the circle, we need to check if it passes through the center of the circle and if its endpoints lie on the circle.
From the figure, it is clear that: 1. The line DC passes through the center O of the circle. 2. The endpoints D and C lie on the circumference of the circle.
Since the line DC satisfies both conditions, it is indeed a diameter of the circle.
Final answer: True | ✓ | True | ✓ |
| Claude 3.5 Sonnet | Let's analyze this step-by-step:
1. In a circle, a diameter is defined as a line segment that passes through the center of the circle and has its endpoints on the circle. It's the longest chord of the circle.
2. In this figure, we can see that line DC passes through point O, which appears to be the center of the circle (as it's typically denoted).
3. Both points D and C are on the circumference of the circle.
4. The line DC appears to divide the circle into two equal halves, which is a property of a diameter.
5. While we don't have explicit measurements, the visual representation strongly suggests that DC is indeed a diameter.
Reasoning: Given that DC passes through the center O and has its endpoints on the circle, it meets the definition of a diameter.
Final answer: True | ✓ | True | ✓ |
| GPT-4o | In the figure, \( DC \) is a line that passes through the center \( O \) of the circle, making it a diameter. A diameter is a line segment that passes through the center of the circle and has its endpoints on the circle. Since \( DC \) meets these criteria, it is indeed a diameter.
Final answer: True. | ✓ | True | ✓ |
| Gemini 1.5 Pro | A diameter is a line segment that passes through the center of a circle and whose endpoints lie on the circle. The line segment DC passes through the center O and its endpoints C and D lie on the circle. Therefore, DC is a diameter.
Final Answer: The final answer is $\boxed{True}$ | ✓ | True | ✓ |

Table 31: Examples of dataset and model responses for Geometry-Diameter-Radius (2)

| Image | Question | Gold Answer |
|---|---|---|
|  | In the figure, the line AO is a radius of a circle. True or False? | True |

| Model | Answer w/ CoT | | Answer w/o CoT | |
|---|---|---|---|---|
| InternVL2-4B | True.
Explanation: In the given figure, AO is a line segment that starts from point A and ends at point O, which is the center of the circle. Since O is the center of the circle, AO is a radius of the circle.
Final answer: True | ✓ | True | ✓ |
| InternVL2-8B | True
In the figure, the line segment AO is drawn from the center of the circle (point O) to a point on the circumference (point A). By definition, any line segment that connects the center of a circle to a point on the circle's circumference is a radius. Therefore, AO is a radius of the circle. | ✓ | True | ✓ |
| InternVL2-26B | In the given figure, the line segment AO is drawn from the center of the circle (O) to a point on the circle (A). Since AO is a line segment connecting the center of the circle to a point on the circle, it is a radius of the circle.
Therefore, the statement "In the figure, the line AO is a radius of a circle" is true.
True. | ✓ | True | ✓ |
| InternVL2-76B | The line AO is a radius of the circle because it extends from the center of the circle (O) to a point on the circle (A). By definition, a radius is a straight line from the center of a circle to any point on its circumference. Therefore, AO meets this criterion and is indeed a radius.
Final answer: True | ✓ | True | ✓ |
| Claude 3.5 Sonnet | Let's analyze this figure:
1. We see a circle with three points labeled A, B, and C on its circumference.
2. There's a line segment drawn from point A to the center of the circle, labeled O.
3. By definition, a radius of a circle is a line segment from the center of the circle to any point on its circumference.
4. The line AO meets this definition exactly - it goes from the center O to point A on the circumference.
5. While there are other line segments in the figure (BO and CO), we're specifically asked about AO.
Reasoning: Since AO is a line segment that connects the center of the circle to a point on its circumference, it fits the definition of a radius perfectly.
Final answer: True | ✓ | True | ✓ |
| GPT-4o | In the figure, \( O \) is the center of the circle, and \( A \) is a point on the circle. The line \( AO \) connects the center of the circle to a point on the circle, which is the definition of a radius.
Therefore, the statement is **True**. | ✓ | True. | ✓ |
| Gemini 1.5 Pro | The center of the circle is labeled as O. A radius of a circle is any line segment that connects the center to a point on the circumference (edge) of the circle. Point A is on the circumference of the circle. The line segment AO connects the center O to a point A on the circumference.
Final Answer: True | ✓ | True | ✓ |

Table 32: Examples of dataset and model responses for Geometry-Diameter-Radius (3)

| Image | Question | Gold Answer |
|---|---|---|
|  | CH_3 is attached to a single line, not a double line. True or False? You must answer this question using visual information and do not use knowledge of chemistry unless explicitly instructed to do so. | False |

| Model | Answer w/ CoT | | Answer w/o CoT | |
|---|---|---|---|---|
| InternVL2-4B | The image shows a chemical structure where CH_3 is attached to a single line, indicating a single bond between the carbon and the hydrogen atoms. False | ✓ | True | ✗ |
| InternVL2-8B | The image shows a chemical structure where a CH$_3$ group is attached to a nitrogen atom (N) via a single line. This indicates a single covalent bond between the CH$_3$ group and the nitrogen atom. Therefore, the statement "CH$_3$ is attached to a single line, not a double line" is true. Final answer: True | ✗ | True | ✗ |
| InternVL2-26B | The structure shows a single line representing a single bond between the carbon (C) and the nitrogen (N) atoms. The double line represents a double bond between the carbon and the oxygen (O) atoms. Therefore, the statement "CH_3 is attached to a single line, not a double line" is true. True. | ✗ | True | ✗ |
| InternVL2-76B | The image shows a chemical structure with a CH$_3$ group attached to a nitrogen atom (N) via a single line. This single line represents a single covalent bond between the carbon of the CH$_3$ group and the nitrogen atom. Final answer: True | ✗ | True | ✗ |
| Claude 3.5 Sonnet | Based on the visual information provided in the image: I can see that the CH3 group is indeed attached to a single line in the molecular structure diagram, not a double line. The single line connects the CH3 group to what appears to be a nitrogen atom (N) within the larger molecular structure. Final answer: True | ✗ | False | ✓ |
| GPT-4o | The CH$_3$ group is attached to a single line, which represents a single bond. True. | ✗ | True | ✗ |
| Gemini 1.5 Pro | The CH$_3$ is attached to what appears to be a vertex where three lines meet. This is a single bond, not a double bond (which would be represented by two lines). Final Answer: True | ✗ | True | ✗ |

Table 33: Examples of dataset and model responses for Chemistry-Shape-Single (1)

| Image | Question | Gold Answer |
|---|---|---|
|  | H_a and H_d are attached to opposite vertices of a hexagonal structure. True or False? You must answer this question using visual information and do not use knowledge of chemistry unless explicitly instructed to do so. | False |

| Model | Answer w/ CoT | | Answer w/o CoT | |
|---|---|---|---|---|
| InternVL2-4B | The structure depicted in the image is a hexagonal ring with alternating oxygen and nitrogen atoms. The hydrogen atoms (H_a, H_b, H_c, H_d, H_e) are attached to the carbon atoms of the hexagonal ring.
To determine if H_a and H_d are attached to opposite vertices of the hexagonal structure, we need to examine the positions of these hydrogen atoms in relation to the ring.
- H_a is attached to one of the carbon atoms of the hexagonal ring. - H_d is also attached to one of the carbon atoms of the hexagonal ring.
Since both H_a and H_d are attached to the same type of carbon atoms (carbon atoms of the hexagonal ring), they are not attached to opposite vertices of the hexagonal structure.
Therefore, the statement is False.
Final answer: False | ✓ | True | ✗ |
| InternVL2-8B | False.
In the given structure, H_a and H_d are not attached to opposite vertices of a hexagonal structure. Instead, they are both attached to the same carbon atom in the middle of the hexagonal ring. The hexagonal structure is formed by the six carbon atoms, and H_b, H_c, and H_e are attached to the other carbon atoms in the ring. H_a and H_d are part of the same carbon atom, not opposite vertices of the hexagonal structure. | ✓ | False | ✓ |
| InternVL2-26B | To determine if H_a and H_d are attached to opposite vertices of a hexagonal structure, we need to examine the given chemical structure.
1. Identify the hexagonal structure: The hexagonal structure in the given chemical diagram is the benzene ring, which is a six-membered ring with alternating double bonds. 2. Locate H_a and H_d: H_a is attached to one vertex of the benzene ring, and H_d is attached to the opposite vertex of the same ring. 3. Verify the attachment: Since H_a and H_d are attached to opposite vertices of the benzene ring, the statement is true.
Therefore, the final answer is True. | ✗ | True | ✗ |
| InternVL2-76B | To determine if H_a and H_d are attached to opposite vertices of a hexagonal structure, we need to analyze the positions of these hydrogen atoms in the given molecular structure.
1. Identify the hexagonal structure: The hexagonal structure in the molecule is the six-membered ring containing alternating double bonds (the benzene ring).
2. Locate H_a and H_d: - H_a is attached to the carbon atom that is part of the hexagonal structure and is adjacent to the nitrogen atom in the five-membered ring. - H_d is attached to the nitrogen atom in the five-membered ring that is directly connected to the hexagonal structure.
3. Determine their relative positions: - H_a is attached to a carbon atom on the hexagonal structure. - H_d is attached to a nitrogen atom that is part of the five-membered ring connected to the hexagonal structure.
Since H_d is not directly attached to the hexagonal structure but rather to an atom in the connected five-membered ring, H_a and H_d are not attached to opposite vertices of the hexagonal structure.
Final answer: False | ✓ | False | ✓ |
| Claude 3.5 Sonnet | To answer this question based solely on the visual information provided in the image:
I'll examine the structure and the positions of Ha and Hd.
The structure is not a simple hexagon, but rather a fused bicyclic system. Ha is attached to a carbon at the junction of the two rings, which forms one of the vertices of the overall structure. Hd is attached to a nitrogen atom that is part of one of the rings, but not at a vertex opposite to Ha.
The positions of Ha and Hd are not at opposite corners or vertices of the overall structure. They are on the same side of the molecule as depicted.
Based on this visual analysis:
False. Ha and Hd are not attached to opposite vertices of the structure shown in the image. | ✓ | False | ✓ |
| GPT-4o | H_a and H_d are attached to adjacent vertices of the hexagonal structure, not opposite ones. They are on the same side of the hexagon.
False. | ✓ | False | ✓ |
| Gemini 1.5 Pro | H_a and H_d are attached to the same carbon atom, which forms a vertex of the hexagonal structure. They are not attached to opposite vertices.
False. | ✓ | False | ✓ |

Table 34: Examples of dataset and model responses for `Chemistry-Shape-Single` (2)

| Image | Question | Gold Answer |
|---|---|---|
|  | H and O are attached to opposite vertices of the hexagonal structure. True or False? You must answer this question using visual information and do not use knowledge of chemistry unless explicitly instructed to do so. | False |

| Model | Answer w/ CoT | | Answer w/o CoT | |
|---|---|---|---|---|
| InternVL2-4B | The image shows a hexagonal ring structure with a double bond (indicated by the double line) between two of the vertices. The hydrogen (H) and oxygen (O) atoms are attached to the vertices of the hexagon, not necessarily opposite each other. Final answer: False. | ✓ | False | ✓ |
| InternVL2-8B | False. The hexagonal structure in the image is a benzene ring, which is a common aromatic ring in organic chemistry. In the given structure, the oxygen atom (O) is attached to one of the carbon atoms in the ring, and the hydrogen atom (H) is attached to another carbon atom. These two atoms are not located at opposite vertices of the hexagonal structure. Instead, they are adjacent to each other within the ring. Therefore, the statement that H and O are attached to opposite vertices of the hexagonal structure is false. | ✓ | False | ✓ |
| InternVL2-26B | To determine if H and O are attached to opposite vertices of the hexagonal structure, we need to examine the positions of H and O in the given diagram. In the diagram, the hexagonal structure is a benzene ring. The oxygen (O) is attached to one vertex of the hexagon, and the hydrogen (H) is attached to another vertex, which is directly opposite the oxygen. This means that H and O are indeed attached to opposite vertices of the hexagonal structure. Therefore, the statement is true. True. | ✗ | True | ✗ |
| InternVL2-76B | The hexagonal structure in the image represents a benzene ring. The hydrogen (H) and oxygen (O) atoms are attached to adjacent carbon atoms in the ring, not to opposite vertices. Therefore, the statement is false. Final answer: False | ✓ | False | ✓ |
| Claude 3.5 Sonnet | To answer this question based solely on the visual information provided in the image: I can see that the O (oxygen) atom is attached to one of the vertices of the hexagonal structure at the top. The H (hydrogen) atom is attached to a different vertex, which is not directly opposite to the O. The H appears to be attached to a vertex that is one position clockwise from the bottom vertex of the hexagon. For the O and H to be on opposite vertices, they would need to be at the top and bottom of the hexagon, or on opposite corners of the sides. This is not the case in the image. Therefore, based purely on the visual information: False. H and O are not attached to opposite vertices of the hexagonal structure. | ✓ | False | ✓ |
| GPT-4o | In the image, the hexagonal structure represents a cyclohexane ring. The oxygen (O) is attached to one vertex of the hexagon, and the hydrogen (H) is attached to a different vertex. However, they are not on opposite vertices of the hexagon. Final answer: False. | ✓ | False | ✓ |
| Gemini 1.5 Pro | The Oxygen (O) is attached to a vertex of the hexagon. The Hydrogen (H) is attached to a point along the edge of the hexagon below the Oxygen and adjacent to the isopropyl group (the two CH3's attached to a central carbon). Therefore, they are not at opposite vertices. False. | ✓ | False | ✓ |

Table 35: Examples of dataset and model responses for Chemistry-Shape-Single (3)

| Image | Question | Gold Answer |
|---|---|---|
| (a) (b) (c) (d) | List all figures that satisfy the following conditions from a, b, c, and d. You must answer this question using visual information and do not need to use your knowledge of chemistry unless explicitly instructed to do so. Conditions: Including a hexagonal structure. | a,b,c,d |

| Model | Answer w/ CoT | | Answer w/o CoT | |
|---|---|---|---|---|
| InternVL2-4B | To determine which figures satisfy the condition of including a hexagonal structure, we need to identify the figures that have a hexagon within their structure. - Figure (a) shows a hexagonal ring structure. - Figure (b) does not have a hexagonal structure. - Figure (c) does not have a hexagonal structure. - Figure (d) does not have a hexagonal structure. Therefore, the only figure that satisfies the condition of including a hexagonal structure is figure (a). Final answer: a | ✗ | a, b, c, d | ✓ |
| InternVL2-8B | The figures that satisfy the condition of including a hexagonal structure are: - (a) and (b) Explanation: - Figure (a) clearly shows a hexagonal structure. - Figure (b) also shows a hexagonal structure. - Figures (c) and (d) do not have a hexagonal structure. Therefore, the figures that satisfy the condition are (a) and (b). | ✗ | a, b, c, d | ✓ |
| InternVL2-26B | To determine which figures satisfy the condition of including a hexagonal structure, we need to identify the structures that have a hexagonal ring within them. - Figure (a) has a benzene ring, which is a hexagon. - Figure (b) has a benzene ring, which is a hexagon. - Figure (c) has a benzene ring, which is a hexagon. - Figure (d) has a benzene ring, which is a hexagon. All four figures (a, b, c, and d) have a hexagonal structure, specifically a benzene ring. Final answer: a, b, c, d | ✓ | a, b, c, d | ✓ |
| InternVL2-76B | To determine which figures satisfy the condition of including a hexagonal structure, we need to visually inspect each figure for the presence of a hexagon. - Figure (a): This figure shows a hexagonal structure with a double bond attached to one of the carbons. - Figure (b): This figure also shows a hexagonal structure with a double bond attached to one of the carbons. - Figure (c): This figure shows a hexagonal structure with two double bonds attached to two different carbons. - Figure (d): This figure shows a hexagonal structure with a double bond attached to one of the carbons. All figures (a, b, c, and d) include a hexagonal structure. Final answer: a, b, c, d | ✓ | a, b, c, d | ✓ |
| Claude 3.5 Sonnet | Let's examine each figure: (a) This structure has a hexagonal ring with an external double bond attached. (b) This structure has a hexagonal ring with an external double bond attached to the ring itself. (c) This structure has a hexagonal ring with an external methyl group attached. (d) This structure has a hexagonal ring with an internal double bond and an external methyl group. All of these structures (a, b, c, and d) satisfy the condition of including a hexagonal structure. Each figure shows a six-membered ring, which is the defining characteristic of a hexagonal structure in organic chemistry notation. Explanation: A hexagonal structure in this context refers to the six-sided ring that forms the base of each molecule depicted. This hexagonal ring is present in all four figures, regardless of the additional features like double bonds or methyl groups. Final answer: a, b, c, d | ✓ | a, b, c, d | ✓ |
| GPT-4o | All the figures (a, b, c, and d) include a hexagonal structure. Each structure has a six-membered ring, which is the hexagonal part. Final answer: a, b, c, d. | ✓ | a, b, c, d | ✓ |
| Gemini 1.5 Pro | All of the figures (a, b, c, and d) include a hexagonal structure as the base of their molecule. Each one has a six-sided ring. Final Answer: a, b, c, d | ✓ | a, b, c, d | ✓ |

Table 36: Examples of dataset and model responses for Chemistry-Shape-Multi (1)

| Image | Question | Gold Answer |
|---|---|---|
|  | List all figures that satisfy the following conditions from a, b, c, and d. You must answer this question using visual information and do not need to use your knowledge of chemistry unless explicitly instructed to do so. Conditions: Including a pentagonal structure. | a |

| Model | Answer w/ CoT | | Answer w/o CoT | |
|---|---|---|---|---|
| InternVL2-4B | To determine which figures satisfy the condition of including a pentagonal structure, we need to identify the figures that have a five-sided polygon (pentagon). - Figure (a) is a bicyclic compound with a five-membered ring, which is a pentagon. - Figure (b) is a benzene ring, which is a hexagon. - Figure (c) is a bicyclic compound with a five-membered ring, which is a pentagon. - Figure (d) is a square, which is a four-sided polygon. Based on this analysis, figures (a) and (c) satisfy the condition of including a pentagonal structure. Final answer: a, c | ✗ | (b), (c), (d) | ✗ |
| InternVL2-8B | To identify the figures that include a pentagonal structure, we need to examine each figure for the presence of a five-sided polygon. - **Figure (a)**: This structure contains a pentagonal ring. - **Figure (b)**: This structure contains a hexagonal ring. - **Figure (c)**: This structure contains a pentagonal ring. - **Figure (d)**: This structure contains a linear structure with a chlorine atom at each end. Based on the conditions provided, the figures that include a pentagonal structure are (a) and (c). **Final answer: (a, c)** | ✗ | a, b, c | ✗ |
| InternVL2-26B | To determine which figures satisfy the condition of including a pentagonal structure, we need to identify the structures that have a five-sided shape. - Figure (a) shows a pentagonal structure with two chlorine atoms attached to it. - Figure (b) is a hexagonal structure with three chlorine atoms attached to it. - Figure (c) is a triangular structure with three chlorine atoms attached to it. - Figure (d) is a square structure with one chlorine atom attached to it. The only figure that includes a pentagonal structure is (a). Final answer: (a) | ✓ | a | ✓ |
| InternVL2-76B | To determine which figures include a pentagonal structure, we need to visually inspect each figure for a five-sided polygon. - Figure (a) contains a five-sided ring, which is a pentagon. - Figure (b) contains a six-sided ring, which is a hexagon. - Figure (c) contains a three-sided ring, which is a triangle. - Figure (d) contains a four-sided ring, which is a square. Based on this analysis, the only figure that includes a pentagonal structure is (a). Final answer: (a) | ✓ | a | ✓ |
| Claude 3.5 Sonnet | Let's examine each figure: (a) This structure shows a pentagonal (five-sided) ring with two chlorine atoms attached. (b) This structure shows a hexagonal (six-sided) ring with two chlorine atoms attached. (c) This structure shows a triangular shape with four chlorine atoms attached. (d) This structure shows a square (four-sided) ring with two chlorine atoms attached. Reasoning: The question asks for figures that include a pentagonal structure. Looking at the shapes, only one of these figures contains a five-sided ring, which is the definition of a pentagonal structure. Final answer: a | ✓ | a | ✓ |
| GPT-4o | The figure that includes a pentagonal structure is (a). It has a five-sided ring, which is characteristic of a pentagon. Final answer: a | ✓ | a | ✓ |
| Gemini 1.5 Pro | The condition is including a pentagonal structure. Figure (a) contains a pentagon. Final Answer: The final answer is $\boxed{a}$ | ✓ | a | ✓ |

Table 37: Examples of dataset and model responses for `Chemistry-Shape-Multi` (2)

| Image | Question | Gold Answer |
|---|---|---|
|  | List all figures that satisfy the following conditions from a, b, c, and d. You must answer this question using visual information and do not need to use your knowledge of chemistry unless explicitly instructed to do so.
Conditions: Including a pentagonal structure. | a,d |

| Model | Answer w/ CoT | | Answer w/o CoT | |
|---|---|---|---|---|
| InternVL2-4B | To determine which figures satisfy the condition of including a pentagonal structure, we need to identify the structures that have a five-membered ring.
- Figure (a) shows a benzene ring, which is a hexagonal structure. - Figure (b) shows a benzene ring, which is a hexagonal structure. - Figure (c) shows a benzene ring, which is a hexagonal structure. - Figure (d) shows a benzene ring, which is a hexagonal structure.
None of the figures (a, b, c, or d) include a pentagonal structure.
Final answer: None of the figures (a, b, c, or d) satisfy the condition of including a pentagonal structure. | ✗ | (a), (b), (c), (d) | ✗ |
| InternVL2-8B | To identify the figures that include a pentagonal structure, we need to examine each figure for the presence of a five-sided polygon.
- **Figure (a)**: This structure is a hexagon, not a pentagon. - **Figure (b)**: This structure is a hexagon, not a pentagon. - **Figure (c)**: This structure is a hexagon, not a pentagon. - **Figure (d)**: This structure is a pentagon.
Therefore, the only figure that includes a pentagonal structure is **Figure (d)**.
Final answer: (d) | ✗ | a, c | ✗ |
| InternVL2-26B | To determine which figures satisfy the condition of including a pentagonal structure, we need to identify the structures that have a five-sided ring.
- Figure (a) shows a pentagonal structure with a methyl group attached to one of the vertices. - Figure (b) shows a pentagonal structure with a methyl group attached to one of the vertices. - Figure (c) shows a pentagonal structure with a hydroxyl group attached to one of the vertices. - Figure (d) shows a pentagonal structure with a chlorine atom attached to one of the vertices.
All four figures (a, b, c, and d) satisfy the condition of including a pentagonal structure.
Final answer: a, b, c, d | ✗ | a, b, c, d | ✗ |
| InternVL2-76B | To determine which figures satisfy the condition of including a pentagonal structure, we need to visually inspect each figure for the presence of a five-sided ring.
- Figure (a) contains a five-membered ring with one oxygen atom, which is a pentagonal structure. - Figure (b) also contains a five-membered ring with one oxygen atom, which is a pentagonal structure. - Figure (c) contains a five-membered ring with one oxygen atom and a hydroxyl group, which is a pentagonal structure. - Figure (d) contains a five-membered ring with one chlorine atom, which is a pentagonal structure.
All figures (a, b, c, and d) include a pentagonal structure.
Final answer: a, b, c, d | ✗ | a, b, c, d | ✗ |
| Claude 3.5 Sonnet | Let's examine each figure for a pentagonal structure:
(a) This molecule has a pentagonal (5-sided) ring structure. (b) This molecule has a 7-membered ring, not pentagonal. (c) This molecule also has a 7-membered ring, not pentagonal. (d) This molecule has a pentagonal (5-sided) ring structure.
Reasoning: A pentagonal structure is defined by having 5 sides or vertices. Only figures (a) and (d) contain a 5-membered ring, which satisfies the condition of including a pentagonal structure.
Final answer: a, d | ✓ | a, d | ✓ |
| GPT-4o | All the figures (a, b, c, and d) include a pentagonal structure. Each structure has a five-membered ring, which is a pentagon.
Final answer: a, b, c, d. | ✗ | a, b, c, d | ✗ |
| Gemini 1.5 Pro | The figures (a), (b), (c), and (d) all depict pentagonal structures as part of their molecular diagrams.
Final Answer: a, b, c, d | ✗ | a, b, c, d | ✗ |

Table 38: Examples of dataset and model responses for Chemistry-Shape-Multi (3)

| Image | Question | Gold Answer |
|---|---|---|
| 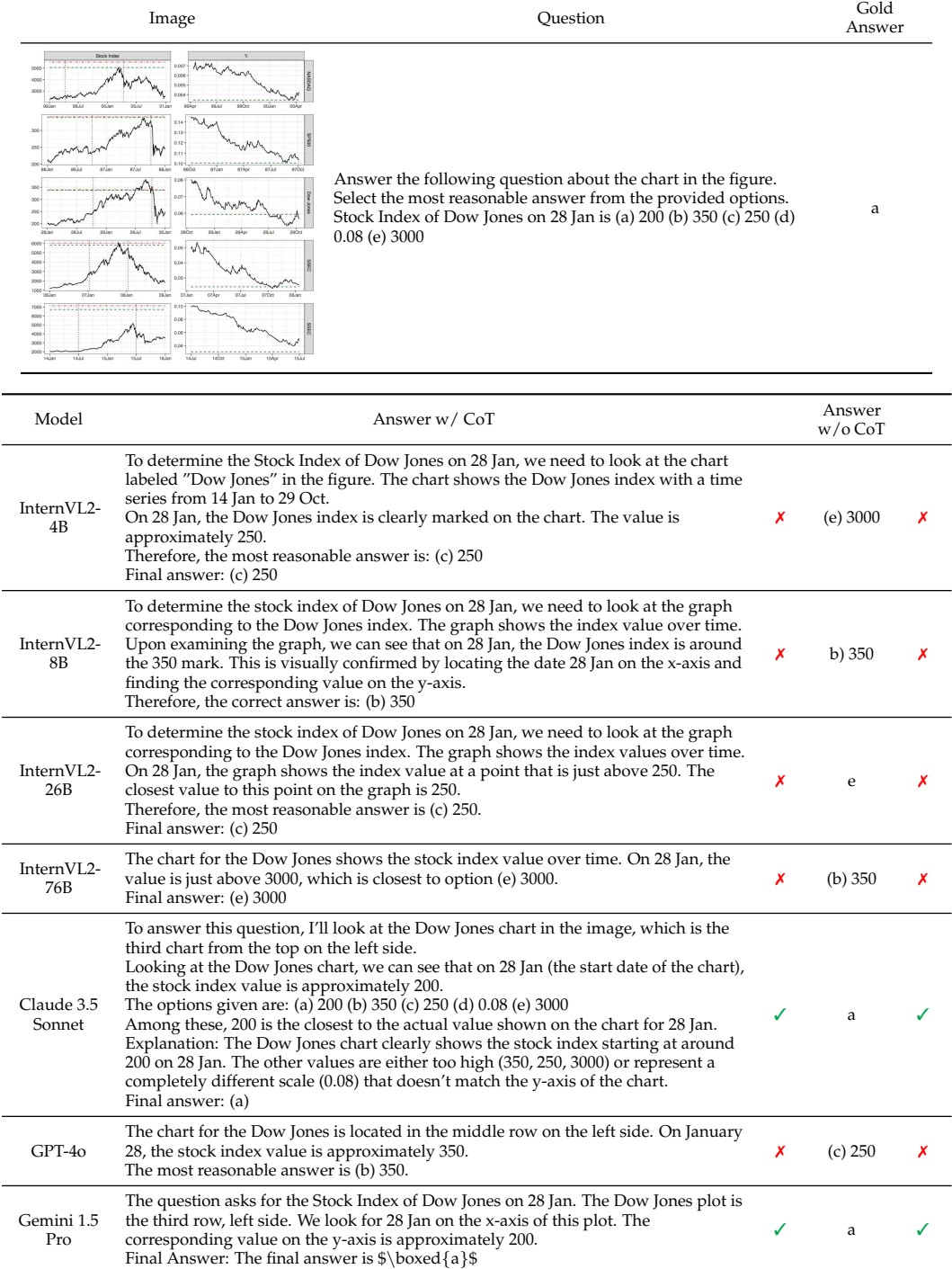 | Answer the following question about the chart in the figure. Select the most reasonable answer from the provided options. Stock Index of Dow Jones on 28 Jan is (a) 200 (b) 350 (c) 250 (d) 0.08 (e) 3000 | a |

| Model | Answer w/ CoT | | Answer w/o CoT | |
|---|---|---|---|---|
| InternVL2-4B | To determine the Stock Index of Dow Jones on 28 Jan, we need to look at the chart labeled "Dow Jones" in the figure. The chart shows the Dow Jones index with a time series from 14 Jan to 29 Oct. On 28 Jan, the Dow Jones index is clearly marked on the chart. The value is approximately 250. Therefore, the most reasonable answer is: (c) 250 Final answer: (c) 250 | ✗ | (e) 3000 | ✗ |
| InternVL2-8B | To determine the stock index of Dow Jones on 28 Jan, we need to look at the graph corresponding to the Dow Jones index. The graph shows the index value over time. Upon examining the graph, we can see that on 28 Jan, the Dow Jones index is around the 350 mark. This is visually confirmed by locating the date 28 Jan on the x-axis and finding the corresponding value on the y-axis. Therefore, the correct answer is: (b) 350 | ✗ | b) 350 | ✗ |
| InternVL2-26B | To determine the stock index of Dow Jones on 28 Jan, we need to look at the graph corresponding to the Dow Jones index. The graph shows the index values over time. On 28 Jan, the graph shows the index value at a point that is just above 250. The closest value to this point on the graph is 250. Therefore, the most reasonable answer is (c) 250. Final answer: (c) 250 | ✗ | e | ✗ |
| InternVL2-76B | The chart for the Dow Jones shows the stock index value over time. On 28 Jan, the value is just above 3000, which is closest to option (e) 3000. Final answer: (e) 3000 | ✗ | (b) 350 | ✗ |
| Claude 3.5 Sonnet | To answer this question, I'll look at the Dow Jones chart in the image, which is the third chart from the top on the left side. Looking at the Dow Jones chart, we can see that on 28 Jan (the start date of the chart), the stock index value is approximately 200. The options given are: (a) 200 (b) 350 (c) 250 (d) 0.08 (e) 3000 Among these, 200 is the closest to the actual value shown on the chart for 28 Jan. Explanation: The Dow Jones chart clearly shows the stock index starting at around 200 on 28 Jan. The other values are either too high (350, 250, 3000) or represent a completely different scale (0.08) that doesn't match the y-axis of the chart. Final answer: (a) | ✓ | a | ✓ |
| GPT-4o | The chart for the Dow Jones is located in the middle row on the left side. On January 28, the stock index value is approximately 350. The most reasonable answer is (b) 350. | ✗ | (c) 250 | ✗ |
| Gemini 1.5 Pro | The question asks for the Stock Index of Dow Jones on 28 Jan. The Dow Jones plot is the third row, left side. We look for 28 Jan on the x-axis of this plot. The corresponding value on the y-axis is approximately 200. Final Answer: The final answer is $\boxed{a}$ | ✓ | a | ✓ |

Table 39: Examples of dataset and model responses for Charts-Extraction (1)

| Image | Question | Gold Answer |
|---|---|---|
| 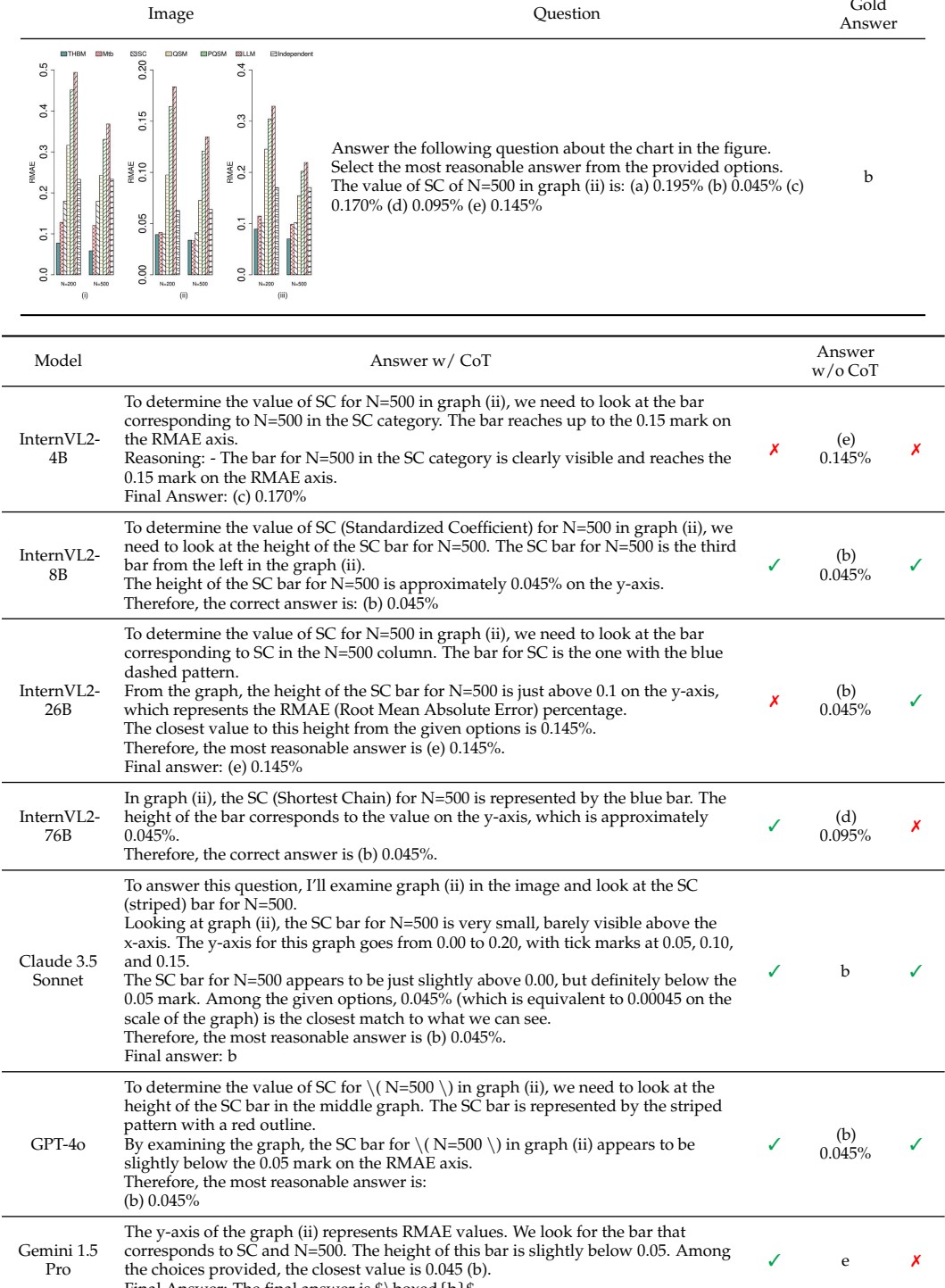 | Answer the following question about the chart in the figure. Select the most reasonable answer from the provided options. The value of SC of N=500 in graph (ii) is: (a) 0.195% (b) 0.045% (c) 0.170% (d) 0.095% (e) 0.145% | b |

| Model | Answer w/ CoT | | Answer w/o CoT | |
|---|---|---|---|---|
| InternVL2-4B | To determine the value of SC for N=500 in graph (ii), we need to look at the bar corresponding to N=500 in the SC category. The bar reaches up to the 0.15 mark on the RMAE axis. Reasoning: - The bar for N=500 in the SC category is clearly visible and reaches the 0.15 mark on the RMAE axis. Final Answer: (c) 0.170% | ✗ | (e) 0.145% | ✗ |
| InternVL2-8B | To determine the value of SC (Standardized Coefficient) for N=500 in graph (ii), we need to look at the height of the SC bar for N=500. The SC bar for N=500 is the third bar from the left in the graph (ii). The height of the SC bar for N=500 is approximately 0.045% on the y-axis. Therefore, the correct answer is: (b) 0.045% | ✓ | (b) 0.045% | ✓ |
| InternVL2-26B | To determine the value of SC for N=500 in graph (ii), we need to look at the bar corresponding to SC in the N=500 column. The bar for SC is the one with the blue dashed pattern. From the graph, the height of the SC bar for N=500 is just above 0.1 on the y-axis, which represents the RMAE (Root Mean Absolute Error) percentage. The closest value to this height from the given options is 0.145%. Therefore, the most reasonable answer is (e) 0.145%. Final answer: (e) 0.145% | ✗ | (b) 0.045% | ✓ |
| InternVL2-76B | In graph (ii), the SC (Shortest Chain) for N=500 is represented by the blue bar. The height of the bar corresponds to the value on the y-axis, which is approximately 0.045%. Therefore, the correct answer is (b) 0.045%. | ✓ | (d) 0.095% | ✗ |
| Claude 3.5 Sonnet | To answer this question, I'll examine graph (ii) in the image and look at the SC (striped) bar for N=500. Looking at graph (ii), the SC bar for N=500 is very small, barely visible above the x-axis. The y-axis for this graph goes from 0.00 to 0.20, with tick marks at 0.05, 0.10, and 0.15. The SC bar for N=500 appears to be just slightly above 0.00, but definitely below the 0.05 mark. Among the given options, 0.045% (which is equivalent to 0.00045 on the scale of the graph) is the closest match to what we can see. Therefore, the most reasonable answer is (b) 0.045%. Final answer: b | ✓ | b | ✓ |
| GPT-4o | To determine the value of SC for \( N=500 \) in graph (ii), we need to look at the height of the SC bar in the middle graph. The SC bar is represented by the striped pattern with a red outline. By examining the graph, the SC bar for \( N=500 \) in graph (ii) appears to be slightly below the 0.05 mark on the RMAE axis. Therefore, the most reasonable answer is: (b) 0.045% | ✓ | (b) 0.045% | ✓ |
| Gemini 1.5 Pro | The y-axis of the graph (ii) represents RMAE values. We look for the bar that corresponds to SC and N=500. The height of this bar is slightly below 0.05. Among the choices provided, the closest value is 0.045 (b). Final Answer: The final answer is $\boxed{b}$ | ✓ | e | ✗ |

Table 40: Examples of dataset and model responses for `Charts-Extraction` (2)

| Image | Question | Gold Answer |
|---|---|---|
|  | Answer the following question about the chart in the figure. Select the most reasonable answer from the provided options. PPL of NEXUS at Perturbation = 0.08 is (a) $5\times10^2$ (b) $1\times10^2$ (c) $4\times10^2$ (d) $3\times10^2$ (e) $2\times10^2$ | d |

| Model | Answer w/ CoT | | Answer w/o CoT | |
|---|---|---|---|---|
| InternVL2-4B | To determine the PPL (Perplexity) of NEXUS at a perturbation of 0.08, we need to locate the point on the graph where the perturbation value is 0.08 and then find the corresponding PPL value on the y-axis. 1. Locate the perturbation value of 0.08 on the x-axis. 2. Draw a vertical line from the perturbation value of 0.08 up to the NEXUS line. 3. Draw a horizontal line from the point where the vertical line intersects the NEXUS line to the y-axis. From the graph, the PPL value at a perturbation of 0.08 for NEXUS is approximately $2\times10^2$. Final answer: (e) $2\times10^2$ | ✗ | e) $2\times10^2$ | ✗ |
| InternVL2-8B | To determine the PPL (Posterior Predictive Likelihood) of NEXUS at Perturbation = 0.08, we need to locate the point on the graph where the perturbation value is 0.08 and identify the corresponding PPL value on the y-axis. 1. Locate the perturbation value of 0.08 on the x-axis. 2. Observe the line corresponding to NEXUS (blue line with diamond markers). 3. Find the y-value (PPL) at the point where the perturbation is 0.08. From the graph, at Perturbation = 0.08, the PPL for NEXUS is approximately $3\times10^2$. Therefore, the correct answer is (d) $3\times10^2$. | ✓ | b | ✗ |
| InternVL2-26B | To determine the PPL of NEXUS at Perturbation = 0.08, we need to locate the point on the graph where the perturbation is 0.08 and find the corresponding PPL value for NEXUS. 1. Locate the point on the x-axis where Perturbation = 0.08. 2. Follow the vertical line up from this point to the graph lines. 3. Identify the line corresponding to NEXUS (cyan diamond markers). 4. The PPL value at this point is approximately $3\times10^2$. Therefore, the PPL of NEXUS at Perturbation = 0.08 is $3\times10^2$. Final answer: (d) $3\times10^2$ | ✓ | (d) $3\times10^2$ | ✓ |
| InternVL2-76B | At Perturbation = 0.08, the PPL of NEXUS is indicated by the cyan line with diamond markers. The value on the graph at this perturbation level is approximately $3\times10^2$. Final answer: (d) $3\times10^2$ | ✓ | (d) $3\times10^2$ | ✓ |
| Claude 3.5 Sonnet | To answer this question, I'll look at the line for NEXUS on the graph at the perturbation value of 0.08 on the x-axis. Following the NEXUS line (light blue with diamond markers) to the 0.08 mark on the x-axis, I can see that the corresponding y-axis value is just above the $3\times10^2$ mark. The PPL value for NEXUS at 0.08 perturbation appears to be slightly higher than $3\times10^2$, but definitely lower than $4\times10^2$. The closest option provided is $3\times10^2$. Therefore, the most reasonable answer is (d) $3\times10^2$. Final answer: d | ✓ | d | ✓ |
| GPT-4o | To determine the PPL of NEXUS at Perturbation = 0.08, we need to look at the cyan diamond line on the graph. At Perturbation = 0.08, the PPL value for NEXUS is approximately $3\times10^2$. Therefore, the most reasonable answer is: (d) $3\times10^2$ | ✓ | d | ✓ |
| Gemini 1.5 Pro | The NEXUS line has a diamond marker. At Perturbation = 0.08, the NEXUS line corresponds to a PPL value of just above $3\times10^2$. Final Answer: (d) | ✓ | d | ✓ |

Table 41: Examples of dataset and model responses for Charts-Extraction (3)

| Image | Question | Gold Answer |
|---|---|---|
| 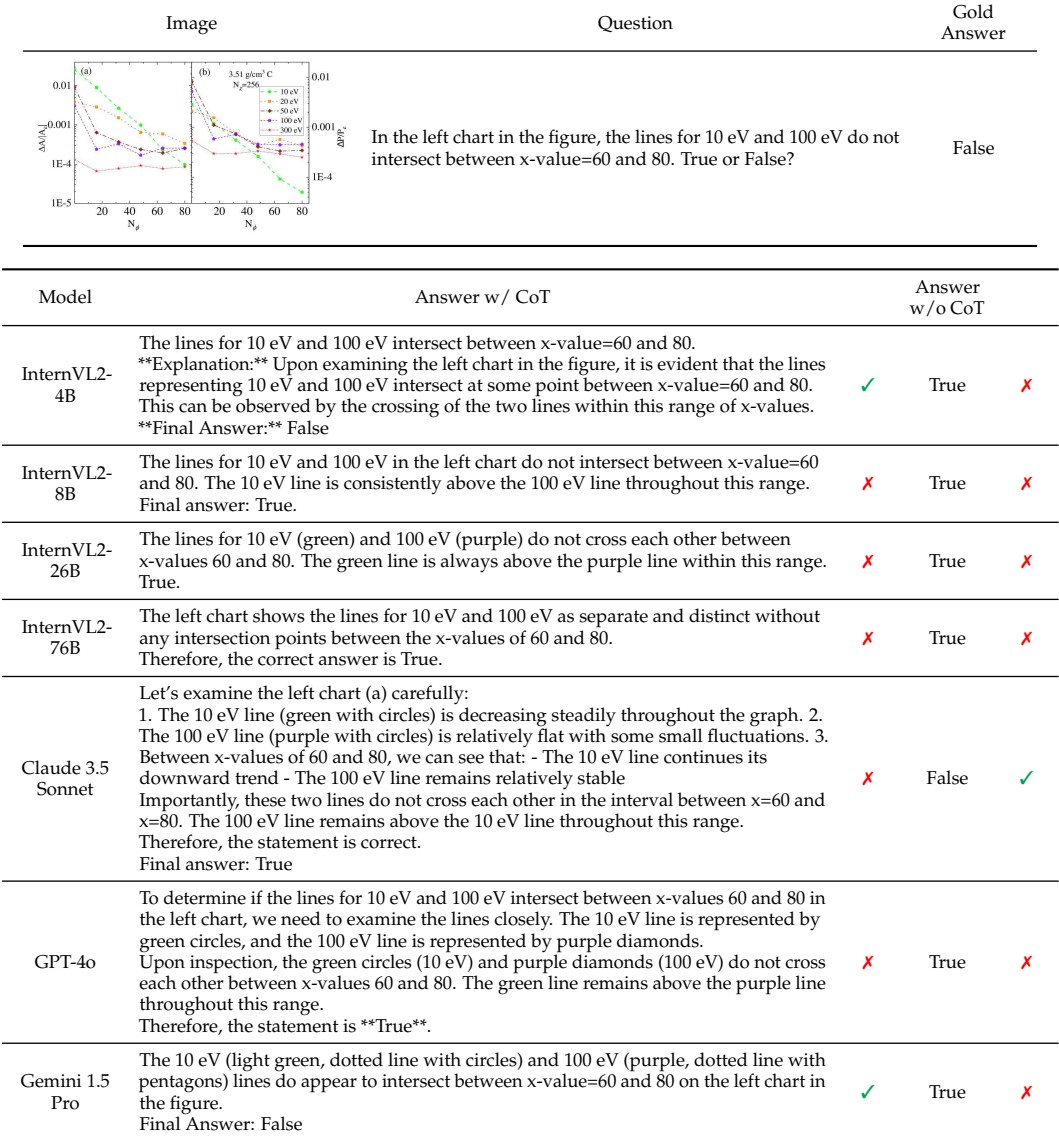 | In the left chart in the figure, the lines for 10 eV and 100 eV do not intersect between x-value=60 and 80. True or False? | False |

| Model | Answer w/ CoT | | Answer w/o CoT | |
|---|---|---|---|---|
| InternVL2-4B | The lines for 10 eV and 100 eV intersect between x-value=60 and 80. **Explanation:** Upon examining the left chart in the figure, it is evident that the lines representing 10 eV and 100 eV intersect at some point between x-value=60 and 80. This can be observed by the crossing of the two lines within this range of x-values. **Final Answer:** False | ✓ | True | ✗ |
| InternVL2-8B | The lines for 10 eV and 100 eV in the left chart do not intersect between x-value=60 and 80. The 10 eV line is consistently above the 100 eV line throughout this range. Final answer: True. | ✗ | True | ✗ |
| InternVL2-26B | The lines for 10 eV (green) and 100 eV (purple) do not cross each other between x-values 60 and 80. The green line is always above the purple line within this range. True. | ✗ | True | ✗ |
| InternVL2-76B | The left chart shows the lines for 10 eV and 100 eV as separate and distinct without any intersection points between the x-values of 60 and 80. Therefore, the correct answer is True. | ✗ | True | ✗ |
| Claude 3.5 Sonnet | Let's examine the left chart (a) carefully: 1. The 10 eV line (green with circles) is decreasing steadily throughout the graph. 2. The 100 eV line (purple with circles) is relatively flat with some small fluctuations. 3. Between x-values of 60 and 80, we can see that: - The 10 eV line continues its downward trend - The 100 eV line remains relatively stable Importantly, these two lines do not cross each other in the interval between x=60 and x=80. The 100 eV line remains above the 10 eV line throughout this range. Therefore, the statement is correct. Final answer: True | ✗ | False | ✓ |
| GPT-4o | To determine if the lines for 10 eV and 100 eV intersect between x-values 60 and 80 in the left chart, we need to examine the lines closely. The 10 eV line is represented by green circles, and the 100 eV line is represented by purple diamonds. Upon inspection, the green circles (10 eV) and purple diamonds (100 eV) do not cross each other between x-values 60 and 80. The green line remains above the purple line throughout this range. Therefore, the statement is **True**. | ✗ | True | ✗ |
| Gemini 1.5 Pro | The 10 eV (light green, dotted line with circles) and 100 eV (purple, dotted line with pentagons) lines do appear to intersect between x-value=60 and 80 on the left chart in the figure. Final Answer: False | ✓ | True | ✗ |

Table 42: Examples of dataset and model responses for Charts-Intersection (1)

| Image | Question | Gold Answer |
|---|---|---|
| 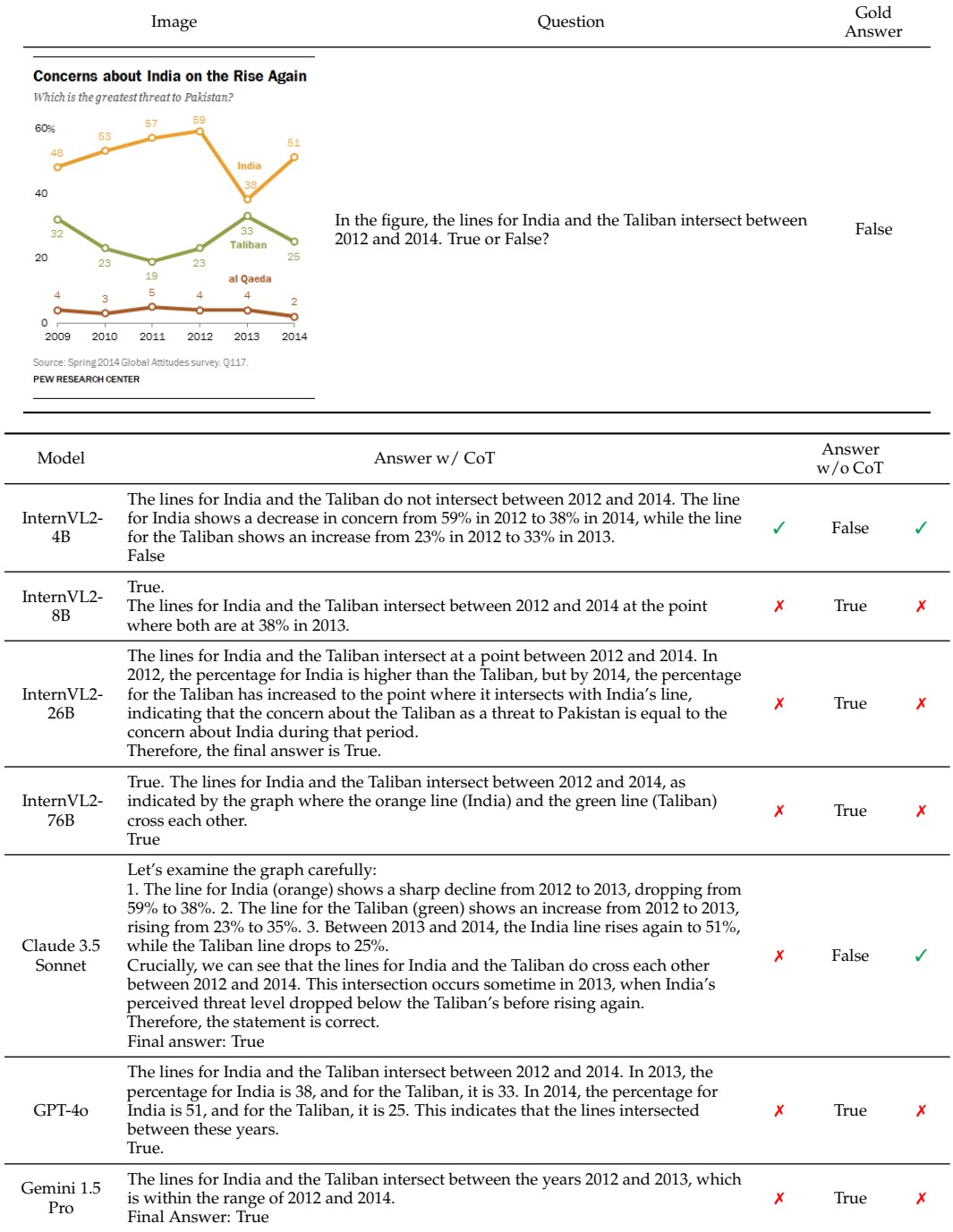 | In the figure, the lines for India and the Taliban intersect between 2012 and 2014. True or False? | False |

| Model | Answer w/ CoT | | Answer w/o CoT | |
|---|---|---|---|---|
| InternVL2-4B | The lines for India and the Taliban do not intersect between 2012 and 2014. The line for India shows a decrease in concern from 59% in 2012 to 38% in 2014, while the line for the Taliban shows an increase from 23% in 2012 to 33% in 2013. False | ✓ | False | ✓ |
| InternVL2-8B | True. The lines for India and the Taliban intersect between 2012 and 2014 at the point where both are at 38% in 2013. | ✗ | True | ✗ |
| InternVL2-26B | The lines for India and the Taliban intersect at a point between 2012 and 2014. In 2012, the percentage for India is higher than the Taliban, but by 2014, the percentage for the Taliban has increased to the point where it intersects with India's line, indicating that the concern about the Taliban as a threat to Pakistan is equal to the concern about India during that period. Therefore, the final answer is True. | ✗ | True | ✗ |
| InternVL2-76B | True. The lines for India and the Taliban intersect between 2012 and 2014, as indicated by the graph where the orange line (India) and the green line (Taliban) cross each other. True | ✗ | True | ✗ |
| Claude 3.5 Sonnet | Let's examine the graph carefully: 1. The line for India (orange) shows a sharp decline from 2012 to 2013, dropping from 59% to 38%. 2. The line for the Taliban (green) shows an increase from 2012 to 2013, rising from 23% to 35%. 3. Between 2013 and 2014, the India line rises again to 51%, while the Taliban line drops to 25%. Crucially, we can see that the lines for India and the Taliban do cross each other between 2012 and 2014. This intersection occurs sometime in 2013, when India's perceived threat level dropped below the Taliban's before rising again. Therefore, the statement is correct. Final answer: True | ✗ | False | ✓ |
| GPT-4o | The lines for India and the Taliban intersect between 2012 and 2014. In 2013, the percentage for India is 38, and for the Taliban, it is 33. In 2014, the percentage for India is 51, and for the Taliban, it is 25. This indicates that the lines intersected between these years. True. | ✗ | True | ✗ |
| Gemini 1.5 Pro | The lines for India and the Taliban intersect between the years 2012 and 2013, which is within the range of 2012 and 2014. Final Answer: True | ✗ | True | ✗ |

Table 43: Examples of dataset and model responses for Charts-Intersection (2)

| Image | Question | Gold Answer |
|---|---|---|
| 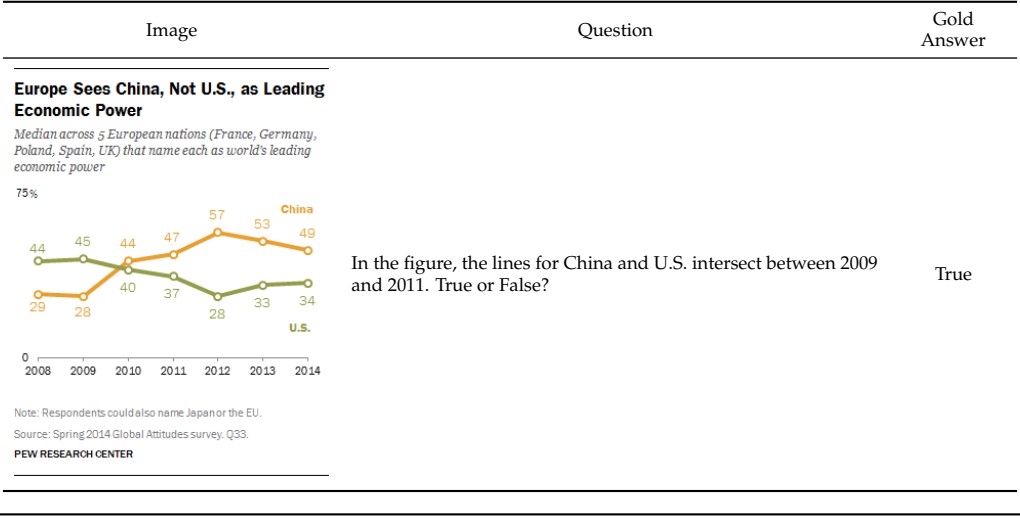 | In the figure, the lines for China and U.S. intersect between 2009 and 2011. True or False? | True |

| Model | Answer w/ CoT | | Answer w/o CoT | |
|---|---|---|---|---|
| InternVL2-4B | The lines for China and the U.S. do not intersect between 2009 and 2011. The line representing China shows a steady increase in the percentage of respondents who see China as the leading economic power during this period, while the line for the U.S. shows a decrease. Final answer: False. | ✗ | False | ✗ |
| InternVL2-8B | False. The lines for China and the U.S. do not intersect between 2009 and 2011. The line for China starts at 29 in 2008, increases to 40 in 2010, and then rises to 57 in 2012. The line for the U.S. starts at 44 in 2008, decreases to 37 in 2011, and then increases to 33 in 2012. The two lines do not cross each other during this period. | ✗ | False | ✗ |
| InternVL2-26B | The lines for China and U.S. intersect at a point between 2009 and 2011, as can be seen where the green line (U.S.) crosses the orange line (China) on the graph. True | ✓ | False | ✗ |
| InternVL2-76B | True. The lines for China and U.S. intersect between the years 2009 and 2011, indicating a crossover point where the perception of leading economic power shifted from the U.S. to China according to the median across the 5 European nations surveyed. The graph shows that in 2009, the U.S. was seen as the leading economic power by a larger percentage than China. However, by 2011, China surpassed the U.S. in this perception, as indicated by the point where the two lines cross. | ✓ | True | ✓ |
| Claude 3.5 Sonnet | Let's examine the graph closely: 1. In 2009, the U.S. line is at 28% while China's line is at 33%. 2. In 2010, the U.S. line rises to 37% while China's line rises to 40%. 3. In 2011, the U.S. line continues to rise to 47% while China's line rises to 44%. We can see that between 2009 and 2011, the lines for China and the U.S. do indeed cross each other. The intersection occurs somewhere between 2010 and 2011, as China's percentage starts higher but is overtaken by the U.S. during this period. Therefore, the statement is True. | ✓ | True | ✓ |
| GPT-4o | The lines for China and the U.S. intersect between 2009 and 2011. In 2009, both are at 44%, and after that, China's line rises above the U.S. line. Therefore, the statement is true. True. | ✓ | True | ✓ |
| Gemini 1.5 Pro | The lines intersect between 2009 and 2010, not between 2009 and 2011. Final Answer: False | ✗ | True | ✓ |

Table 44: Examples of dataset and model responses for Charts-Intersection (3)

| Image | Question | Gold Answer |
|-------|----------|-------------|
| 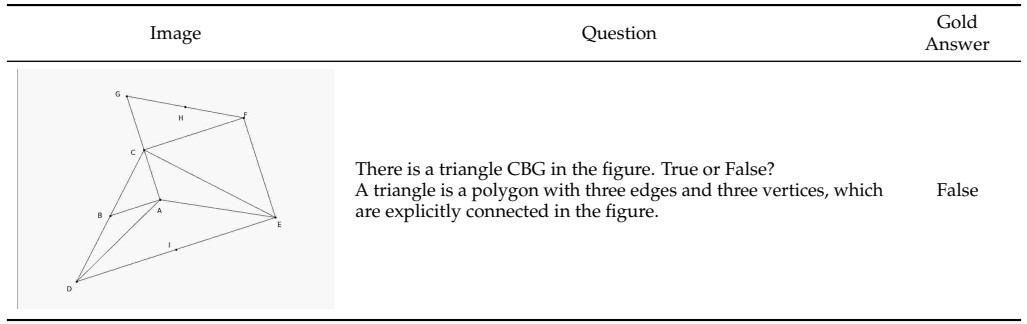 | There is a triangle CBG in the figure. True or False? A triangle is a polygon with three edges and three vertices, which are explicitly connected in the figure. | False |

| Model | Answer w/ CoT | | Answer w/o CoT | |
|-------|---------------|---|----------------|---|
| InternVL2-4B | – | – | True | ✗ |
| InternVL2-8B | – | – | True | ✗ |
| InternVL2-26B | – | – | True | ✗ |
| InternVL2-76B | – | – | True | ✗ |
| Claude 3.5 Sonnet | – | – | True | ✗ |
| GPT-4o | – | – | True | ✗ |
| Gemini 1.5 Pro | – | – | True | ✗ |

Table 45: Examples of dataset and model responses for `Syntheticgeometry-Triangle` (1)

| Image | Question | Gold Answer |
|-------|----------|-------------|
| 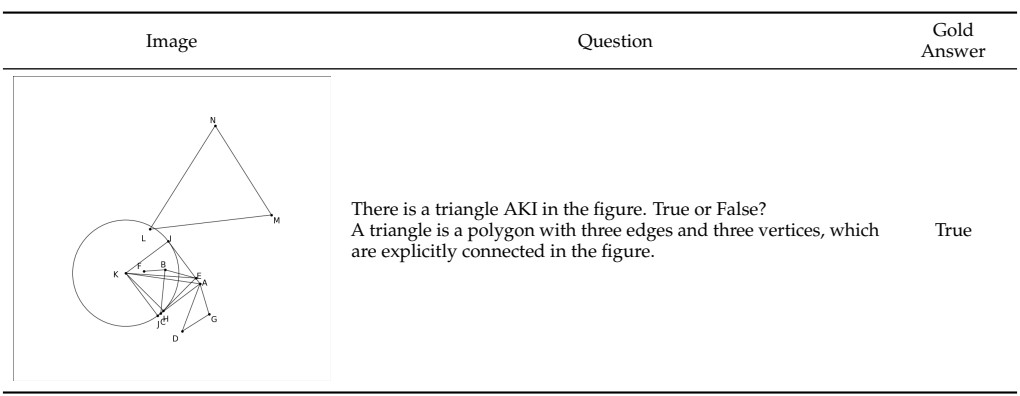 | There is a triangle AKI in the figure. True or False? A triangle is a polygon with three edges and three vertices, which are explicitly connected in the figure. | True |

| Model | Answer w/ CoT | | Answer w/o CoT | |
|-------|---------------|---|----------------|---|
| InternVL2-4B | – | – | True | ✓ |
| InternVL2-8B | – | – | False | ✗ |
| InternVL2-26B | – | – | True | ✓ |
| InternVL2-76B | – | – | True | ✓ |
| Claude 3.5 Sonnet | – | – | False | ✗ |
| GPT-4o | – | – | True | ✓ |
| Gemini 1.5 Pro | – | – | True | ✓ |

Table 46: Examples of dataset and model responses for Syntheticgeometry-Triangle (2)

| Image | Question | Gold Answer |
|-------|----------|-------------|
| 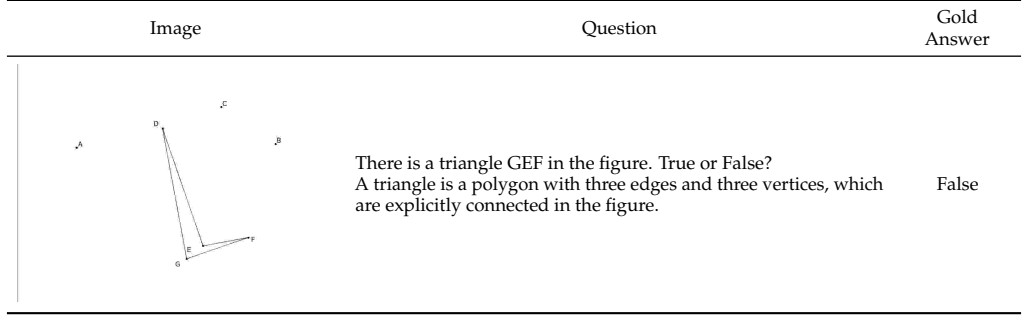 | There is a triangle GEF in the figure. True or False?
A triangle is a polygon with three edges and three vertices, which are explicitly connected in the figure. | False |

| Model | Answer w/ CoT | | Answer w/o CoT | |
|-------|--------------|---|----------------|---|
| InternVL2-4B | – | – | True | ✗ |
| InternVL2-8B | – | – | True | ✗ |
| InternVL2-26B | – | – | True | ✗ |
| InternVL2-76B | – | – | True | ✗ |
| Claude 3.5 Sonnet | – | – | True | ✗ |
| GPT-4o | – | – | False | ✓ |
| Gemini 1.5 Pro | – | – | True | ✗ |

Table 47: Examples of dataset and model responses for `Syntheticgeometry-Triangle` (3)

| Image | Question | Gold Answer |
|---|---|---|
| 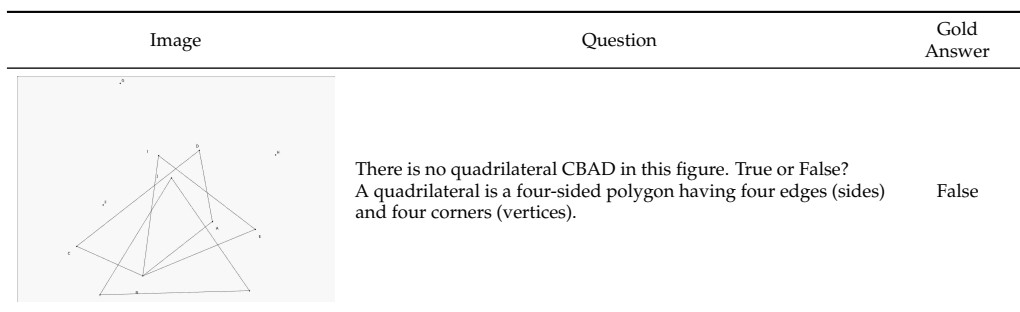 | There is no quadrilateral CBAD in this figure. True or False? A quadrilateral is a four-sided polygon having four edges (sides) and four corners (vertices). | False |

| Model | Answer w/ CoT | | Answer w/o CoT | |
|---|---|---|---|---|
| InternVL2-4B | – | – | True | ✗ |
| InternVL2-8B | – | – | False | ✓ |
| InternVL2-26B | – | – | True | ✗ |
| InternVL2-76B | – | – | False | ✓ |
| Claude 3.5 Sonnet | – | – | False | ✓ |
| GPT-4o | – | – | True | ✗ |
| Gemini 1.5 Pro | – | – | False | ✓ |

Table 48: Examples of dataset and model responses for Syntheticgeometry-Quadrilateral (1)

| Image | Question | Gold Answer |
|-------|----------|-------------|
| 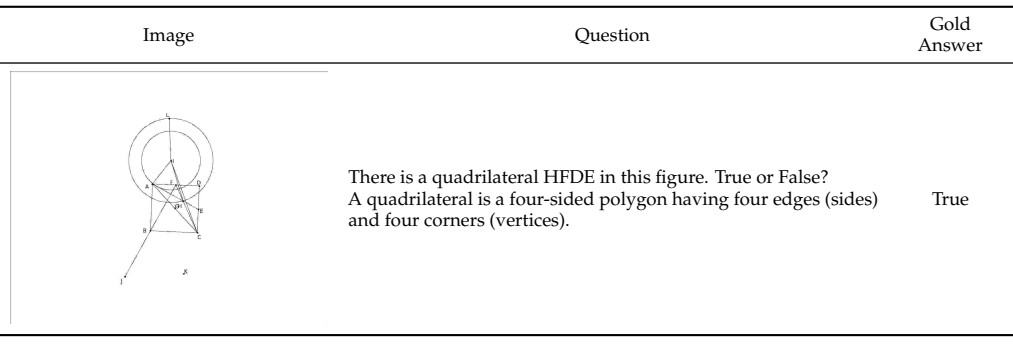 | There is a quadrilateral HFDE in this figure. True or False? A quadrilateral is a four-sided polygon having four edges (sides) and four corners (vertices). | True |

| Model | Answer w/ CoT | Answer w/o CoT | |
|-------|---------------|----------------|---|
| InternVL2-4B | – | – | False ✗ |
| InternVL2-8B | – | – | True ✓ |
| InternVL2-26B | – | – | True ✓ |
| InternVL2-76B | – | – | True ✓ |
| Claude 3.5 Sonnet | – | – | True ✓ |
| GPT-4o | – | – | True ✓ |
| Gemini 1.5 Pro | – | – | False ✗ |

Table 49: Examples of dataset and model responses for Syntheticgeometry-Quadrilateral (2)

| Image | Question | Gold Answer |
|---|---|---|
| 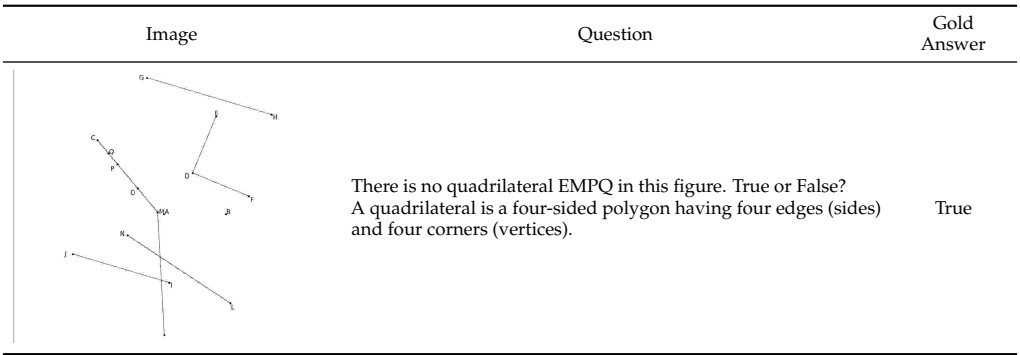 | There is no quadrilateral EMPQ in this figure. True or False? A quadrilateral is a four-sided polygon having four edges (sides) and four corners (vertices). | True |

| Model | Answer w/ CoT | | Answer w/o CoT | |
|---|---|---|---|---|
| InternVL2-4B | – | – | True | ✓ |
| InternVL2-8B | – | – | False | ✗ |
| InternVL2-26B | – | – | True | ✓ |
| InternVL2-76B | – | – | False | ✗ |
| Claude 3.5 Sonnet | – | – | False | ✗ |
| GPT-4o | – | – | True | ✓ |
| Gemini 1.5 Pro | – | – | False | ✗ |

Table 50: Examples of dataset and model responses for `Syntheticgeometry-Quadrilateral` (3)

| Image | Question | Gold Answer |
|---|---|---|
| 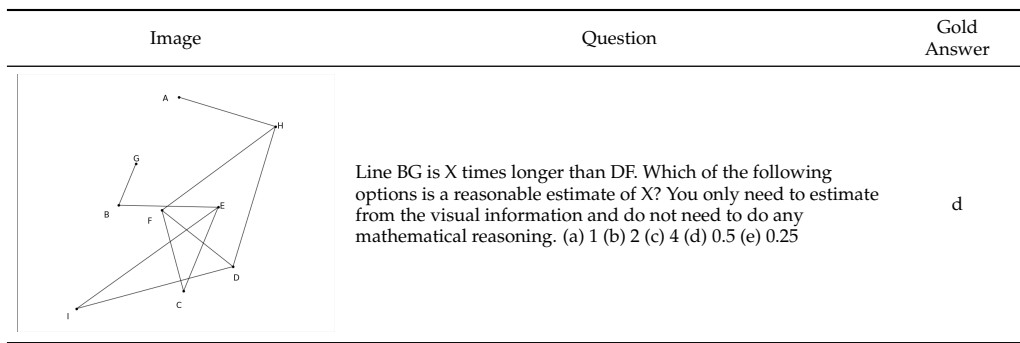 | Line BG is X times longer than DF. Which of the following options is a reasonable estimate of X? You only need to estimate from the visual information and do not need to do any mathematical reasoning. (a) 1 (b) 2 (c) 4 (d) 0.5 (e) 0.25 | d |

| Model | Answer w/ CoT | | Answer w/o CoT | |
|---|---|---|---|---|
| InternVL2-4B | – | – | b | ✗ |
| InternVL2-8B | – | – | b) 2 | ✗ |
| InternVL2-26B | – | – | b | ✗ |
| InternVL2-76B | – | – | (c) 4 | ✗ |
| Claude 3.5 Sonnet | – | – | b | ✗ |
| GPT-4o | – | – | (b) | ✗ |
| Gemini 1.5 Pro | – | – | b | ✗ |

Table 51: Examples of dataset and model responses for Syntheticgeometry-Length (1)

| Image | Question | Gold Answer |
|---|---|---|
| 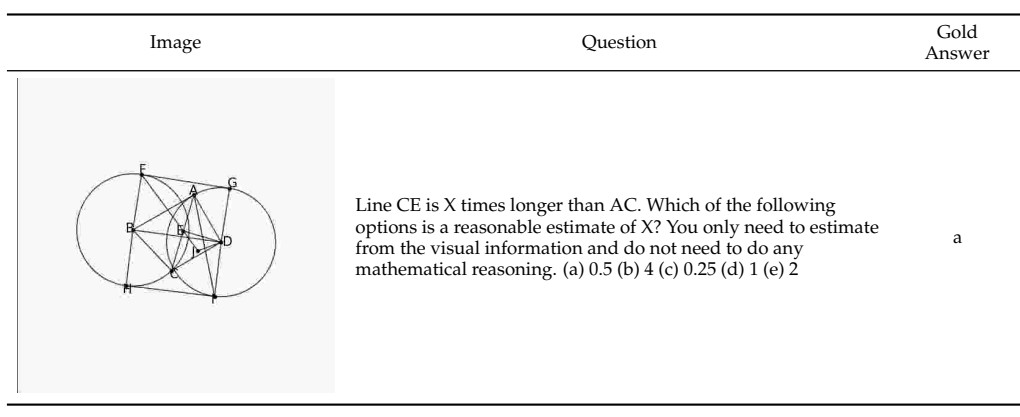 | Line CE is X times longer than AC. Which of the following options is a reasonable estimate of X? You only need to estimate from the visual information and do not need to do any mathematical reasoning. (a) 0.5 (b) 4 (c) 0.25 (d) 1 (e) 2 | a |

| Model | Answer w/ CoT | | Answer w/o CoT | |
|---|---|---|---|---|
| InternVL2-4B | – | – | e | ✗ |
| InternVL2-8B | – | – | b | ✗ |
| InternVL2-26B | – | – | (b) 4 | ✗ |
| InternVL2-76B | – | – | (e) 2 | ✗ |
| Claude 3.5 Sonnet | – | – | e | ✗ |
| GPT-4o | – | – | (e) 2 | ✗ |
| Gemini 1.5 Pro | – | – | e | ✗ |

Table 52: Examples of dataset and model responses for Syntheticgeometry-Length (2)

| Image | Question | Gold Answer |
|---|---|---|
| 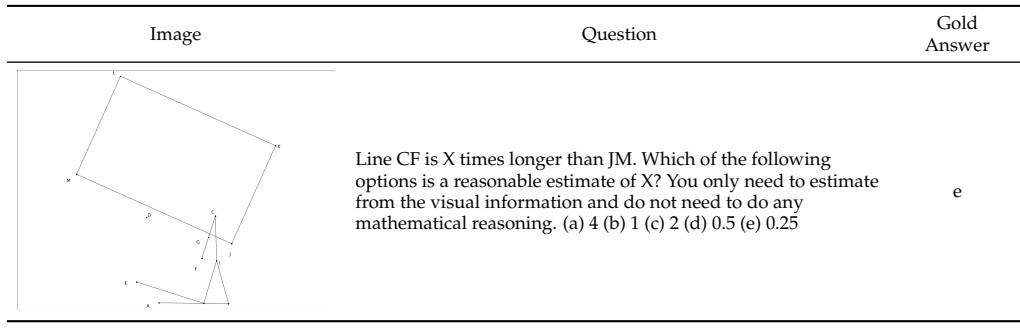 | Line CF is X times longer than JM. Which of the following options is a reasonable estimate of X? You only need to estimate from the visual information and do not need to do any mathematical reasoning. (a) 4 (b) 1 (c) 2 (d) 0.5 (e) 0.25 | e |

| Model | Answer w/ CoT | | Answer w/o CoT | |
|---|---|---|---|---|
| InternVL2-4B | – | – | a | ✗ |
| InternVL2-8B | – | – | a) 4 | ✗ |
| InternVL2-26B | – | – | (a) 4 | ✗ |
| InternVL2-76B | – | – | (c) 2 | ✗ |
| Claude 3.5 Sonnet | – | – | c | ✗ |
| GPT-4o | – | – | (c) 2 | ✗ |
| Gemini 1.5 Pro | – | – | c | ✗ |

Table 53: Examples of dataset and model responses for Syntheticgeometry-Length (3)

| Image | Question | Gold Answer |
|---|---|---|
| 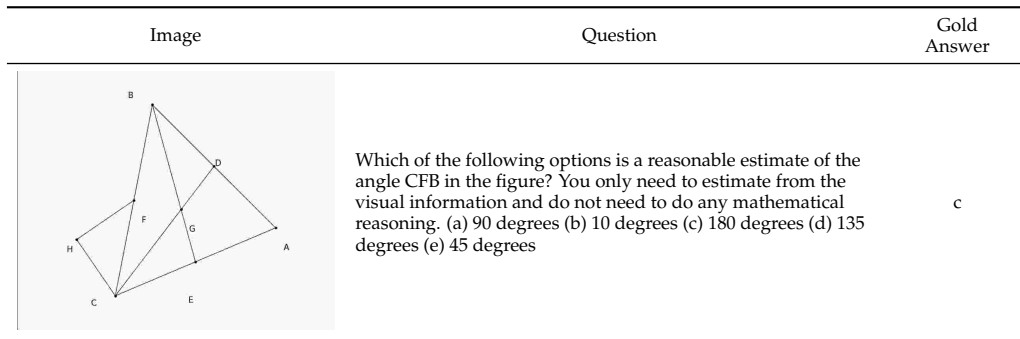 | Which of the following options is a reasonable estimate of the angle CFB in the figure? You only need to estimate from the visual information and do not need to do any mathematical reasoning. (a) 90 degrees (b) 10 degrees (c) 180 degrees (d) 135 degrees (e) 45 degrees | c |

| Model | Answer w/ CoT | | Answer w/o CoT | |
|---|---|---|---|---|
| InternVL2-4B | – | – | e | ✗ |
| InternVL2-8B | – | – | a) 90 degrees | ✗ |
| InternVL2-26B | – | – | (a) 90 degrees | ✗ |
| InternVL2-76B | – | – | (e) 45 degrees | ✗ |
| Claude 3.5 Sonnet | – | – | e | ✗ |
| GPT-4o | – | – | e | ✗ |
| Gemini 1.5 Pro | – | – | e | ✗ |

Table 54: Examples of dataset and model responses for `Syntheticgeometry-Angle` (1)

| Image | Question | Gold Answer |
|---|---|---|
| 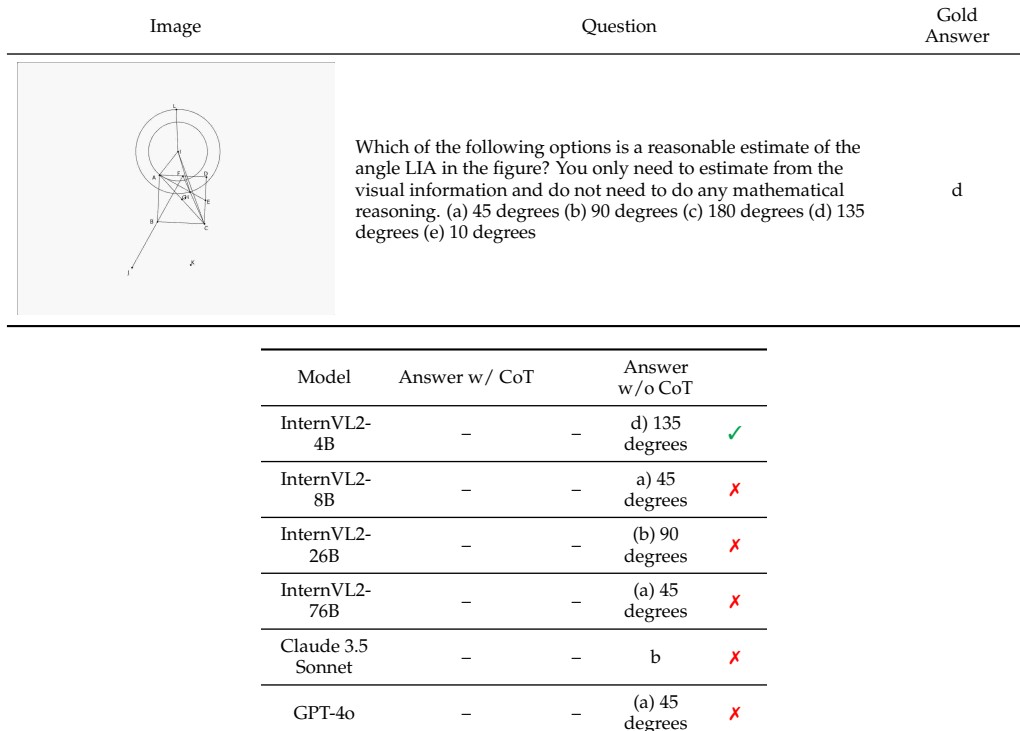 | Which of the following options is a reasonable estimate of the angle LIA in the figure? You only need to estimate from the visual information and do not need to do any mathematical reasoning. (a) 45 degrees (b) 90 degrees (c) 180 degrees (d) 135 degrees (e) 10 degrees | d |

| Model | Answer w/ CoT | Answer w/o CoT | |
|---|---|---|---|
| InternVL2-4B | – | – | d) 135 degrees ✓ |
| InternVL2-8B | – | – | a) 45 degrees ✗ |
| InternVL2-26B | – | – | (b) 90 degrees ✗ |
| InternVL2-76B | – | – | (a) 45 degrees ✗ |
| Claude 3.5 Sonnet | – | – | b ✗ |
| GPT-4o | – | – | (a) 45 degrees ✗ |
| Gemini 1.5 Pro | – | – | a ✗ |

Table 55: Examples of dataset and model responses for Syntheticgeometry-Angle (2)

| Image | Question | Gold Answer |
|-------|----------|-------------|
| 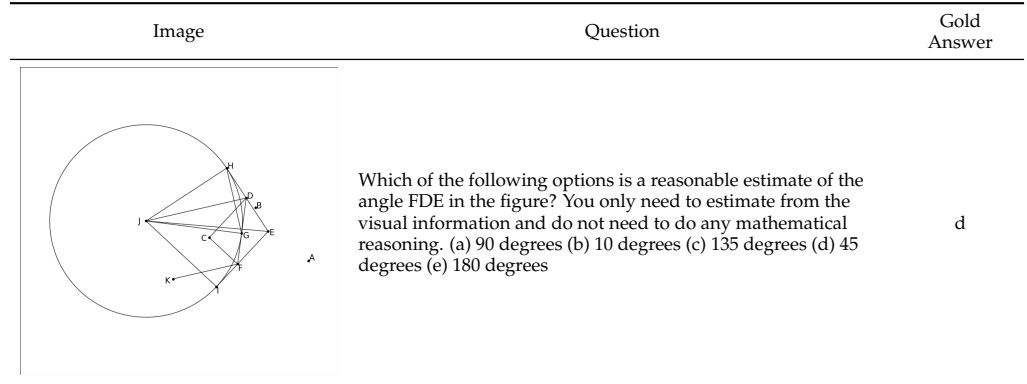 | Which of the following options is a reasonable estimate of the angle FDE in the figure? You only need to estimate from the visual information and do not need to do any mathematical reasoning. (a) 90 degrees (b) 10 degrees (c) 135 degrees (d) 45 degrees (e) 180 degrees | d |

| Model | Answer w/ CoT | Answer w/o CoT | |
|-------|---------------|----------------|---|
| InternVL2-4B | – | – | d) 45 degrees ✓ |
| InternVL2-8B | – | – | a) 90 degrees ✗ |
| InternVL2-26B | – | – | (a) 90 degrees ✗ |
| InternVL2-76B | – | – | (d) 45 degrees ✓ |
| Claude 3.5 Sonnet | – | – | d ✓ |
| GPT-4o | – | – | (d) 45 degrees ✓ |
| Gemini 1.5 Pro | – | – | d ✓ |

Table 56: Examples of dataset and model responses for `Syntheticgeometry-Angle` (3)

| Image | Question | Gold Answer |
|---|---|---|
| 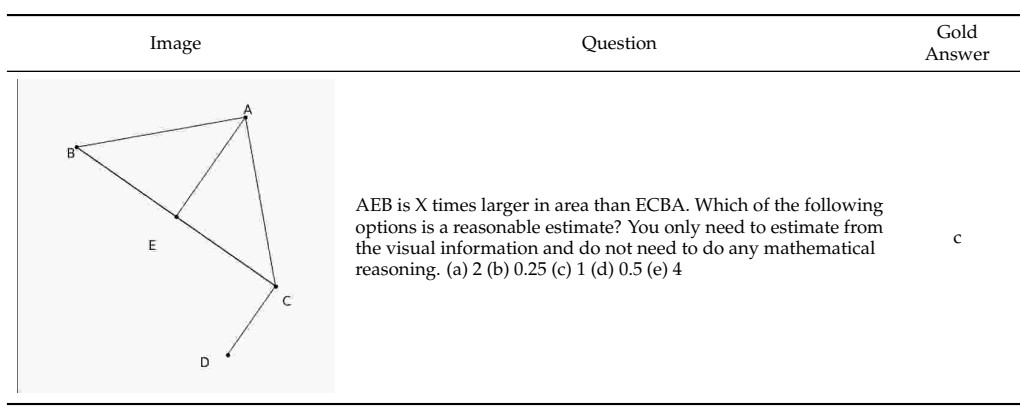 | AEB is X times larger in area than ECBA. Which of the following options is a reasonable estimate? You only need to estimate from the visual information and do not need to do any mathematical reasoning. (a) 2 (b) 0.25 (c) 1 (d) 0.5 (e) 4 | c |

| Model | Answer w/ CoT | Answer w/o CoT | |
|---|---|---|---|
| InternVL2-4B | – | – | e ✗ |
| InternVL2-8B | – | – | e ✗ |
| InternVL2-26B | – | – | (a) 2 ✗ |
| InternVL2-76B | – | – | (d) 0.5 ✗ |
| Claude 3.5 Sonnet | – | – | c ✓ |
| GPT-4o | – | – | (a) 2 ✗ |
| Gemini 1.5 Pro | – | – | d ✗ |

Table 57: Examples of dataset and model responses for `Syntheticgeometry-Area` (1)

| Image | Question | Gold Answer |
|---|---|---|
| 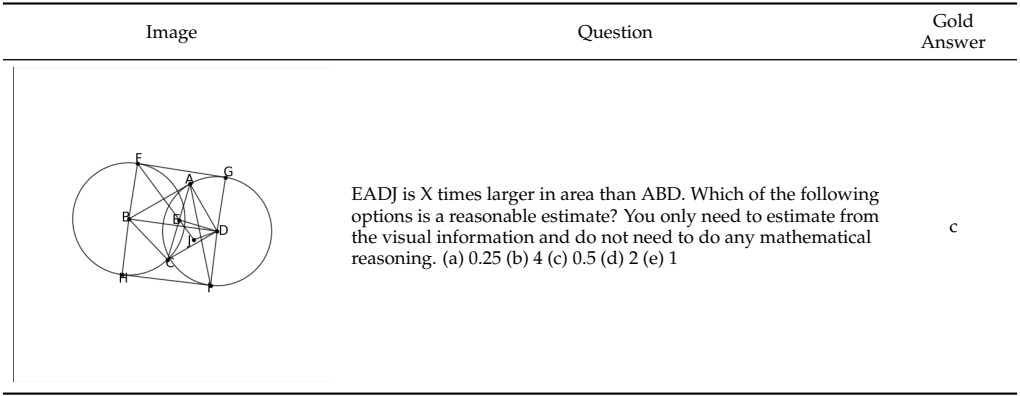 | EADJ is X times larger in area than ABD. Which of the following options is a reasonable estimate? You only need to estimate from the visual information and do not need to do any mathematical reasoning. (a) 0.25 (b) 4 (c) 0.5 (d) 2 (e) 1 | c |

| Model | Answer w/ CoT | | Answer w/o CoT | |
|---|---|---|---|---|
| InternVL2-4B | – | – | d | ✗ |
| InternVL2-8B | – | – | b | ✗ |
| InternVL2-26B | – | – | (b) 4 | ✗ |
| InternVL2-76B | – | – | (b) 4 | ✗ |
| Claude 3.5 Sonnet | – | – | d | ✗ |
| GPT-4o | – | – | e | ✗ |
| Gemini 1.5 Pro | – | – | c | ✓ |

Table 58: Examples of dataset and model responses for `Syntheticgeometry-Area` (2)

| Image | Question | Gold Answer |
|---|---|---|
| 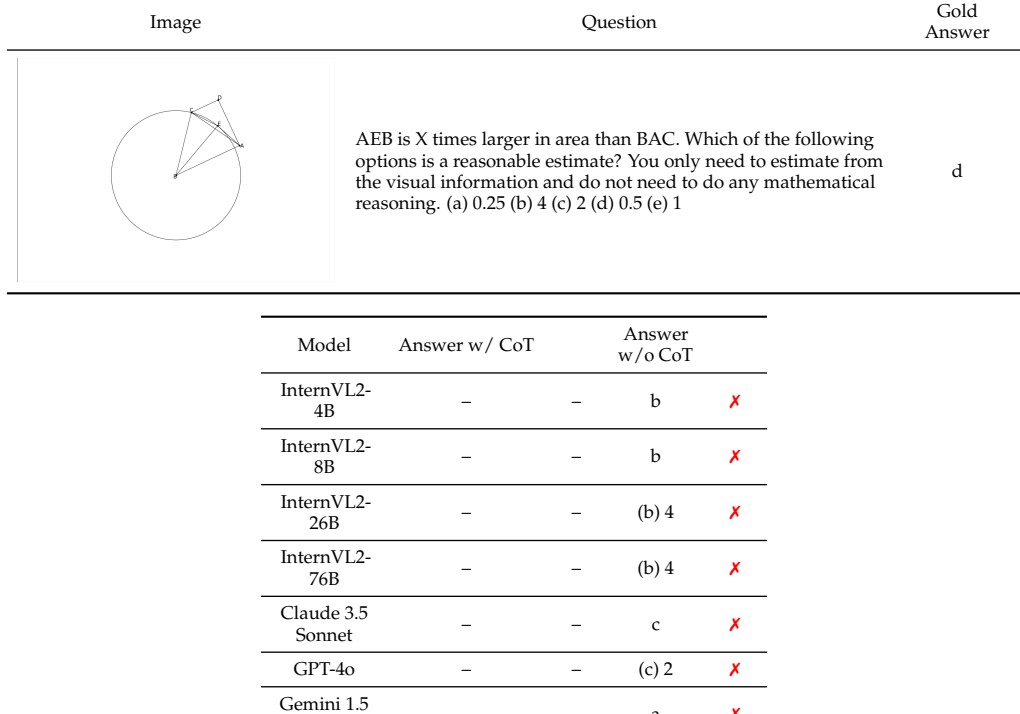 | AEB is X times larger in area than BAC. Which of the following options is a reasonable estimate? You only need to estimate from the visual information and do not need to do any mathematical reasoning. (a) 0.25 (b) 4 (c) 2 (d) 0.5 (e) 1 | d |

| Model | Answer w/ CoT | | Answer w/o CoT | |
|---|---|---|---|---|
| InternVL2-4B | – | – | b | ✗ |
| InternVL2-8B | – | – | b | ✗ |
| InternVL2-26B | – | – | (b) 4 | ✗ |
| InternVL2-76B | – | – | (b) 4 | ✗ |
| Claude 3.5 Sonnet | – | – | c | ✗ |
| GPT-4o | – | – | (c) 2 | ✗ |
| Gemini 1.5 Pro | – | – | a | ✗ |

Table 59: Examples of dataset and model responses for Syntheticgeometry-Area (3)

| Image | Question | Gold Answer |
|-------|----------|-------------|
| 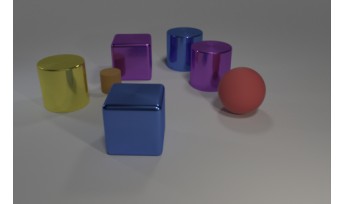 | The blue cylinder is X times taller than the red sphere. Which of the following options is a reasonable estimate? (a) 2 (b) 0.5 (c) 1 | c |

| Model | Answer w/ CoT | | Answer w/o CoT | |
|-------|---------------|---|----------------|---|
| InternVL2-4B | – | – | a | ✗ |
| InternVL2-8B | – | – | b) 0.5 | ✗ |
| InternVL2-26B | – | – | (c) 1 | ✓ |
| InternVL2-76B | – | – | (a) 1 | ✗ |
| Claude 3.5 Sonnet | – | – | a | ✗ |
| GPT-4o | – | – | a | ✗ |
| Gemini 1.5 Pro | – | – | a | ✗ |

Table 60: Examples of dataset and model responses for `3D-Size` (1)

| Image | Question | Gold Answer |
|-------|----------|-------------|
| 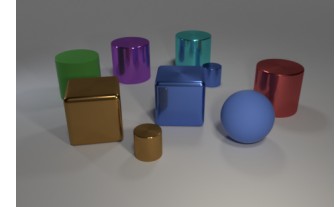 | The cyan cylinder is X times taller than the brown cylinder. Which of the following options is a reasonable estimate? (a) 0.5 (b) 1 (c) 2 | c |

| Model | Answer w/ CoT | | Answer w/o CoT | |
|-------|---------------|---|----------------|---|
| InternVL2-4B | – | – | c | ✓ |
| InternVL2-8B | – | – | (b) 1 | ✗ |
| InternVL2-26B | – | – | (c) 2 | ✓ |
| InternVL2-76B | – | – | (c) 2 | ✓ |
| Claude 3.5 Sonnet | – | – | c | ✓ |
| GPT-4o | – | – | c | ✓ |
| Gemini 1.5 Pro | – | – | c | ✓ |

Table 61: Examples of dataset and model responses for `3D-Size` (2)

| Image | Question | Gold Answer |
|---|---|---|
| 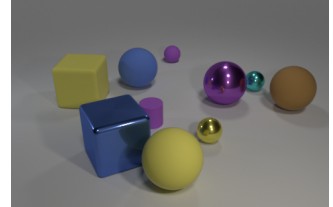 | The yellow rubber sphere is X times wider than the yellow cube. Which of the following options is a reasonable estimate? (a) 1 (b) 0.5 (c) 2 | a |

| Model | Answer w/ CoT | | Answer w/o CoT | |
|---|---|---|---|---|
| InternVL2-4B | – | – | b | ✗ |
| InternVL2-8B | – | – | b) 0.5 | ✗ |
| InternVL2-26B | – | – | (a) 1 | ✓ |
| InternVL2-76B | – | – | (a) 1 | ✓ |
| Claude 3.5 Sonnet | – | – | b | ✗ |
| GPT-4o | – | – | a | ✓ |
| Gemini 1.5 Pro | – | – | a | ✓ |

Table 62: Examples of dataset and model responses for `3D-Size` (3)

| Image | Question | Gold Answer |
|---|---|---|
| 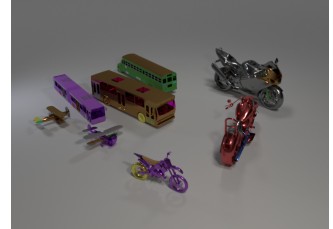 | Which of the following options is a reasonable estimate of the angle between the directions of the gray motorcycle and the brown airplane in the figure? We define 0 degrees if the direction is the same and 180 degrees if the direction is opposite. (a) 180 degrees (b) 45 degrees (c) 0 degrees (d) 135 degrees (e) 90 degrees | e |

| Model | Answer w/ CoT | | Answer w/o CoT | |
|---|---|---|---|---|
| InternVL2-4B | – | | a | ✗ |
| InternVL2-8B | – | | e | ✓ |
| InternVL2-26B | – | | (a) 180 degrees | ✗ |
| InternVL2-76B | – | | (e) 90 degrees | ✓ |
| Claude 3.5 Sonnet | – | | d | ✗ |
| GPT-4o | – | | e | ✓ |
| Gemini 1.5 Pro | – | | d | ✗ |

Table 63: Examples of dataset and model responses for 3D-Angle (1)

| Image | Question | Gold Answer |
|---|---|---|
| 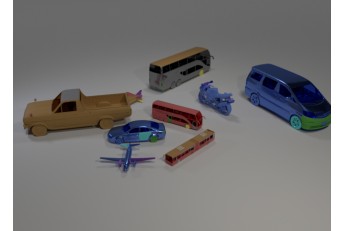 | Which of the following options is a reasonable estimate of the angle between the directions of the left blue car and the fighter jet in the figure? We define 0 degrees if the direction is the same and 180 degrees if the direction is opposite. (a) 90 degrees (b) 135 degrees (c) 180 degrees (d) 0 degrees (e) 45 degrees | d |

| Model | Answer w/ CoT | | Answer w/o CoT | |
|---|---|---|---|---|
| InternVL2-4B | – | – | e | ✗ |
| InternVL2-8B | – | – | b) 135 degrees | ✗ |
| InternVL2-26B | – | – | (a) 90 degrees | ✗ |
| InternVL2-76B | – | – | (a) 90 degrees | ✗ |
| Claude 3.5 Sonnet | – | – | e | ✗ |
| GPT-4o | – | – | b | ✗ |
| Gemini 1.5 Pro | – | – | b | ✗ |

Table 64: Examples of dataset and model responses for 3D-Angle (2)

| Image | Question | Gold Answer |
|---|---|---|
| 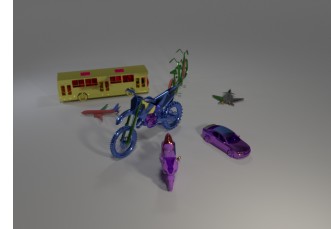 | Which of the following options is a reasonable estimate of the angle between the directions of the airplane and the purple motorcycle in the figure? We define 0 degrees if the direction is the same and 180 degrees if the direction is opposite. (a) 90 degrees (b) 180 degrees (c) 135 degrees (d) 45 degrees (e) 0 degrees | a |

| Model | Answer w/ CoT | | Answer w/o CoT | |
|---|---|---|---|---|
| InternVL2-4B | – | – | a | ✓ |
| InternVL2-8B | – | – | a | ✓ |
| InternVL2-26B | – | – | (a) 90 degrees | ✓ |
| InternVL2-76B | – | – | (a) 90 degrees | ✓ |
| Claude 3.5 Sonnet | – | – | c | ✗ |
| GPT-4o | – | – | d | ✗ |
| Gemini 1.5 Pro | – | – | a | ✓ |

Table 65: Examples of dataset and model responses for `3D-Angle` (3)

