# OpenReview forum: "VisOnlyQA: Large Vision Language Models Still Struggle with Visual Perception of Geometric Information"
_colmweb.org/COLM/2025/Conference — COLM 2025_

### Official Review · Reviewer_v9QR · 2025-05-11

**Rating:** 7
**Confidence:** 4
**Ethics Flag:** 1

**Summary:**

This paper introduces the VisOnlyQA benchmark, which is designed to test how well LVLMs understand basic geometric properties, without requiring complex reasoning or background knowledge. The authors evaluate a wide range of models and find that even the strongest ones perform poorly on these tasks. They also explore whether adding more training data or using larger models helps, and provide useful findings.

**Questions To Authors:**

1. In the VisOnlyQA-Eval-Real results, InternVL2-26B performs worse than InternVL2-4B, even though it has a larger vision encoder. Could the authors explain this unexpected result?
2. What is the confidence interval for each reported evaluation score? When the paper claims that performance is close to random guessing on certain tasks, what confidence level is being used to support that claim?

**Reasons To Accept:**

1. The paper is well motivated and easy to follow. It raises an important and specific question about the ability of LVLMs to understand geometric information.
2. The benchmark is carefully designed to focus only on geometric perception. By reducing the need for reasoning or background knowledge, it allows for a more focused and controlled evaluation.
3. The experiments include a broad set of both open-source and proprietary models, along with ablation studies that explore different factors.
4. The results show that many current LVLMs have trouble with basic geometric understanding, and that improvements through fine-tuning or larger model size are limited and vary by task.

**Reasons To Reject:**

1. Although the paper shows that models perform poorly, it doesn’t go into detail about why current architectures struggle with geometric perception. A deeper analysis of possible bottlenecks—like transformer attention maps or patch sizes—would strengthen the paper.
2. It’s unclear how much the vision encoder contributes to the task compared to the language model. One possible reason for the poor performance could be that vision encoders haven’t seen enough geometric data during training. It would help to run separate fine-tuning experiments on just the vision encoder or just the language model to better understand their individual roles.
3. While the paper does try to balance real and synthetic data, the fine-tuning relies mostly on synthetic examples. It’s not clear if this creates a bias between training and evaluation, or how well the improvements would transfer to real-world tasks.

---

> ### Author Response · Authors · 2025-06-03
> **Re: Official Review of Submission55 by Reviewer v9QR (1)**
>
> We sincerely appreciate your insightful comments and valuable feedback!
>
> > A deeper analysis of possible bottlenecks—like transformer attention maps or patch sizes—would strengthen the paper.
>
> In Section 4.4, we discuss that LLMs can be a bottleneck for visual perception in LVLMs. We show that LVLMs with larger LLMs exhibit better visual perception, which is a novel and counterintuitive observation.
>
> For further analysis, we use [VLM-Visualizer](https://github.com/zjysteven/VLM-Visualizer) to visualize attention maps of LLaVA-1.5-7B on the Angle and Length tasks. We observe that attention maps on images often focus on text labels (e.g., A, B, C) and show weaker attention on geometric shapes such as lines. It is possible that the current architecture or pretraining is overly optimized for specific visual information like text and less suited for understanding geometric shapes. We will include more analysis of intermediate behaviors of LVLMs on our dataset in the final version of the paper.
>
> We also agree that further analysis of architectural parameters, such as patch sizes, would provide insights into model bottlenecks. While such ablation studies require pretraining LVLMs with different architectures from scratch, which is not feasible with our resources, we consider it a promising direction for future research.
>
> > One possible reason for the poor performance could be that vision encoders haven’t seen enough geometric data during training. It would help to run separate fine-tuning experiments on just the vision encoder or just the language model to better understand their individual roles.
>
> We agree that the lack of training data is a possible reason for the poor performance, and this consideration motivated our experiments to fine-tune LVLMs on our dataset (Line 275). However, our experiment does not show consistent improvements even after fine-tuning, suggesting that the lack of training data is not the only reason for the weak geometric perception of LVLMs (Line 307).
>
> **(Updated on June 4th)**
>
> Although our experiments suggest that additional training does not fully solve this issue, we agree that fine-tuning just the vision encoder or just the language model would provide further insights. We conducted an additional experiment on InternVL2-8B in the Length and Area tasks, where our original fine-tuning reasonably improved performance. Our original setting in the paper fine-tunes LLMs of LVLMs, which is a popular approach and is the default setting of the InternVL code published by the authors.
>
> The results, shown in the following table, indicate that fine-tuning LLMs leads to greater performance gains than fine-tuning ViTs, particularly on out-of-distribution (real) figures. This finding reinforces our claim that LLMs play a critical role in geometric perception, even though the task does not require complex reasoning or external knowledge. We will include more results in the final version of the paper.
>
> | Fine-tuning | Length (Synthetic) | Length (Real) | Area (Synthetic) | Area (Real) |
> |----------------|--------------------------|-----------------------|-----------------------|-----------------------|
> | w/o Fine-tuning | 21.0 | 29.0 | 23.0 | 27.0 |
> | ViT Only            | 62.0 | 47.0 | 46.0 | 29.0 |
> | LLM Only          | 78.0 | 70.0 | 66.0 | 45.0 |
> | All Parameters  | 83.0 | 71.0 | 68.0 | 42.0 |
>
> > While the paper does try to balance real and synthetic data, the fine-tuning relies mostly on synthetic examples. It’s not clear if this creates a bias between training and evaluation, or how well the improvements would transfer to real-world tasks.
>
> We evaluate the fine-tuned models on both synthetic and real figures, which correspond to in-distribution and out-of-distribution data. We designed this experimental setup to avoid potential evaluation biases. We observe that fine-tuned LVLMs often perform poorly even on in-distribution data (synthetic figures), although they also exhibit a certain degree of generalization to out-of-distribution (real) figures in some tasks (Figure 5).

---

> > ### Comment · Reviewer_v9QR · 2025-06-04
> >
> > Thanks for the explanation. Looking forward to your ablation results of fine-tuning LLM/vision encoder only if time allows.
> >
> > I have a follow-up question regarding
> > >> While the paper does try to balance real and synthetic data, the fine-tuning relies mostly on synthetic examples. It’s not clear if this creates a bias between training and evaluation, or how well the improvements would transfer to real-world tasks.
> >
> > > We evaluate the fine-tuned models on both synthetic and real figures, which correspond to in-distribution and out-of-distribution data. We designed this experimental setup to avoid potential evaluation biases. We observe that fine-tuned LVLMs often perform poorly even on in-distribution data (synthetic figures), although they also exhibit a certain degree of generalization to out-of-distribution (real) figures in some tasks (Figure 5).
> >
> > Could you clarify in Table 2, what's the proportion of synthetic to real images in VisOnlyQA-Train?

---

> > ### Author Response · Authors · 2025-06-05
> > **Re: Official Comment by Reviewer v9QR**
> >
> > Thank you for your kind response! **The ablation results have been added to my previous response.** Kindly refer to the updated version for further information.
> >
> > > Could you clarify in Table 2, what's the proportion of synthetic to real images in VisOnlyQA-Train?
> >
> > Thank you for your question. VisOnlyQA-Train includes only synthetic figures because collecting and annotating a large number of real-world figures for training is challenging. In our experiments, we adopt a setting where LVLMs are trained on synthetic data and evaluated on real-world figures, which reflects a realistic scenario in the development of LVLMs.

---

> > > ### Author Response · Authors · 2025-06-09
> > > **Kind Reminder**
> > >
> > > This is a kind reminder that I have included the additional results in my initial response and have answered your follow-up question in my previous message. Please feel free to let me know if you have any further questions!

---

> > > > ### Comment · Reviewer_v9QR · 2025-06-11
> > > >
> > > > Thank you for running the additional experiment. Your response addresses most of my concerns, so I’ve updated the score to 7.

---

> > > > > ### Author Response · Authors · 2025-06-11
> > > > > **Re: Official Comment by Reviewer v9QR**
> > > > >
> > > > > Thank you for your response and for updating the score! We are glad our reply addressed most of your concerns. We will continue to improve the paper by reflecting on your feedback. We sincerely appreciate your constructive review and engagement in this discussion.

---

> ### Author Response · Authors · 2025-06-03
> **Re: Official Review of Submission55 by Reviewer v9QR (2)**
>
> > InternVL2-26B performs worse than InternVL2-4B, even though it has a larger vision encoder
>
> InternVL2-26B performs worse than InternVL2-8B on average in VisOnlyQA-Eval-Real. Although the performance difference is small and not statistically significant, there is indeed no improvement from the 8B to the 26B model. As a possible explanation, the two models use largely different architectures with different ViTs and LLMs, as shown in the following table. InternVL2-26B, despite being larger, may have an architecture that is less well-suited for the geometric perception tasks in the Real split.
>
> | Model           | Vision Encoder              | Language Model           |
> |----------------|-----------------------------|--------------------------|
> | InternVL2-8B     | InternViT-300M-448px          | internlm2_5-7b-chat      |
> | InternVL2-26B   | InternViT-6B-448px-V1-5      | internlm2-chat-20b        |
>
> > When the paper claims that performance is close to random guessing on certain tasks, what confidence level is being used to support that claim?
>
> In response to your suggestion, we conducted a paired bootstrap test (n = 1000) and computed the 95% confidence intervals of the difference between model performance and random performance (model performance - random performance). This is a popular statistical method for comparing model performance against baseline performance (e.g., Table 2 in [this paper](https://arxiv.org/abs/2501.19393)).
>
> The results for InternVL2 76B and Gemini 1.5 Pro on the Real split are shown in the table below. We observe that, for the Triangle, Quadrilateral, Chemistry (s), and Intersection tasks, the confidence intervals for both models include negative values, indicating that the improvement over random performance is not statistically significant. We will include this result in the final version.
>
> | Model             | Triangle       | Quadrilateral   | Length        | Angle         | Area          | Diameter | Chemistry (s)   | Chemistry (m)    | Extraction     | Intersection   |
> |------------------|----------------|------------------|----------------|----------------|----------------|------------------|----------------|----------------|----------------|----------------|
> | InternVL2-76B  | [-0.21, 0.08] | [-0.22, 0.06]   | [-0.04, 0.19] | [0.02, 0.26]  | [0.11, 0.37]  | [-0.08, 0.20]     | [-0.08, 0.28] | [0.08, 0.44]  | [0.31, 0.55]  | [-0.10, 0.17] |
> | Gemini 1.5 Pro | [-0.18, 0.11] | [-0.10, 0.16]   | [0.01, 0.25]  | [0.07, 0.32]  | [0.20, 0.46]  | [0.07, 0.33]      | [-0.08, 0.32] | [0.26, 0.58]  | [0.34, 0.66]  | [-0.12, 0.28] |

---

### Official Review · Reviewer_EwqX · 2025-05-12

**Rating:** 7
**Confidence:** 3
**Ethics Flag:** 1

**Summary:**

The paper addresses a gap in evaluating how effectively Large Vision-Language Models (LVLMs) perceive geometric information in images, which includes basic properties like shape, angle, and size. The authors introduce "VisOnlyQA," a dataset specifically created to evaluate LVLMs' geometric perception capabilities through 12 distinct tasks involving shapes, charts, chemical structures, and 3D shapes.

**Reasons To Accept:**

- The proposed benchmark is challenging for state-of-the-art LVLMs, including proprietary models, while humans achieve nearly perfect performance.

- The authors have conducted extensive experiments to demonstrate the effectiveness of their proposed benchmark.

**Reasons To Reject:**

- I wonder whether there is any selection bias in the evaluation setting. Since the answer formats in the proposed benchmark are typically True/False or Multiple-Choice Questions (MCQA), the authors need to clarify if the models tend to favor certain answers (e.g., consistently choosing "True" or option (a)) or if they genuinely reason about the provided images and queries.

- This is a minor question: I would like to see additional results from more advanced models, such as Gemini-2.5-Pro and Qwen2.5-VL-72B.

- As shown in Figure 4, there is a substantial portion of perception errors, which is understandable. However, I would appreciate further discussion on addressing these visual perception errors—specifically, what factors other than the LLM's capabilities (discussed in Table 5) might significantly influence performance.

---

> ### Author Response · Authors · 2025-06-03
> **Re: Official Review of Submission55 by Reviewer EwqX**
>
> We sincerely appreciate your insightful comments and valuable feedback!
>
> > I wonder whether there is any selection bias in the evaluation setting.
>
> We designed our dataset to remove the influence of selection bias (Line 173). Specifically, we include an equal number of questions for each answer option and shuffle the order of the options for multiple-choice questions. For True/False questions, we include positive and negative versions of the questions (e.g., "There is a triangle ABC in this figure" and "There is **no** triangle ABC in this figure") for all questions to eliminate the influence of model preference toward a particular answer (True or False).
>
> > I would like to see additional results from more advanced models, such as Gemini-2.5-Pro and Qwen2.5-VL-72B.
>
> Thanks for your suggestion. We conducted an additional evaluation and reported the results for the recent models on the Real split in the following table. Gemini 2.5 Pro (preview-05-06) exhibits large improvements in the diameter, chemistry, and chart tasks, narrowing the gap with human performance. However, its performance on most tasks involving geometric shapes remains considerably lower than human performance. We will include more results on recent models in the final version.
>
> | Model           | Triangle | Quadrilateral | Length | Angle | Area | Diameter | Chemistry (s) | Chemistry (m) | Extraction | Intersection | Average |
> |----------------|----------|----------------|--------|-------|------|----------|------------|------------|-------------|---------------|-------|
> | Qwen2-VL-72B    | 44.0     | 52.0           | 27.0   | 27.0  | 37.0 | 61.0     | 56.0       | 36.0       | 53.0        | 53.0          | 44.4  |
> | Qwen2.5-VL-72B  | 48.0     | 52.0           | 33.0   | 37.0  | 50.0 | 64.0     | 62.0       | 40.0       | 53.0        | 53.0          | 49.0  |
> | Gemini 1.5 Pro  | 47.0     | 53.0           | 33.0   | 40.0  | 53.0 | 70.0     | 62.0       | 52.0       | 67.0        | 53.0          | 52.6  |
> | Gemini 2.5 Pro  | 66.0     | 52.0           | 55.0   | 59.0  | 56.0 | 90.0     | 92.0       | 88.0       | 86.0        | 72.0          | 79.0  |
> | Human           | 96.7     | 90.0           | 93.3   | 93.3  | 86.7 | 100.0    | 93.3       | 93.0       | 93.3        | 95.0          | 93.5  |
>
>
> > what factors other than the LLM's capabilities (discussed in Table 5) might significantly influence performance
>
> Thank you for the suggestion. Our experiments show that training data and the capabilities of LLMs largely influence the geometric perception capabilities of LVLMs. However, we observe that even fine-tuned models and large models still often exhibit poor performance, indicating that simply scaling the training data or parameter size may not lead to human-level performance (Line 214). This observation implies that further improvements may require fundamental updates to the architecture of LVLMs. For example, in visual encoders, the process of splitting images into tiles may negatively affect their geometric perception.
>
> In addition, as shown in our response to Reviewer PGab, we observe that LVLMs exhibit poor geometric perception even for very simple geometric shapes consisting of only two or three lines. Another analysis, presented in our response to Reviewer v9QR, shows that the attention maps of LVLMs do not effectively focus on the target geometric elements. These additional findings further suggest that the architecture of current LVLMs has fundamental limitations both in visual encoders and LLMs.

---

> > ### Comment · Reviewer_EwqX · 2025-06-05
> > **Official Comment by Reviewer**
> >
> > Thanks for addressing my question. It will be very helpful in understanding your work.

---

> > > ### Author Response · Authors · 2025-06-06
> > > **Re: Official Comment by Reviewer**
> > >
> > > I appreciate your kind response! Please let me know if you have any further questions.

---

### Official Review · Reviewer_PGab · 2025-05-13

**Rating:** 4
**Confidence:** 5
**Ethics Flag:** 1

**Summary:**

This paper proposes VisOnlyQA, a dataset designed for evaluating the geometric perception capabilities of large vision-language models (LVLMs). VisOnlyQA consists of 12 tasks for various aspects of geometric perception ability. The figures in VisOnlyQA are sourced from existing datasets or synthesized using existing codebases. The authors invested effort in designing the questions. Experimental results indicate that current LVLMs exhibit poor performance in geometric perception. Even after fine-tuning, a gap remains compared to human performance.

**Questions To Authors:**

Please see above.

**Reasons To Accept:**

- Research on the geometric perception capabilities of LVLMs is an important and fundamental problem.
- The authors tested a wide range of LVLMs (20 models).
- It's great to see the experiments regarding chain-of-thought to verify if reasoning ability is necessary.

**Reasons To Reject:**

- Given that all figures come from either existing works or existing projects, I believe the authors should demonstrate a significant effort in designing the textual questions. However, the current writing does not provide sufficient detail to reflect such effort. Below, I list some potential questions that should be addressed. Additionally, there are likely other aspects the authors should consider to make the proposed dataset more convincing:
  - What criteria were used to design the questions?
  - How is the difficulty level of the questions controlled? How to control the difficulty level?
  - What is the quality of the designed questions? Do they always make sense? I do not see any description of a quality verification process or filtering.
- Given that the proposed dataset is challenging for most LVLMs, even after fine-tuning, I suggest that the authors break down the questions into several simpler ones to better identify the bottlenecks in LVLM performance. For instance, all examples in Figure 1 seem to be quite complex. What if the figures were simplified and the questions focused on very basic tasks, such as comparing two angles or the lengths of two lines? Do LVLMs truly understand the concept of degrees? It's important to investigate the root cause of their poor performance. Can LVLMs recognize very simple geometric shapes? Perhaps they can, but they might fail at comparison, calculation, or at combining multiple concepts to solve a problem. I believe this work requires deeper insight, rather than merely presenting an evaluation.

---

> ### Author Response · Authors · 2025-06-03
> **Re: Official Review of Submission55 by Reviewer PGab (1)**
>
> We appreciate your detailed and constructive feedback!
>
> > I believe the authors should demonstrate a significant effort in designing the textual questions
>
> > What criteria were used to design the questions?
>
> Our main effort lies in designing questions that evaluate geometric perception independently from other capabilities. Evaluation of the geometric perception capability of LVLMs is a well-motivated and fundamental problem (Line 27), but existing datasets are not suitable for this purpose (Lines 31–40, Table 1). To address this challenge, our design criterion is to annotate questions that directly ask about various geometric properties in images (Line 123). The annotated questions focus on the fundamental ability of geometric perception, while being disentangled from complex reasoning or external knowledge (Line 124).
>
> > How is the difficulty level of the questions controlled? How to control the difficulty level?
>
> We designed our dataset to include geometric perception tasks across a range of complexity levels. We consider the task complexity to depend on both geometric shapes and textual questions, and tasks in our dataset can be categorized into three complexity levels as follows:
>
> * Simple -- Question: recognizing geometric information, Image: geometric shapes
>   * Geometry-Triangle, Quadrilateral, Angle, Diameter
> * Medium -- Question: *comparing* geometric information of multiple elements, Image: geometric shapes
>   * Geometry-Length, Areas
> * Complex -- Question: *comparing* geometric information of multiple elements, Image: complex scientific images
>   * Chemistry, Charts, 3D
>
> However, based on our experiments, we observe that the difficulty for current LVLMs does not necessarily correlate with the complexity of the tasks or images. For example, LVLMs exhibit near-random performance on the simple task of recognizing triangles, while recent models achieve better performance on more complex tasks involving chemistry figures.
>
> > What is the quality of the designed questions? Do they always make sense? I do not see any description of a quality verification process or filtering.
>
> The quality of our questions is ensured through three aspects. First, the questions are generated by an annotator and verified by another annotator. Second, after the dataset is finalized, we evaluated its quality through human performance (Table 4). Table 4 shows that human performance is nearly perfect, achieving 93.5% accuracy. Third, we conducted error analysis in Section 4.2 by manually analyzing whether LVLMs correctly interpret the questions (Question Understanding Error), and we observed that even relatively weak models correctly understand the questions in over 90% of the cases. These results indicate that the questions in VisOnlyQA are clear for humans and of high quality.
>
> In response to your concern, we also conducted an additional manual analysis to directly assess the quality of the questions. We manually evaluated 100 randomly selected cases in the Real split (10 cases from each task) and confirmed that all questions are correct and make sense. We will include this result in the final version of the paper.

---

> ### Author Response · Authors · 2025-06-03
> **Re: Official Review of Submission55 by Reviewer PGab (2)**
>
> > Given that the proposed dataset is challenging for most LVLMs, even after fine-tuning, I suggest that the authors break down the questions into several simpler ones to better identify the bottlenecks in LVLM performance.
>
> > What if the figures were simplified and the questions focused on very basic tasks, such as comparing two angles or the lengths of two lines?
>
> Thank you for the suggestion. We create a new dataset consisting of synthetic geometric shapes with different complexities. Here, we use the number of lines in geometric shapes as a surrogate for the complexity. We generate new synthetic images and questions for the Triangle, Length, and Angle tasks. Each subset includes geometric shapes with 2, 3, 4, 5, and 6 lines, and each subset consists of 50 instances.
>
> The results of InternVL2 76B and Gemini 1.5 Pro are shown in the following tables. We observe that the performance of LVLMs is not largely influenced by the complexity of geometric shapes. Notably, their performance remains very low in cases where the figures contain only two or three lines. This result corroborates our findings that current LVLMs have fundamental limitations and cannot correctly perceive basic geometric properties even in very simple geometric shapes.
>
> * InternVL2 76B
>
> | # of Lines | 2      | 3      | 4      | 5    | 6    |
> |---------------|-------|-------|------|------|------|
> | Triangle    |     --  | 50.0 | 52.0 | 50.0 | 50.0 |
> | Length      | 34.0 | 20.0 | 24.0 | 22.0 | 30.0 |
> | Angle        | 18.0 | 20.0 | 22.0 | 20.0 | 22.0 |
>
> * Gemini 1.5 Pro
>
> | # of Lines | 2    | 3    | 4    | 5    | 6    |
> |--------------|------|------|------|------|------|
> | Triangle   |     --  | 56.0 | 54.0 | 62.0 | 48.0 |
> | Length     | 38.0 | 42.0 | 42.0 | 44.0 | 44.0 |
> | Angle       | 30.0 | 24.0 | 30.0 | 26.0 | 22.0 |
>
>
> > Do LVLMs truly understand the concept of degrees?
>
> In section 4.2, we conducted a manual analysis of chain-of-thought reasoning generated by LVLMs. According to the reasoning articulated in their responses, Question Understanding Error is rare, indicating that LVLMs correctly understand the concept of geometric properties such as degrees asked in our dataset. In addition, we can expect the fine-tuned models to correctly understand the tasks they are asked to solve, but their performance often remains poor (Figure 5). These results suggest that current LVLMs exhibit limited geometric perception even when they correctly understand the underlying concepts and task requirements.

---

> > ### Comment · Reviewer_PGab · 2025-06-06
> >
> > Thanks for your reply. I appreciate your effort on the new dataset and the experiments. However, given that LVLMs perform poorly on the new dataset which emphasizes basic geometric reasoning, I believe the originally proposed dataset may not provide much insight, as it seem overly complicated. I suggest putting more focus on studying fundamental geometric properties, as these seem to be the underlying challenge. The originally proposed dataset could still be useful as a supplementary benchmark for harder cases once LVLMs can reliably solve simpler geometry problems.
> >
> > In addition, I believe the current human verification process still needs improvement. A 93.5% accuracy rate should not be considered near-perfect. Do you have any ideas about the 6.5% of examples where human errors occurred? For a benchmark dataset, I would expect all examples to be carefully designed and thoroughly verified. A typical data curation process includes multiple annotators per example, inter-annotator agreement checks, and additional filtering. While I understand that your dataset may require a slightly different process, substantial effort in verification is still essential.
> >
> > By the way, it would be interesting to include multiple questions with different difficulty for the same image.
> >
> > Given the reasons above, I will not change my score at this time.

---

> > > ### Author Response · Authors · 2025-06-07
> > > **Re: Official Comment by Reviewer PGab**
> > >
> > > Thank you for your response!
> > >
> > > > I believe the originally proposed dataset may not provide much insight, as it seem overly complicated
> > >
> > > We consider that the analysis conducted in response to your suggestion already supports the effectiveness of our dataset for evaluating the geometric perception abilities of LVLMs. Specifically, the result suggests that figure complexity does not have a large influence on the evaluation of geometric perception in the current LVLMs. For example, Gemini 1.5 Pro consistently achieves around 40% accuracy on the Length task across geometric shapes of varying complexity, with the number of lines ranging from 2 to 6, which is also close to its performance on our original dataset (34.0%). These results indicate that the figure complexity of our dataset does not negatively impact evaluation, supporting the validity of our original dataset for analyzing the geometric perception abilities of LVLMs.
> > >
> > > In summary, simple shapes are suitable for controlled analysis, and we will include more experiments in the final version. However, our original dataset can also effectively evaluate and analyze geometric perception in LVLMs on realistic and diverse figures. We consider both directions to be meaningful.
> > >
> > > > Do you have any ideas about the 6.5% of examples where human errors occurred?
> > >
> > > We observe that most human errors are due to their mistakes rather than the issues in the dataset. For example, in the Length task, which follows the format "Line BD is X times longer than DE", a common mistake is to answer "X = 4" when the correct answer is "X = 0.25". Another frequently observed error involves misreading the order of symbols (e.g., confusing ABC with ACB in the Angle task). We will clarify this point in the final version of the paper.
> > >
> > > We also note that the additional analysis presented in our previous response shows that all 100 randomly sampled cases are valid, indicating that our dataset is of very high quality.
> > >
> > > > By the way, it would be interesting to include multiple questions with different difficulty for the same image.
> > >
> > > Thank you for the suggestion. We agree that a more detailed analysis of questions with varying difficulty levels would provide further insights. As an example, we evaluated the Length task under different angles between two lines. The following table presents results on a new dataset composed of geometric shapes with three lines. We initially expected that comparing the lengths of two lines would be easier when the angle between them is 0. However, the result shows that the angle has a minor influence on the performance. Notably, although Gemini 1.5 Pro performs substantially better than random, its accuracy remains nearly constant across different angle conditions. This result suggests that current LVLMs face fundamental challenges in geometric perception, regardless of question difficulty. We will include additional results in the final version.
> > >
> > > | Angle between Two Lines | 0    | 45    | 90    |
> > > |--------------|------|------|------|
> > > | InternVL2 76B  | 24.0 | 16.0 | 22.0 |
> > > | Gemini 1.5 Pro | 36.0 | 38.0 | 36.0 |

---

> > > > ### Comment · Reviewer_PGab · 2025-06-11
> > > >
> > > > Thank you for your response.
> > > >
> > > > I still believe that a more comprehensive study and careful design of the questions is necessary, from simple shape understanding, to the composition of those shape concepts, and finally to the proposed more complex geometric tasks. This would provide a clearer overall understanding of how VLMs process geometric shapes and where the bottlenecks are. I appreciate the initial experiments on basic shapes provided during the rebuttal. I think this is a good start. With more carefully designed, controlled analyses along with the currently proposed dataset, this could become a strong work. However, from my perspective, the results on basic shape understanding and composition are still in an early stage and remain incomplete yet.
> > > >
> > > > On another note, I still think a more thorough quality check is needed, with detailed instructions provided and a more fine-grained analysis, especially given that multiple levels of difficulty and composition are expected. This quality check process should be conducted independently of the human performance evaluation.
> > > >
> > > > I will keep my original score.

---

> > > > > ### Author Response · Authors · 2025-06-11
> > > > > **Re: Official Comment by Reviewer PGab**
> > > > >
> > > > > Thank you for your response!
> > > > >
> > > > > While we believe that our dataset serves as an effective benchmark and that our analysis provides useful and novel insights, we understand that analyzing simpler geometric shapes is also crucial. We will include further analysis beyond what was presented in our previous responses.
> > > > >
> > > > > To clarify, the result of the quality check, which we included in our previous response, was obtained independently of the human performance evaluation. The manual quality check shows that all 100 randomly sampled responses contain valid questions and correct answers, indicating that the quality of our dataset is very high. We will ensure that this result is included in the final version of the paper.

---

### Decision · Program_Chairs · 2025-07-08

**Decision:**

Accept

**Comment:**

The paper proposes a benchmark for evaluating if vlms can understand geometric information highly relevant to figure understanding. For example, can models reason about angles, geometric relationships between objects, ect.  The dataset involves 12 distinct types of tasks, sourced from a combination of existing sources of images and new synthetic or human annotations on the question side. The paper provides a large analysis, showing there is significant room to improve, both for open source and api models.

Two reviewers found the paper well motivated, having good breadth, and that the evaluation points out a genuine missing capability in existing systems. The AC strongly agrees, and especially in the contexts where figure understanding is important, this benchmark points out desirable lower level capabilities.

One review somewhat disagreed about the value of the paper. They felt that if a benchmark finds models significantly lacking, it should be simplified until we have sufficient signal to diagnose the underlying problem. This would indeed be desirable and authors provided some more experiments in rebuttal on simplified data (although not enough to really diagnose why models struggle). The AC agrees this would desirable, but doesn't agree that all benchmarks that point out a significant shortcoming should also diagnose the source of it.